# Annual carbon emissions from land-use change in China from 1000 to 2019

Fan Yang[1,2,3], Guanpeng Dong[1,2,3], Xiaoyu Meng[1,2,3], Richard A. Houghton[4], Yang Gao[1,2,3], Fanneng He[5], Meijiao Li[6], Wenjin Li[1], Bing Li[1,2,3], Zhihao Liu[1], Qinqin Mao[7], Pengfei Wu[1], Yuanzhi Yao[8], Xudong Zhai[1], Hongjuan Zhang[1,2,3], Chao Yue[9]

[1]Key Research Institute of Yellow River Civilization and Sustainable Development, Henan University, Kaifeng 475001, China

[2]Faculty of Geographical Science and Engineering, Henan University, Zhengzhou, 450046, China;

[3]Laboratory of Climate Change Mitigation and Carbon Neutrality, Henan University, Zhengzhou 450046, China

[4]Woodwell Climate Research Center, Falmouth, MA 02540, USA

[5]Key Laboratory of Land Surface Pattern and Simulation, Institute of Geographic Sciences and Natural Resources Research, Chinese Academy of Sciences, Beijing 100101, China

[6]College of Resources and Environment, Shanxi University of Finance and Economics, Taiyuan 030006, China

[7]Piesat Information Technology Co., Ltd., Xian, 710100, China

[8]School of Geographic Science, East China Normal University, Shanghai, 200241, China

[9]College of Natural Resources and Environment, Northwest Agriculture and Forestry University, Yangling 712100, China

*Correspondence to:* Guanpeng Dong (gpdong@vip.henu.edu.cn), Fanneng He (hefn@igsnrr.ac.cn), and Chao Yue (chaoyue@ms.iswc.ac.cn)

**Abstract.** Long-term land-use changes have a profound impact on terrestrial ecosystems and the associated carbon balance. Current estimates of China's historical carbon emissions induced by land-use change vary widely. Here, current mainland China was taken as the study area, and the 32 provincial units (excluding Macao and Hong Kong) were merged into 25 regions. We utilized a bookkeeping method to quantify China's annual carbon budget resulting from land-use change between 1000 and 2019, driven by a millennial dataset of land-use change in China at provincial level, assisted by comprehensive soil and vegetation carbon density datasets. This approach, which was supported by high-confidence land-use change data, a comprehensive carbon density database compiled from over 10,000 existing field samples, and the latest published disturbance-response curves, enhanced the accuracy of carbon budget estimates. The results revealed that cumulative carbon emissions from land-use change in China reached 19.61 Pg C over the past millennium. Moreover, critical turning points occurred in the early 18th century and early 1980s, with emissions accelerating in the 18th century and transitioning from carbon source to carbon sink in the early 1980s. Our findings revealed that the values were 68%–328% higher than the previous 300-year estimates, suggesting that historical carbon emissions from land-use change in China may have been significantly underestimated. This study provides a robust historical baseline for assessing both present and future terrestrial ecosystem carbon budgets at national and provincial scales. The dataset is available at https://doi.org/10.5281/zenodo.14557386 (Yang et al., 2025).

## 1 Introduction

Carbon fluxes from historical and current land-use change, including both gross emissions and sinks, globally constitute a net carbon source and represent a critical component of the global carbon budget (Houghton and Nassikas, 2017). Reversing land use practices that cause emissions can provide insights into the potential of land management to remove carbon from the atmosphere. Improved quantification of the carbon dynamics associated with land-use change is hence needed to provide a better understanding of the global carbon cycle and the future carbon sink potential of terrestrial ecosystems (Friedlingstein et al., 2023; Obermeier et al., 2024).

Although the estimated contemporary carbon emissions from land-use change account for only 10–15% of anthropogenic carbon emissions (Friedlingstein et al., 2022; Friedlingstein et al., 2020), their historical contributions were much higher. Land-use change has been estimated to contribute nearly 20 ppm to current atmospheric $CO_2$ concentrations, with this contribution dating back at least 1,000 years (Pongratz et al., 2009). Over the past 150 years, carbon emissions from land-use change have accounted for up to 33% of global anthropogenic carbon emissions (Houghton et al., 2012). Recent carbon accounting has shown that since 1750, land-use change has been a major source of $CO_2$ emissions, accounting for 54% of the cumulative $CO_2$ emissions from 1750 to 2020, with fossil fuel $CO_2$ emissions not surpassing those from land-use change until the mid-1960s (Dorgeist et al., 2024; Wedderburn-Bisshop, 2024). Furthermore, historical carbon emissions from land-use change provide crucial insights into how the global carbon cycle responds to environmental changes (Houghton and Castanho, 2023; Yue et al., 2020; Houghton and Nassikas, 2017).

Given the profound impact of land-use change, particularly over long timescales, numerous studies have focused on long-term global estimates of carbon emissions from land-use change (Houghton and Castanho, 2023; Mendelsohn and Sohngen, 2019; Houghton and Nassikas, 2017; Kaplan et al., 2011; Pongratz et al., 2009). However, uncertainties persist in these estimates (Winkler et al., 2023), with net land-use change carbon fluxes exhibiting the highest relative uncertainty in global carbon budget assessments (Friedlingstein et al., 2022). These uncertainties arise not only from differences in estimation models, parameters, and carbon density datasets but also from historical land-use change data. In particular, reliable land-use change datasets prior to the mid-20th century are often lacking for many countries, including China.

One typical approach to reconstructing historical land-use change is to use historical population data as a proxy combined with linear backcasting (Pongratz et al., 2008; Klein Goldewijk, 2001; Ramankutty and Foley, 1999). Although this method works reasonably well for estimating cropland and pasture areas, it is less suitable for calculating changes in forest cover, which has a high impact on estimated terrestrial ecosystem carbon budgets because of the higher carbon densities of forest ecosystems relative to that of cropland or grassland. As a result, researchers often subtract the area of cropland and pasture from the potential natural vegetation to estimate forest cover change (Hurtt et al., 2020; Klein Goldewijk et al., 2017; Pongratz et al., 2008; Ramankutty and Foley, 1999). However, this approach fails to capture large-scale forest cover change

driven by factors such as shifting cultivation, timber and fuel demand in addition to land conversion for agriculture. Consequently, this indirect method can only reflect the conversion relationship among forests, croplands, and pastures and thus often underestimates the actual extent of historical forest change. Therefore, linear backcasting or potential vegetation subtraction often introduces great uncertainties (Kabora et al., 2024; Yang et al., 2020; He et al., 2018) that are carried over into land-use carbon emission estimates.

China has a vast territory and a long history of land use, making it an important contributor to global terrestrial carbon dynamics caused by anthropogenic land-use change and land management. Although most global and regional studies on land-use change focus on the post-industrial era or the past three centuries, China's intensive and extensive land-use activities date back at least a millennium, thus representing a unique historical trajectory (He et al., 2025, 2023). From approximately AD 1000 (coinciding with the Northern Song Dynasty), ecological degradation in China showed a marked rise. This degradation was manifested through multiple pathways: accelerated erosion on the Loess Plateau, recurrent floods in the lower Yellow River Basin, large-scale lake siltation and disappearance in northern China, and progressive soil erosion coupled with natural vegetation loss in the southern hill regions (Wu et al., 2020; Chen et al., 2012). Such millennial-scale land-use transitions would have generated substantial carbon emissions, particularly from deforestation. However, the relatively stable pre-industrial global $CO_2$ concentrations likely obscured these regionally significant anthropogenic carbon fluxes because localized emissions in areas such as China could have been offset by concurrent carbon sinks elsewhere. Additionally, the full trajectory or specific stages of historical land-use change in China can serve as a "historical analogue" for other developing countries. For many countries and regions, systematically revealing the processes and mechanisms of land-use change and associated carbon emissions—driven by long-term population growth and policy shifts—can help overcome the limitations associated with a lack of historical records and reliance on static assumptions.

China has abundant historical documentation from a number of dynasties, such as tax records for cropland areas. Scholars have used these records to reconstruct long-term, high-confidence datasets of cropland areas, thereby providing a strong foundation for estimating historical land-use change carbon emissions. Previous studies have extensively reconstructed historical land use across China and specific regions (He et al., 2023; Jia et al., 2023; Wei et al., 2022; Yang et al., 2022; Yu et al., 2021; Li et al., 2016; Ye et al., 2009), as well as the associated carbon emissions (Yang et al., 2023; Yu et al., 2022; Yang et al., 2019; Li et al., 2014; Ge et al., 2008; Houghton and Hackler, 2003). However, existing estimates vary widely and exhibit great uncertainty. For example, estimates of cumulative net carbon emissions from land-use change in China from 1950 to 2021 based on three internationally recognized bookkeeping models exhibited a relative uncertainty of up to 150% (ratio of the standard deviation to the mean estimate) (Obermeier et al., 2024). Moreover, independent estimates of carbon emissions from land-use change over the past 300 years for China—though primarily derived from similar indigenous historical documents—still showed a relative uncertainty of 102% (Yang et al., 2023; Yang et al., 2019; Ge et al., 2008; Houghton and Hackler, 2003). Although uncertainty can be reduced by improving model selection and parameters, highly

reliable land-use change data remain crucial (Dorgeist et al., 2024; Yu et al., 2022).

This study aims to reconstruct the annual carbon emissions from land-use change in China from 1000 to 2019. To achieve this, we implement four specific improvements: (1) extending the analysis period from the commonly focused past 300 years to the millennial scale (1000–2019) by integrating newly published land-use reconstruction data (He et al., 2023; He et al., 2025) with the Second and Third National Land Surveys; (2) refining land-use conversion rules to clarify the attribution of deforestation beyond conversion to cropland; (3) enriching carbon density sampling data to enhance their representativeness; and (4) applying a bookkeeping model with updated disturbance-response curves to calculate long-term annual carbon fluxes, a method aligned with the IPCC and Global Carbon Project (GCP).

## 2. Material and methods

### 2.1 Study area

China's territorial and administrative boundaries have changed frequently over the past millennium, with the country experiencing a succession of different regimes, including the Liao, Song, Jin, Yuan, Ming, and Qing dynasties, the Republic of China, and the People's Republic of China (Tan, 1982). To facilitate the alignment of data across different historical periods, this study used the current land area of mainland China as the study region and adopted the territorial and administrative coordination scheme proposed by He et al. (2023) (Fig. 1), in which the 32 provincial units (excluding Macao and Hong Kong) were merged into 25 regions. This coordination scheme also serves as a fundamental spatial unit for historical land-use change data in China (cropland, forest, and grassland).

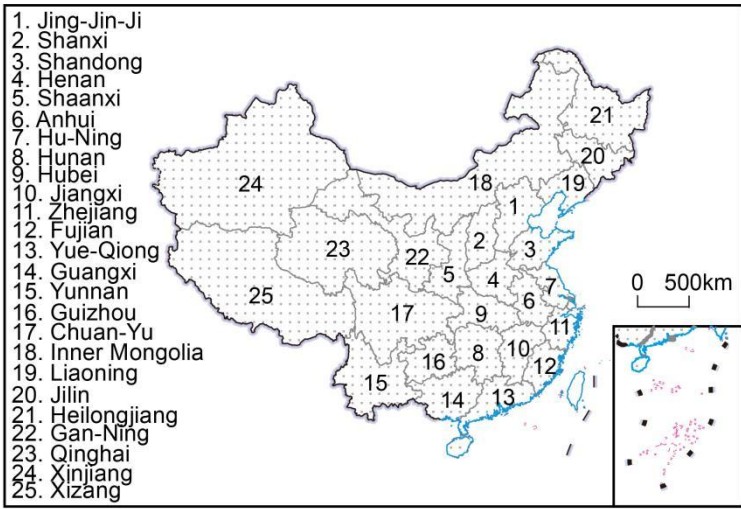

**Figure 1.** Map of the study area showing the 25 merged provincial-level administrative divisions of China. The following provincial-level administrative regions were merged: Beijing, Tianjin, and Hebei were merged into JingJin-Ji (No.1); Shanghai and Jiangsu were merged into Hu-Ning (No.7); Guangdong and Hainan were merged into Yue-Qiong (No.13); Sichuan and Chongqing were merged into Chuan-Yu (No.17); and Gansu and Ningxia were merged into Gan-Ning (No.22). Due to data limitations, this study did not include Taiwan Province.

**2.2 Data sources**

This study used two main types of data: long-term land-use data (cropland, forest, and grassland) and carbon density data (vegetation carbon density and soil carbon density).

**2.2.1 Land-use data**

Land-use data for the period 1000–2019, covering 131 time points, included both historical reconstruction data and survey-based statistics. For the period 1000–1999, provincial cropland data for China were obtained from several previous studies (Table 1). These data were primarily reconstructed for cropland areas using tax records in historical archives dating back to the Northern Song Dynasty (Yang et al., 2024; Li et al., 2020; Li et al., 2018a; Li et al., 2018b; Li et al., 2016; Ge et al., 2004). Provincial forest data for 1000–1998 were sourced from He et al. (2024, 2017, 2008) (Table 1) and are referenced to as historical deforestation data. Provincial grassland data for 1000–2000 were also obtained from He et al. (2024) (Table 1).

**Table 1.** Data sources for land-use change in China

| Data variables | Temporal coverage | Spatial resolution | Data type | Data source/ Reference |
|---|---|---|---|---|
| Cropland | 1000, 1066, 1078, 1162, 1215 | Province | Reconstruction | He et al. (2017) Li et al. (2018a) |
| | 1102 | Province | Reconstruction | Yang et al. (2024) |
| | 1290 | Province | Reconstruction | Li et al. (2018b) |
| | 1393, 1583, 1620 | Province | Reconstruction | Li et al. (2020) |
| | 1661–1949 (21 time points) | Province | Reconstruction | Ge et al. (2004) |
| | 1949–1999 (27 time points) | Province | Statistics | Li et al. (2016) |
| Forest | 1000–1949 (50-year interval) | Province | Reconstruction | He et al (2024) He et al (2008) |
| | 1962, 1976, 1981, 1988, 1993, 1998 | Province | Statistics | He et al (2015) |
| Grassland | 1000, 1100, 1200, 1300, 1400, 1500, 1600, 1700, 1800, 1900, 2000 | Province | Reconstruction | He et al (2024) |

This study used survey-based data from the Second National Land Survey (2009) and Third National Land Survey (2019) (Appendix Table A1) for the period after 2000. These surveys, conducted by the Chinese government, are considered highly credible. As large-scale national projects directed by the Chinese government, these surveys involved extensive, multi-year efforts and were subject to rigorous quality control throughout the entire process, ensuring their high credibility.

The 25 provinces shown in Fig 1 were used as spatial units for historical land-use data in China. Cropland, forest, and grassland data from the national land survey reports were adjusted according to this scheme to ensure consistency.

### 2.2.2 Overview of long-term land-use change data

Unlike modern geographic elements, which can be verified through techniques such as sample collection, field surveys, and remote sensing monitoring, historical land-use change data spanning long periods and large regions are difficult to independently validate because of temporal and spatial constraints. Our data encompass three land-use types: cropland, forest, and grassland, derived from multiple published studies. The reliability of these data is assessed through the examination of data sources, the rationality of the estimation or reconstruction methods, and the degree to which the results align with expert knowledge. Their quantitative changes (expansion and contraction) are consistent over the time series and have been cross-validated against population trends, dynastic policies, and documented historical events. Based on these data, we designed land-use conversion rules (Section 2.3.2) to integrate the independently reconstructed historical data of different land-use types. This integrated data were then used to drive subsequent carbon budget calculations. Historical land-use data for China from global datasets are known to have poor support from local expert knowledge and thus fail to capture more recent land-use dynamics (Yu et al., 2022). For this reason, we utilized regionally reconstructed historical land-use change data for China. We argue that the latter provides a more reliable representation of land-use trajectories in China over the past millennium. Below we further detail the rationale behind this choice.

The historical cropland data used in this study are typical examples of regionally reconstructed data. Historically, China has been a major agricultural nation, with agriculture forming the primary pillar of socioeconomic development in ancient Chinese society. Cropland area directly influences agricultural tax revenues, and as a result, tax records for cropland areas have been extensively documented in the historical literature, making them highly reliable accounts of cropland area. Furthermore, although these records may not precisely correspond to actual cropland, scholars have developed conversion mechanisms to convert tax records to actual cropland area across different historical periods. These methodologies have been used to reconstruct cropland areas over various periods (Yang et al., 2024; Li et al., 2020; Li et al., 2018a; Li et al., 2018b; Li et al., 2016; Ge et al., 2004) and the results have been peer-reviewed and published to ensure the reliability of the data sources, methods, and processes. Although global historical land-use datasets (such as the HYDE 3.2 dataset) have partly incorporated these regional reconstructions to reflect historical cropland changes at the national level for China, they are prone to error at provincial scale. Detailed analyses and assessments of the provincial errors in the global datasets have been performed by Zhao et al. (2022) and Fang et al. (2020).

Historical records of the forests in China are mainly scattered in various historical texts. While quantitatively reconstructing forest cover change based solely on literary sources is challenging, qualitative descriptions can be successfully generated. Accordingly, several key features of forest changes in China over time have been revealed: (1) northern China has a long history of deforestation and as early as a thousand years ago, forests in the North China Plain were

already nearly depleted; (2) over the following millennium, deforestation gradually expanded from plains and hills to mountainous areas; and (3) the deforestation process started from around the middle and lower reaches of the Yellow River and gradually extended to the middle and lower reaches of the Yangtze River, and then to the southern coastal areas of China, Southwest China, and Northeast China. These features provide crucial evidence for assessing the reliability of reconstructed forest data. By constructing a non-linear "inverted S-shaped" relationship between forest cover change and population size data, historical forest area changes used in this study were estimated based on qualitative records of deforestation in Chinese history (He et al., 2025). The "inverted S-shaped" curve reflects the dynamic relationship between historical population size and deforestation. In the early stages, when the population is relatively small, forest resources are plentiful and the rate of deforestation remains slow. As the population grows, deforestation accelerates rapidly, resulting in a significant loss of forest cover. Eventually, despite the population continuing to increase, the scarcity of remaining forests causes the rate of deforestation to slow down. In contrast, global historical land-use datasets depict historical forests in China by subtracting the area of cropland and pasture from the potential forest vegetation area in each grid cell simulated by vegetation modeling. Therefore, this approach primarily reflects the transition of forest cover to human land-use and fails to accurately capture other factors that influence forest area changes, such as fuelwood and timber consumption. For a detailed evaluation of historical forest data in global datasets for China, please refer to Yang et al. (2020).

For historical changes in grassland area, global datasets such as HYDE (Klein Goldewijk et al., 2017), SAGE (Ramankutty and Foley, 1999), and PJ (Pongratz et al., 2008) have been generated based on the FAO's definition of pasture. However, Chinese scholars use the plant geography definition of grassland. This conceptual difference is one of the major reasons for the large discrepancies in grassland area for China between global datasets and the reconstructions generated by Chinese scholars (He et al., 2018). Unlike Europe and North America, where climate-driven land-use patterns for livestock (grassland) dominate, China (especially in the eastern regions) has historically developed a cropland-based husbandry system under a monsoon climate and a relatively smaller-scale grassland agriculture system. Therefore, global datasets based on European and North American land-use practices, which use historical population and per capita pasture area as proxies to derive pasture or grassland data, are not applicable to China. For an evaluation of historical grassland data for China in global datasets, refer to He et al. (2018). Moreover, historical grassland cover data used in He et al. (2024) are based on historical cropland and forest data. These historical data consider the occupation of grassland by cropland expansion in western and northern China and also reflect the dynamic relationship between deforested land and secondary grasslands in eastern and southern China.

Overall, the long-term land-use data used in this study were based on historically reconstructed data rather than retrospective simulation data, with independent reconstructions performed for historical cropland and forest data (Yang et al., 2024; He et al. 2024, 2017, 2008; Li et al., 2020, 2018a, 2018b, 2016; Ge et al., 2004). Consequently, these reconstructed data are closer to historical facts and provide unique value for assessing the environmental effects of long-term human

land-use changes.

### 2.2.3 Carbon density data

This study compiled and harmonized a provincial soil and vegetation carbon density dataset for China, integrating 10,424

sample points from multiple major sources. Soil carbon density data were derived from the following three sources. (1) The

2010s China Land Ecosystem Carbon Density Dataset (Xu et al., 2019). This dataset consolidates field measurement data

from 2004 to 2014 reported in publicly available literature. From this dataset, 1,235 sample points for forest soil carbon

density and 614 sample points for grassland soil carbon density were extracted. (2) The Second National Soil Survey of

China (1979–1985). This survey resulted in the publication of the Soil Chronicles Atlas of China, Volumes 1–6, which record

soil property data from the 1980s. From this, 339 sample points for forest soil properties and 147 sample points for grassland

soil properties were extracted. (3) The Chinese Soil Series (since 2008). This investigation produced the Soil Series Atlas of

China, which consists of 30 volumes (Appendix Table B1). From this dataset, 724 and 529 sample points for forest and

grassland soil properties were extracted, respectively. The spatial distribution of the sample points is presented in Appendix

Fig. B1.

The results of the two large-scale soil surveys were documented in books that recorded soil properties during different

periods in China. This study extracted information from these surveys, including the geographic location (latitude and

longitude), soil depth (0–100 cm), soil type, organic carbon content, soil bulk density, and >2 mm gravel content, and applied

Eq. (1) to calculate the soil carbon density. The formula used to calculate soil carbon density based on soil properties is as

follows:

$$C_S = \sum_{i=1}^{n} SOC_i \times D_i \times BD_i \times (1\text{-}SC_i) \times 0.1 \tag{1}$$

where $C_S$ is the soil organic carbon density, $SOC_i$ is the organic carbon percentage in the $i$-th soil layer (%), $D_i$ is the

thickness of the $i$-th soil layer (cm), $BD_i$ is the bulk density of the $i$-th soil layer (g/cm³), $SC_i$ is the percentage of gravel

(>2mm) in the $i$-th soil layer (%), and $n$ is the number of layers in the 100 cm soil profile. This study only selected sample

points with a soil profile thickness of ≥100 cm and considered only the carbon density within the top 100 cm. For sample

points lacking bulk density data, bulk density was estimated using an empirical transfer function from Yang et al. (2007).

This function is based on the negative correlation between bulk density (BD, g/cm³ ) and soil organic matter (SOM, %) , with

the specific formula:

$$BD = 0.29 + 1.2033 \times e^{-0.0775 \times SOM} \tag{2}$$

Vegetation biomass carbon density data were sourced from the 2010s China Land Ecosystem Carbon Density Dataset (Xu

et al., 2019), including carbon density data from both aboveground (forests: 1,610 points, grasslands: 2,224 points) and

belowground layers (forests: 1,544 points, grasslands: 1,458 points) of forest and grassland ecosystems. The formula for calculating vegetation carbon density is as follows:

$$C_v = C_{above\_ground} + C_{below\_ground} \tag{3}$$

where $C_v$ is the vegetation biomass carbon density, $C_{above\_ground}$ is the aboveground vegetation carbon density, and $C_{below\_ground}$ is the belowground vegetation carbon density.

  The collected vegetation biomass and soil carbon density data were grouped according to the 25 merged province-level administrative divisions described above based on the geographic coordinates of the data points. Overall, for each province, the sample points exhibited a normal distribution (Appendix Figs. B2–B4). The arithmetic mean was used to calculate the

240 provincial-level average carbon density. For provinces with exceptionally high or low values, the median was used to reflect the average carbon density and minimize the influence of outliers. The provincial-level vegetation and soil carbon density data are listed in Table 2.

**Table 2.** Provincial vegetation and soil carbon density data

| Code | Province/region | Forest (Mg/ha) | | Grassland (Mg/ha) | |
| --- | --- | --- | --- | --- | --- |
| | | SOCD | VCD | SOCD | VCD |
| No.1 | Jing-Jin-Ji | 75.39 (n=104) | 43.83 (n=117) | 88.32 (n=53) | 7.61 (n=19) |
| No.2 | Shanxi | 59.98 (n=65) | 40.63 (n=66) | 56.13 (n=115) | 8.77 (n=71) |
| No.3 | Shandong | 60.42 (n=30) | 42.29 (n=26) | / | / |
| No.4 | Henan | 59.03 (n=17) | 42.41 (n=24) | / | / |
| No.5 | Shaanxi | 74.29 (n=174) | 29.78 (n=101) | 64.75 (n=110) | 4.03 (n=45) |
| No.6 | Anhui | 86.90 (n=44) | 63.06 (n=57) | / | / |
| No.7 | Hu-Ning | 91.79 (n=31) | 37.63 (n=27) | / | / |
| No.8 | Hunan | 92.60 (n=174) | 51.94 (n=42) | / | / |
| No.9 | Hubei | 139.57 (n=63) | 48.00 (n=20) | / | / |
| No.10 | Jiangxi | 93.29 (n=162) | 50.81 (n=44) | / | / |
| No.11 | Zhejiang | 115.13 (n=69) | 54.14 (n=35) | / | / |
| No.12 | Fujian | 117.71 (n=114) | 58.80 (n=72) | / | / |
| No.13 | Yue-Qiong | 111.36 (n=233) | 37.33 (n=92) | / | / |
| No.14 | Guangxi | 108.26 (n=156) | 55.87 (n=105) | 99.32 (n=17) | / |
| No.15 | Yunnan | 105.84 (n=110) | 76.26 (n=67) | 100.52 (n=14) | / |
| No.16 | Guizhou | 129.37 (n=64) | 50.31 (n=29) | 284.18 (n=35) | / |
| No.17 | Chuan-Yu | 98.83 (n=132) | 55.96 (n=159) | 143.09 (n=50) | 1.25 (n=142) |
| No.18 | Inner Mongolia | 69.38 (n=179) | 41.60 (n=263) | 88.79 (n=119) | 5.77 (n=416) |
| No.19 | Liaoning | 91.13 (n=70) | 44.74 (n=43) | 77.71 (n=35) | 3.32 (n=25) |
| No.20 | Jilin | 95.09 (n=57) | 73.85 (n=39) | 67.09 (n=30) | 3.06 (n=24) |
| No.21 | Heilongjiang | 145.45 (n=91) | 64.63 (n=114) | 93.58 (n=28) | 2.98 (n=22) |
| No.22 | Gan-Ning | 99.44 (n=88) | 36.80 (n=57) | 54.66 (n=236) | 3.80 (n=159) |
| No.23 | Qinghai | 75.87 (n=20) | 30.54 (n=36) | 108.60 (n=249) | 6.45 (n=385) |
| No.25 | Xinjiang | 64.32 (n=22) | 25.59 (n=42) | 93.97 (n=119) | 4.09 (n=91) |
| No.24 | Xizang | 129.33 (n=35) | 82.43 (n=20) | 58.89 (n=167) | 4.20 (n=291) |

SOCD refers to soil organic carbon density; VCD refers to vegetation carbon density.

## 2.3 Methods

Annual emissions of carbon from land-use change were calculated with a bookkeeping model based on two types of data: rates of land-use change and per hectare effects of land-use change on carbon stocks (Fig. 2). The former was calculated by constructing land-use transition rules, while the latter was derived from the disturbance response curves in the bookkeeping model, combined with provincial vegetation and soil carbon density datasets. Due to data limitations, this accounting does not consider carbon emissions from wood harvesting.

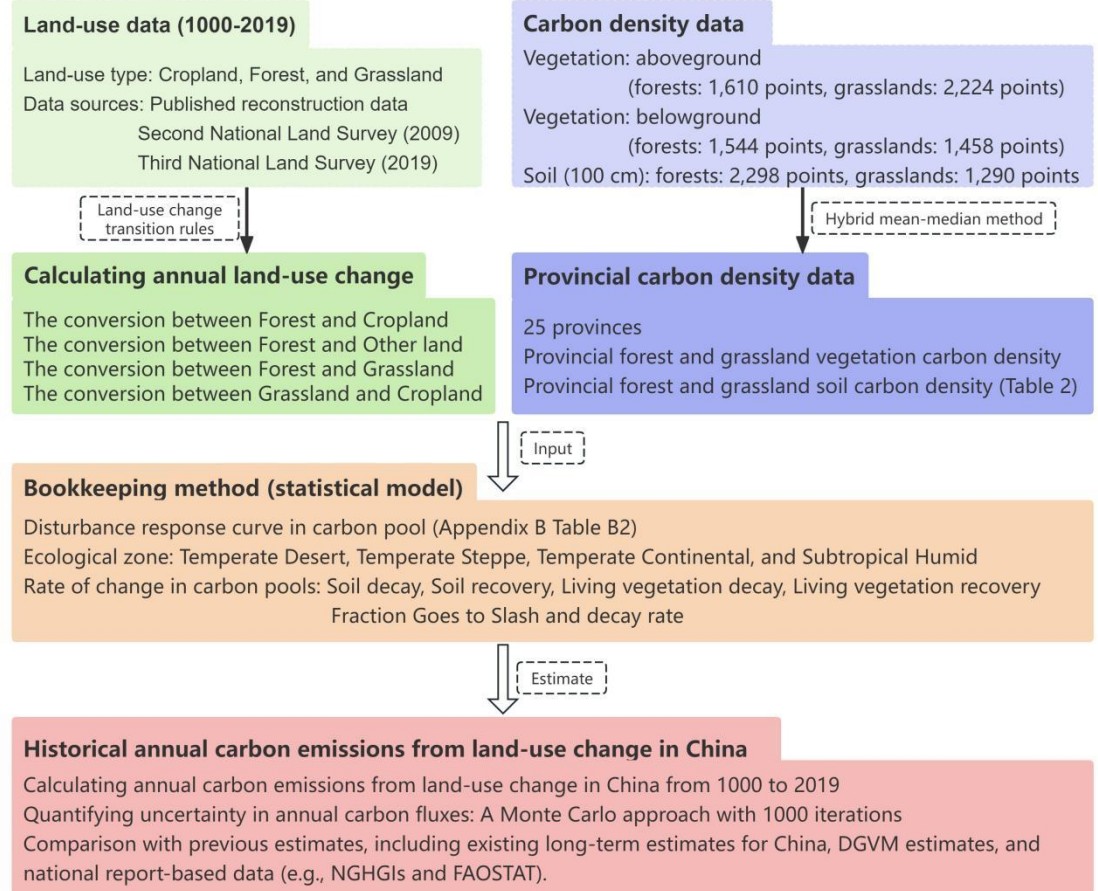

**Figure 2.** Framework for calculating annual carbon emissions based on the bookkeeping model. The color scheme delineates the framework's primary modules: data inputs for land use (green) and carbon density (blue), the core bookkeeping model (orange), and the final results and analysis of carbon emissions (red).

### 2.3.1 Bookkeeping method

The bookkeeping method (a statistical model) proposed by Houghton and Castanho (2023) was employed to estimate the annual carbon emissions caused by land-use changes in China from 1000 to 2019. Due to data limitations, long-term historical land-use reconstructions in China are primarily constrained to land-use "states" (e.g., total cropland or forest area at national/provincial levels for specific years) rather than spatially explicit land-use transitions. This characteristic,

combined with the provincial-level spatial resolution of our data, makes such reconstructions inherently compatible with the bookkeeping model adopted here (Houghton and Castanho, 2023). Bookkeeping is widely used to estimate carbon emissions across multiple spatial and temporal scales and characterizes the impacts of human-induced land-use changes on carbon stocks in vegetation and soil across various terrestrial ecosystems (Qin et al., 2024; Yang et al., 2023; Bastos et al., 2021; Hartung et al., 2021). The bookkeeping model used in this study is primarily driven by land-use change data and utilizes observed vegetation and soil carbon density data and specific disturbance response curves for each land-use transition type. As this method excludes the influence of unchanged land-use types and environmental changes, such as carbon dioxide concentrations and climate change, it quantifies direct anthropogenic fluxes and ignores carbon fluxes driven by environmental changes (Dorgeist et al., 2024; Houghton and Castanho, 2023). Consequently, the results of this method are frequently incorporated into global carbon budget estimates (Friedlingstein et al., 2023).

Our bookkeeping model uses statistical data rather than spatial grid data as input and calculates the net carbon change in terrestrial ecosystems due to land-use changes on an annual basis. The disturbance response curves specify the dynamic changes in carbon pools following land-use transition, including biomass (both aboveground and belowground), litter (branches, trunks, roots, etc.), and soil organic carbon pools over time for each land-use type and per hectare of land-use change until a new carbon density equilibrium is reached (Houghton and Castanho, 2023). The response time for carbon release or absorption due to land-use changes can range from decades to centuries. The values of the disturbance response curves ($f$) were derived from Houghton and Castanho (2023) (see Appendix B Table B2). Therefore, the carbon emission flux estimated at any given time includes both instantaneous and legacy fluxes from previous land-use changes. The calculation formula is as follows:

$$\Delta C_{flux}(j, t)= \sum_{k} \left( R_{LU}(j, k, t) \times C_v(j) \times f_{vge} \right) + \left( R_{LU}(j, k, t) \times C_s(j) \times f_{soil} \right) + \left( R_{LU}(j, k, t) \times C_v(j) \times f_{slash} \right) \qquad (4)$$

where $\Delta C_{flux}(j, t)$ is the carbon emission flux due to land-use change in province $j$ at time $t$, $R_{LU}(j, k, t)$ is the land-use transition amount for type $k$ in province $j$ at time $t$, $C_V(j)$ and $C_s(j)$ are the vegetation and soil carbon densities in province $j$, respectively, and $f$ is the disturbance response curve for vegetation and soil carbon pools.

**2.3.2 Calculating annual land-use change**

This study established and utilized a land-use classification system comprising four categories: cropland, forest, grassland, and other land. The long-term changes in the areas of these four types are shown in Fig. 3. Within this system, the data for the first three land use categories (cropland, forest, and grassland) were derived from high-confidence, long-term reconstructed datasets compiled for China (Table 1). As the sum of these three land use categories in the reconstructed datasets do not cover the entire land area of China, the residual area was introduced to construct a comprehensive and closed

classification system suitable for carbon budget accounting. The establishment of the 'other land' category is consistent with the general practices in other land use-related studies, such as the classification used for carbon emission estimation by Houghton and Castanho (2023) and the major land-use types defined by the FAO (2021). Operationally, the other land category is defined as the residual area within a province's administrative boundaries after subtracting the areas of cropland, forest, and grassland. It should be noted that, unlike contemporary, granular classification standards, our other land category is a composite concept that includes a variety of both human-affected and unaffected land types.

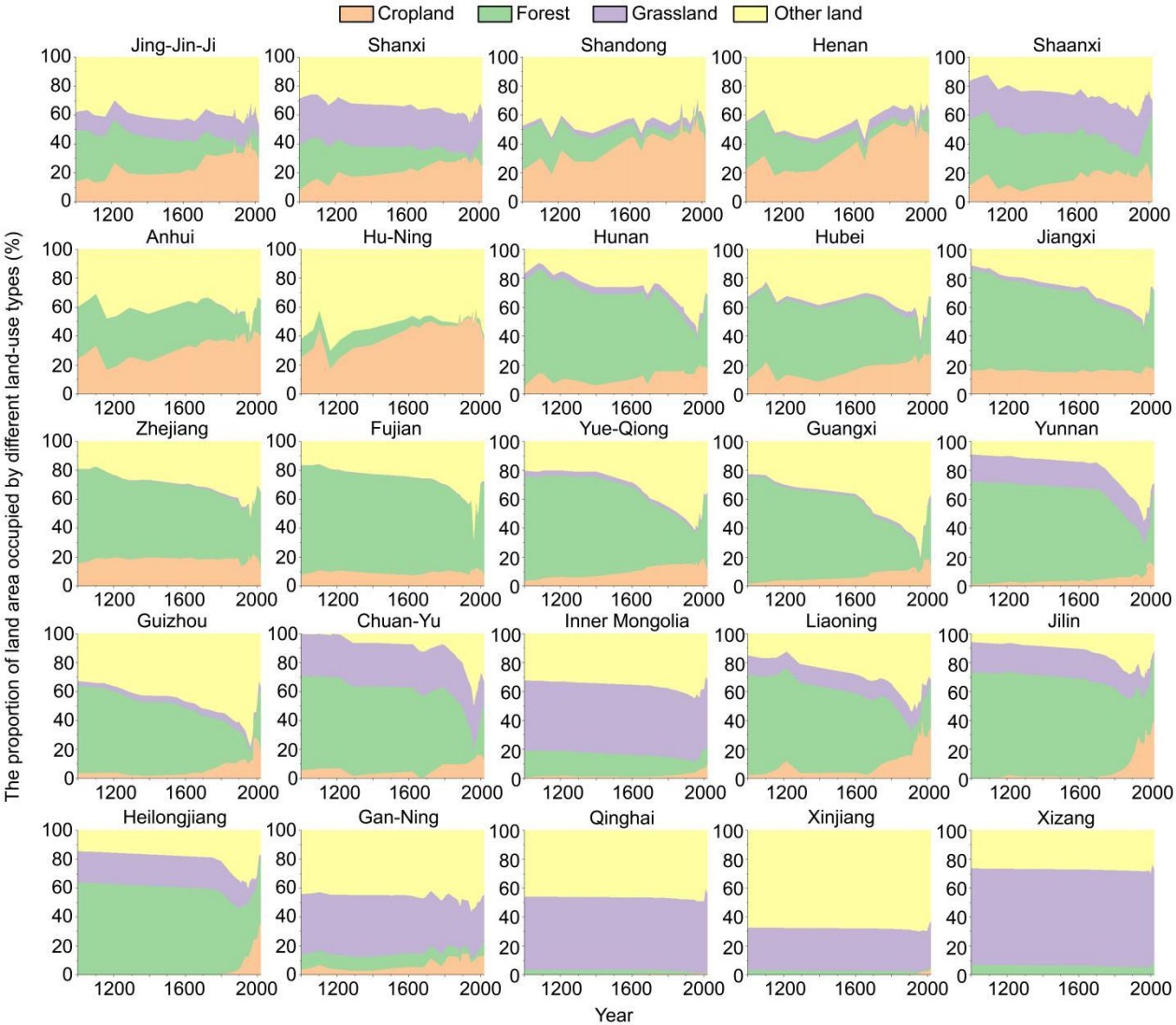

**Figure 3.** Percentage of the area of cropland, forest, grassland, and other land at the provincial scale. Jing-Jin-Ji represents the aggregation of Beijing, Tianjin, and Hebei; Hu-Ning represents Shanghai and Jiangsu; Yue-Qiong represents Guangdong and Hainan; Chuan-Yu represents Sichuan and Chongqing; and Gan-Ning represents Gansu and Ningxia.

Land-use products derived from remote sensing imagery are spatially explicit, thereby enabling the clear identification of land-use type transitions. However, the provincial-level reconstructed data used in this study lacked explicit spatial location information, and the conversion relationships between different land-use types were not always clear. For example, when only two land-use types are involved and the increase (or decrease) in one land-use type exactly matches the decrease (or

increase) in the other type, the conversion between land-use types is relatively straightforward. However, when more than two land-use types are involved in land-use change, the conversion relationships become complex. To address this latter issue, we established different rules to derive land-use conversions (Fig. 4).

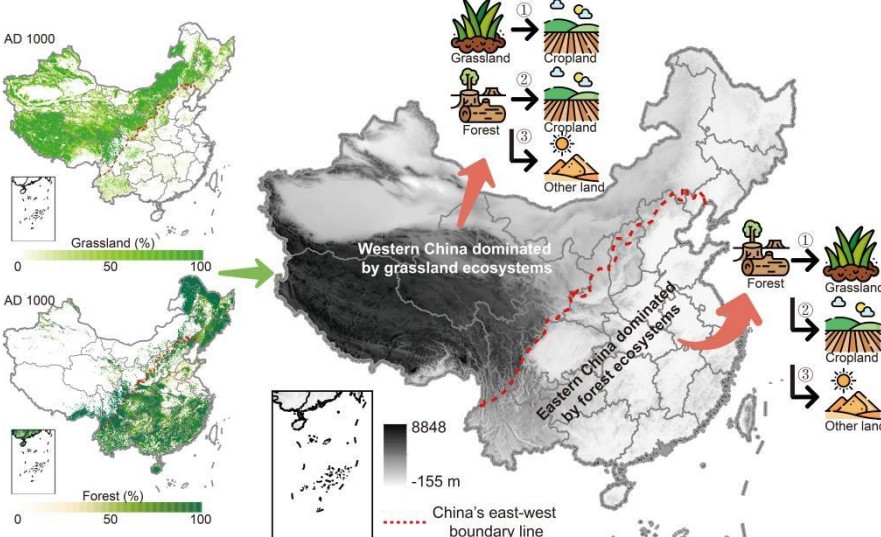

**Figure 4.** Differentiated historical land-use transition rules for Eastern and Western China. These rules define different regional conversion priorities: the primary pathway in the East is from forest to grassland (①) and to cropland (②); while in the West, the pathway is from grassland (①) and forest (②) to cropland. Numbers ①, ②, and ③ represent the priority levels.

The primary prerequisite for establishing the land-use transition rules in this study is to maintain the internal consistency with the reconstruction methodology of the historical land-use dataset we used (He et al., 2025). We followed the regionally differentiated assumptions from that dataset's reconstruction: in western China, changes in grassland are primarily attributed to the encroachment of cropland, leading us to prioritize the conversion from grassland to cropland; in eastern China,

grassland is considered secondary vegetation formed through succession after deforestation, so we prioritized the conversion from forest to grassland. After completing these grassland-related conversions, the conversion from forest to cropland was then determined, and the remaining decrease in forest area was allocated to other land (Fig. 4). We define this rule as the grassland-priority method. This rule has, to the best of our knowledge, the highest accordance with the reality in China's land use history. The carbon budget results presented later in this study are calculated based on this method.

Although the grassland-priority allocation rule, based on historical reconstruction, has an empirical basis, we acknowledge that it does not preclude other possible land use conversions rules, which will help us to explore the uncertainty in the estimated carbon fluxes caused by the assumptions in the land use conversion rule. To this end, we developed two alternative allocation rules. The first one is the area-weighted allocation method. This rule assumes that outgoing land conversions are allocated to different incoming land types in proportion to their area's share of the total outgoing area. It is a mathematical allocation method driven by data stocks and has no structural preferences. The second one is the forest-priority method. This

rule is a direct counter-hypothesis to the grassland-priority method. Its allocation logic is: in western China, it prioritizes the conversion from forest to cropland, followed by grassland to cropland; in eastern China, it prioritizes the conversion from forest to cropland, followed by conversions to grassland and other land uses. Although this rule has no strong basis in historical data, its purpose is to assess the potential impacts of different land conversion assumptions on the estimated carbon

fluxes.

Here we provide an example to demonstrate the three allocation rules as described above. Let's assume that for a given year and a province in western China, forest decreased by 60 ha, grassland by 90 ha, whereas cropland increased by 105 ha. (1) Under the grassland-priority method, this results in 90 ha of grassland and 15 ha of forest being converted to cropland, with the remaining 45 ha of forest loss allocated to other land. (2) The area-weighted method allocates 63 ha from grassland

and 42 ha from forest to cropland, while distributing the remaining losses (27 ha of grassland and 18 ha of forest) to other land. (3) Finally, the forest-priority method allocates the full 60 ha of forest loss to cropland, with the remaining 45 ha of needed cropland coming from grassland, and the surplus 45 ha of grassland loss is converted to other land.

Houghton and Castanho (2023) proposed four alternative explanations for forest conversion to other land: Explanation 1: Forest loss is overestimated; Explanation 2: Forests are converted to shifting cultivation; Explanation 3: Forests are

converted to new cropland, while an equal area of cropland is abandoned and undergoes degradation; and Explanation 4: Forests are converted to new cropland, while an equal area of cropland is abandoned, and subsequently restored to forest over a long period. Historically, shifting cultivation (through deforestation) was common. Shifting cultivation is a primitive and underdeveloped agricultural practice in which farmers clear land by burning and cultivating it extensively to obtain agricultural products. Once the soil fertility is exhausted, farmers abandon cultivation and continue to clear new land. This

practice has been widespread historically and continues today in the tropical rainforest regions of South America, Africa, and Southeast Asia (Heinimann et al., 2017). Based on the characteristics of forest cover change documented in the Chinese historical literature, attributing forest loss to shifting cultivation aligns more closely with historical facts, excluding conversion to cropland and grassland. This form of agriculture has been recorded extensively in Chinese historical documents.

The annual changes in cropland, forest, and grassland areas over the past millennium (Fig. 5a) clearly revealed that between the 18[th] and mid-20[th] centuries, the annual loss of forest area greatly exceeded the annual increase in cropland area. Based on the conversion rules assumed here, we derived the annual change in other land (primarily shifting cultivation) over the past millennium (Fig. 5b). The data revealed that shifting cultivation was prevalent throughout history, although its scale was relatively small before the 18[th] century, with an average annual increase of $6.22 \times 10^4$ ha. However, after explosive

population growth occurred in China, people under the pressure of survival expanded to hilly and mountainous forestlands, and converted large areas of forest via shifting cultivation. The average annual increase in shifting cultivation during this period reached $40.54 \times 10^4$ ha, which was 6.5 times greater than that of the previous period.

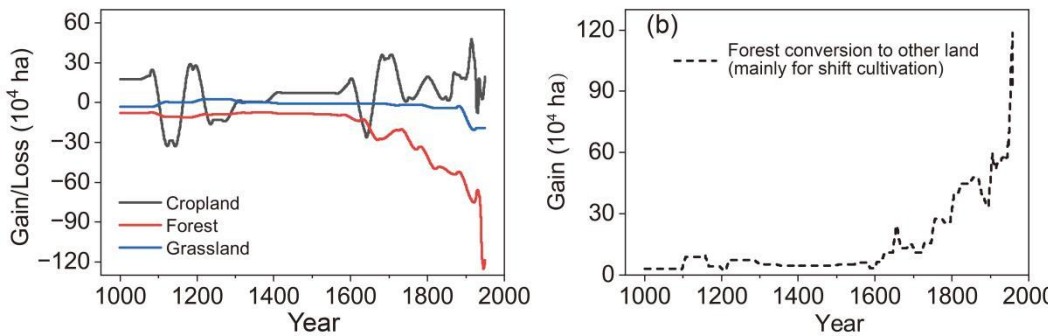

**Figure 5.** Changes in cropland, forest, grassland, and other land areas. (a) Annual net change in cropland, forest, and grassland area, with positive and negative values indicating net gains and losses, respectively. (b) Illustrates a key process driving forest loss: shift cultivation.

### 2.3.3 Uncertainty assessment

The uncertainty in this study arises from three sources: first, the uncertainty in the reconstructed land-use data used as model input; second, the uncertainty introduced by the land-use transition rules; and third, the uncertainty in the carbon density parameters applied in carbon flux accounting.

To comprehensively account for the three aforementioned sources of uncertainty, we designed a systematic simulation scheme. First, we employed three different land-use change allocation rules (the grassland-priority method, the area-weighted method, and the forest-priority method) and calculated their respective annual land conversion areas. For each allocation rule, we performed a Monte Carlo simulation consisting of 1000 iterations, in order to quantify the uncertainty arising from the sources one and three (input data and key parameters). More specifically, we calculated the mean and standard deviation for each carbon pool (aboveground, belowground, soil) in forest and grassland derived from existing sample data. In each iteration, carbon density values were randomly sampled from normal distributions parameterized with these statistics. For the uncertainty in the annual land-use change area for a given allocation rule, the value calculated by following the given rule served as the mean for its sampling distribution, with the standard deviation being set to 10% of this mean, again forming a normal distribution. This simulation design ultimately yielded a total of 3000 simulation results (3 rules × 1000 iterations for each rule). To establish the final uncertainty in the estimated carbon flux, we aggregated the annual carbon emissions from all 3000 simulations for each year and selected their maximum and minimum values. The resulting interval thereby reflects the combined uncertainty originating from input data, key parameters, and the allocation rule.

# 3. Results

## 3.1 Overall carbon emissions

The land-use changes and associated carbon emissions in China over the past millennium are illustrated in Fig. 6. From 1000 to 2019, cumulative carbon emissions resulting from land-use changes totaled 19.61 Pg C, with the highest cumulative emissions of 21.87 Pg C occurring around 1980. Overall, due to lag effects, the carbon emission trajectory did not fully align with the timeline of land-use changes. Specifically, the reversal of forest area decline (i.e., the transition from forest loss to forest regrowth) occurred in the 1960s (Fig. 6a and 6b), whereas the reversal of the carbon budget from carbon source to carbon sink occurred in the 1980s. Approximately 30% of the annual carbon emission flux was attributable to residual emissions from historical periods.

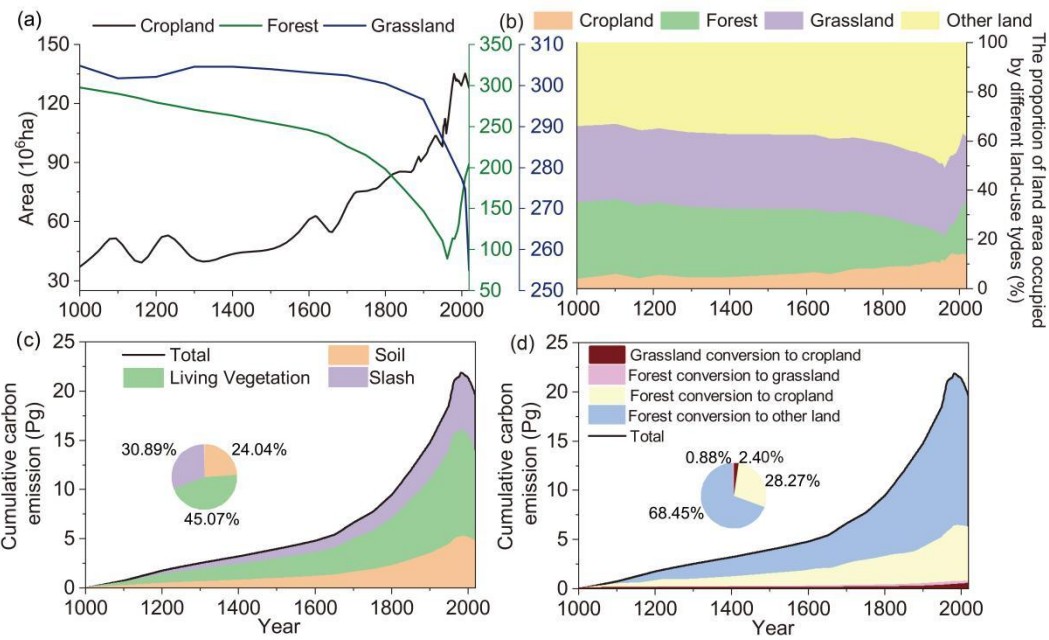

**Figure 6.** Annual land-use changes and carbon emissions in China from 1000 to 2019. (a) Cropland, forest, and grassland areas (absolute values), in units of $10^6$ hectares. (b) Proportions of four land-use types in each period, with all remaining terrestrial cover—excluding the reconstructed cropland, forest, and grassland—classified as other land. (c) Cumulative carbon emissions from land-use changes across different carbon pools. (d) Cumulative carbon emissions from different land-use transitions. In (c) and (d), the two pie charts represent the shares of different carbon pools and land-use transitions in the cumulative carbon emissions over the millennium, respectively.

Based on the clear temporal trajectories, four distinct phases of carbon emissions were identified. Phase 1 (1000–1700): A slow growth phase for carbon sources, driven by deforestation, cropland expansion, and grassland reclamation, which resulted in a cumulative carbon emission of 6.60 Pg C, accounting for 30.17% of the total carbon emissions. The average annual carbon emission in this phase was 9.46 Tg C yr⁻¹ (Fig 7a). Phase 2 (1700–1980): A rapid growth phase for carbon sources during which croplands expanded significantly beyond traditional agricultural areas in China, moving to Southwest, Northeast, and Northwest China, accompanied by large-scale deforestation and grassland reclamation. Cumulative carbon emissions during this period reached 15.27 Pg C, accounting for 69.86% of the total emissions. The average annual emission

was 54.09 Tg C yr$^{-1}$, 5.7 times that of Phase 1. Phase 3 (1980–1998) was a phase dominated by large-scale afforestation, the carbon budget for land-use changes shifted from being a carbon source to a carbon sink. Between 1980 and 1998, the carbon sink amounted to 0.12 Pg C, with an average annual carbon sink of 16.85 Tg C yr$^{-1}$. Phase 4 (1998–2019): An enhanced carbon sink phase attributed to the widespread implementation of large-scale forestry projects. During this period, the total carbon sink reached 1.85 Pg C (Fig. 6c and 6d), with an average annual carbon sink intensity of 88.21 Tg C yr$^{-1}$ (Fig 7a), which was 5.2 times higher than that of Phase 3.

Regarding carbon pool types, the vegetation carbon pool stood out as the largest contributor to total emissions, accounting for 45.07% of the overall emissions (Fig. 6c). This was reflected in an average annual emission intensity of 8.67 Tg C yr$^{-1}$ (Fig. 7b). Following closely was the slash carbon pool, which contributed 30.89%, with an average annual emission intensity of 5.95 Tg C yr$^{-1}$ (Fig. 7c). The soil carbon pool, while still significant, represented a smaller portion at 24.04%, emitting an average of 4.63 Tg C yr$^{-1}$ (Fig. 7d). When considering the impact of land-use changes, the conversion of forest to other land types, particularly shifting cultivation, emerged as the most dominant factor. This conversion alone was responsible for a staggering 68.45% of the total carbon emissions (Fig. 6d), with an average annual emission of 13.17 Tg C yr$^{-1}$ (Fig. 7e). The conversion of forest to cropland followed, contributing 28.27% of the emissions, or 5.44 Tg C yr$^{-1}$ (Fig. 7f). In comparison, the conversion of grassland to cropland had a relatively minor effect, accounting for just 2.40% of the emissions, equivalent to 0.46 Tg C yr$^{-1}$ (Fig. 7g). Finally, the conversion of forest to secondary grassland had an almost negligible impact, representing only 0.88% of total emissions, with an annual release of less than 0.01 Tg C yr$^{-1}$.

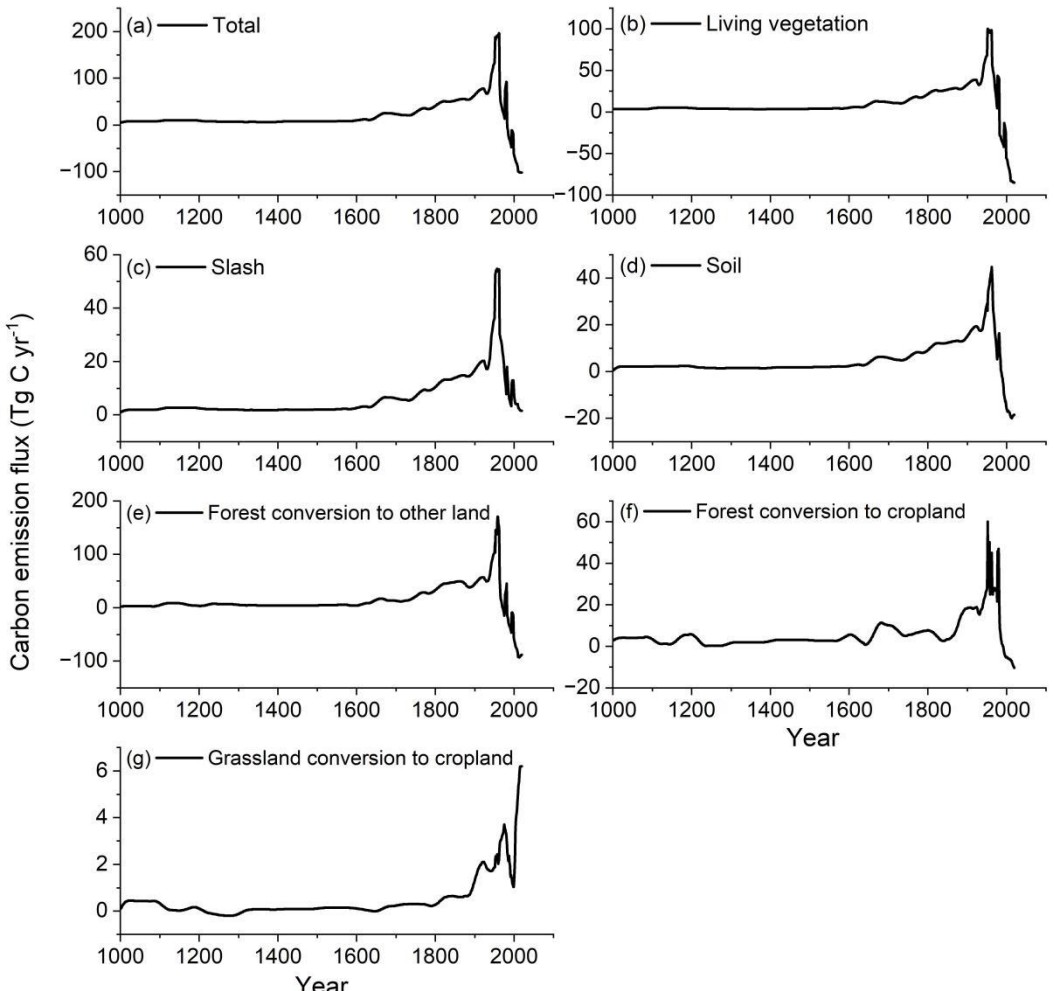

**Figure 7.** Annual carbon emission flux of land-use changes in China from 1000 to 2019: total, carbon pools of soil, vegetation, and slash and different land-use conversions. Annual carbon emission flux from forest conversion to grassland is less than 0.01 Tg C yr[-1] and thus is not presented in graphical form. Negative values indicate carbon sink fluxes.

## 3.2 Regional carbon emissions

To facilitate the analysis of the spatiotemporal evolution of land-use carbon emissions, this study divided China into five major regions: North China, Southeast China, Southwest China, Northeast China, and Northwest China (Fig. 8). North China primarily refers to the North China Plain (Beijing, Tianjin, Hebei, Henan, Shandong, Anhui, and Jiangsu) and the provinces Shanxi and Shaanxi. Southeast China includes the provinces Hubei, Hunan, Jiangxi, Zhejiang, Fujian, Guangdong, Hainan, and Guangxi. Southwest China consists of the provinces Sichuan, Chongqing, Guizhou, and Yunnan. Considering the forest coverage in southeastern Xizang, this region is also categorized as part of Southwest China. Northeast China includes the provinces Liaoning, Jilin, and Heilongjiang as well as Inner Mongolia, where forest resources are mainly distributed in the Greater Khingan Range. Northwest China includes the provinces Gansu, Ningxia, Qinghai, and Xinjiang.

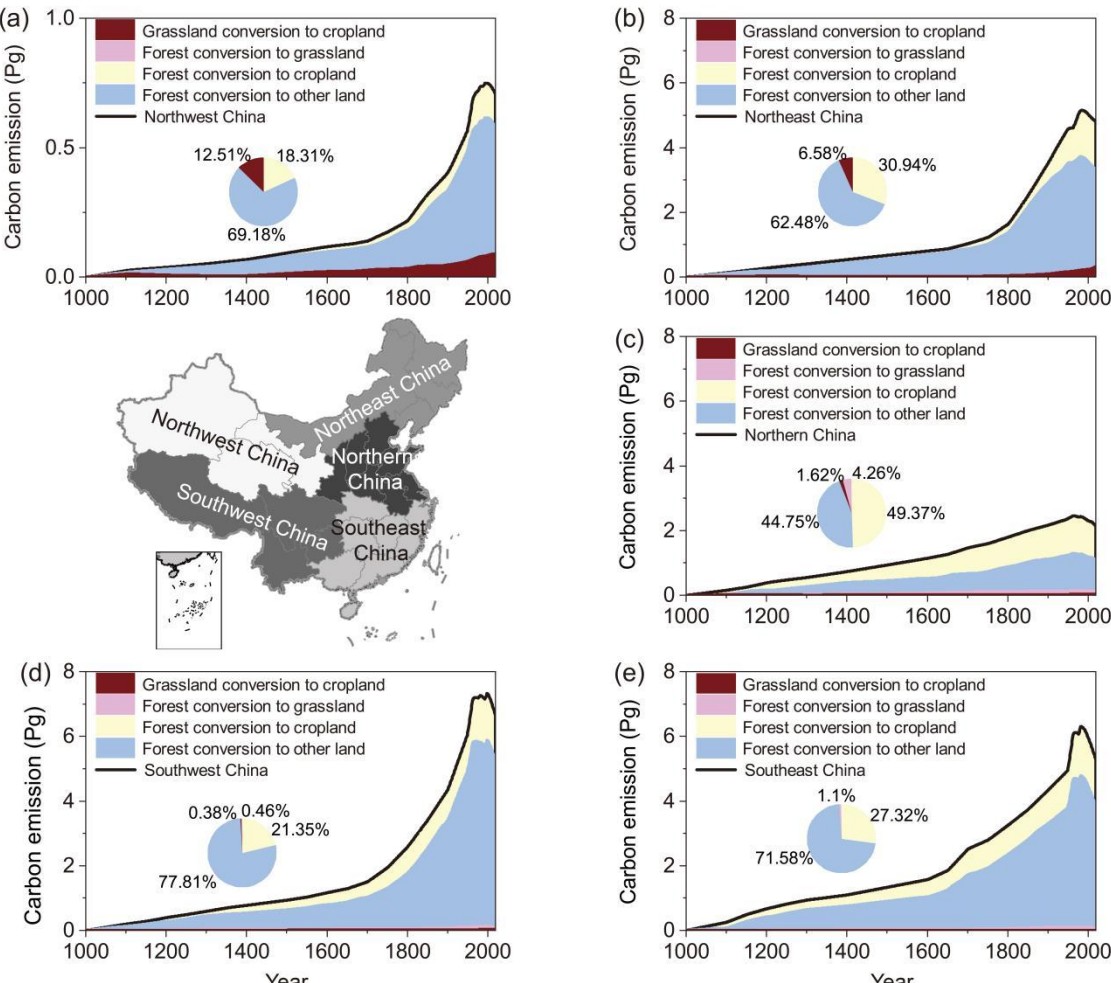

**Figure 8.** Cumulative carbon emissions and their proportions of land-use changes in different regions of China from 1000 to 2019. Within each subplot, the stacked area chart depicts the historical trajectory of cumulative carbon emissions (Pg C), while the inset pie chart quantifies the final proportional contribution from each land-use conversion type.

Over the past millennium, Southwest China has recorded the highest cumulative carbon emissions, totaling 6.66 Pg C, followed by Southeast China with 5.29 Pg C and Northeast China with 4.81 Pg C. In contrast, North China accounted for 2.15 Pg C, and Northwest China recorded only 0.71 Pg C. Carbon emissions from land-use conversion in these regions showed significant variation. Notably, North China was distinct from the other regions, with the highest proportion of carbon emissions resulting from the conversion of forests to croplands, accounting for 49.37% of the total (Fig. 8c). In the other four regions, the conversion of forests to other land types (mainly shifting cultivation) contributed to the highest proportion of carbon emissions, with values of 71.58%, 77.81%, 62.48%, and 69.18%. The conversion of forests to secondary grasslands occurred mainly in North China and Southeast China, contributing 4.26% and 1.1% of total carbon emissions, respectively (Fig. 8c and 8e). The conversion of grasslands to cropland occurred mainly in Northwest and Northeast China, accounting for 12.51% and 6.58%, respectively (Fig. 8a and 8b).

At the provincial scale, the years in which land-use changes shifted from carbon sources to carbon sinks varied across

regions (Appendix Fig. C1). During the carbon source period, seven provinces had cumulative carbon emissions exceeding 1.00 Pg C or an average carbon emission flux greater than 1.00 Tg C yr⁻¹, namely, Chuan-Yu, Yunnan, Heilongjiang, Guangxi, Inner Mongolia, Jilin, and Hunan. Among these, Chuan-Yu had the highest carbon emissions, reaching 3.28 Pg C (with an average carbon emission flux of 3.35 Tg C yr⁻¹) (Fig. 9). The cumulative carbon emissions in eight provinces, namely, Guizhou, Yue-Qiong, Liaoning, Hubei, Fujian, Jiangxi, Xizang, and Jing-Jin-Ji, ranged between 0.55 and 0.97 Pg C (average carbon emission flux of 0.58–0.99 Tg C yr⁻¹). The remaining 10 provinces had cumulative carbon emissions of less than 0.50 Pg C, with Hu-Ning having the lowest at 0.11 Pg C (0.11 Tg C yr⁻¹). During the carbon sink period, the contribution of carbon uptake by each province followed a similar order as the carbon emissions in each province during the carbon source period. The Chuan-Yu region contributed 0.39 Pg C to the carbon sink, with a flux of 10.35 Tg C yr⁻¹. This indicates that provinces with significant carbon emissions owing to widespread deforestation and agricultural expansion in historical periods have played an important role as carbon sinks in recent decades, largely through large-scale afforestation and other interventional measures.

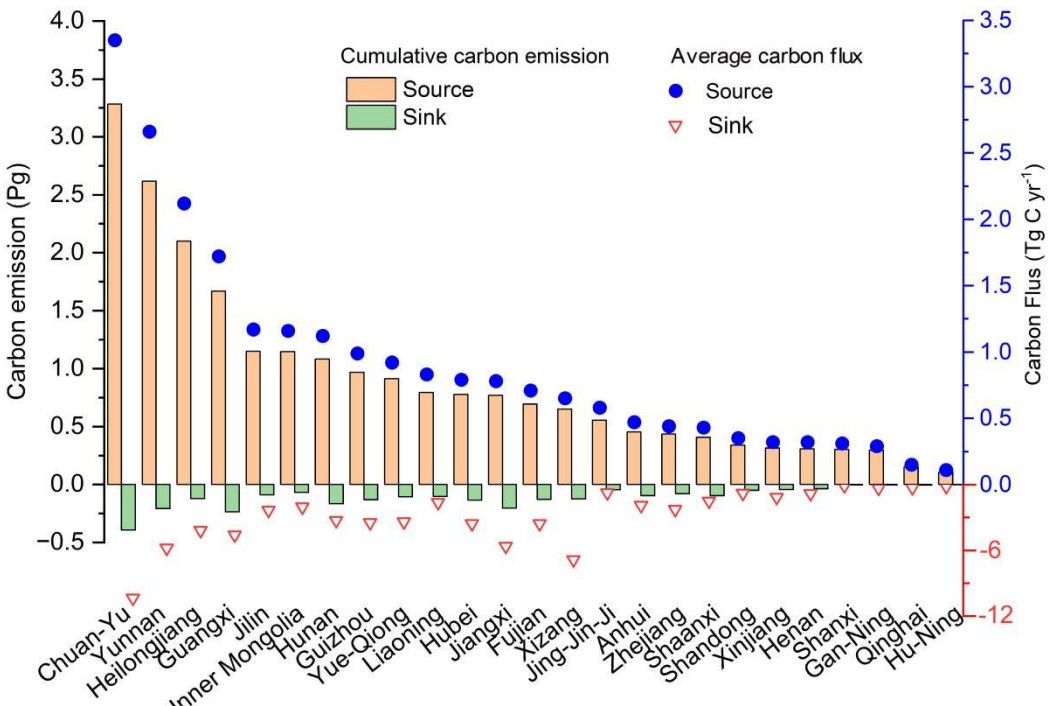

**Figure 9.** Cumulative carbon emissions and average carbon flux at the provincial scale from 1000 to 2019. Jing-Jin-Ji represents the aggregation of Beijing, Tianjin, and Hebei; Hu-Ning represents Shanghai and Jiangsu; Yue-Qiong represents Guangdong and Hainan; Chuan-Yu represents Sichuan and Chongqing; and Gan-Ning represents Gansu and Ningxia.

### 3.3 Uncertainty and Sensitivity Analysis

The results of the comprehensive uncertainty assessment are presented in Fig. 10 (also see Section 2.3.3). This figure displays the millennial-scale time series of carbon emission fluxes estimated following the grassland-priority allocation rule

in land use conversion (also see Section 2.3.2). The uncertainty interval was constructed to systematically integrate

uncertainties arising from three sources: input data, key parameters, and allocation rules in land use conversion. Its upper and

lower bounds were derived by aggregating all results from 3,000 Monte Carlo simulations (1000 simulations for each of the

three allocation rules—the grassland-priority method, the area-weighted method, and the forest-priority method).

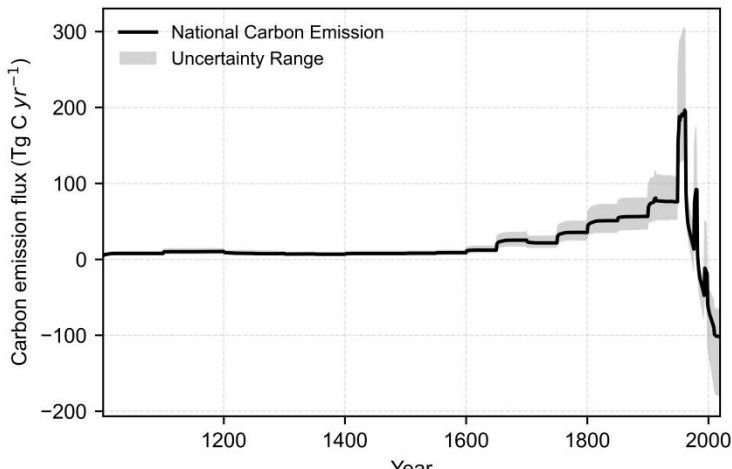

**Figure 10.** Carbon emission fluxes from land-use change in China with their uncertainties. The black line represents the

mean result of 1000 Monte Carlo simulations using the grassland-priority allocation rule in land use conversion. The gray

shaded area represents the uncertainty interval defined by the maximum and minimum values across all 3000 simulation runs

(1000 iterations for each of the three allocation rules: grassland-priority, area-weighted, and forest-priority). This interval

incorporates the combined uncertainties from input data, carbon density parameters, and the allocation rule. For details, refer

to the sections 2.3.2 and 2.3.3.

To examine the influence of different allocation rules on the derived land-use change areas, we compared the annual

land-use transition matrices generated by the grassland-priority method and the area-weighted method. The results indicate

that although certain transition types exhibit numerical differences in area, the primary transition processes, such as the

conversion of forest to cropland, show a high degree of consistency (see Appendix Fig. C2). This suggests that the core

land-use change patterns derived from the reconstructed historical land use datasets are robust. Nevertheless, the choice of

allocation rule remains a significant source of overall uncertainty. As illustrated in Fig. 10, the comprehensive uncertainty

interval proposed in this study systematically integrates three different allocation rules (grassland-priority, area-weighted,

and forest-priority), thereby delineating the potential range of impacts that the choice of allocation rule may have on carbon

flux estimations.

A period-based analysis of the estimation uncertainty reveals its clear temporal evolution (Appendix Table C1). During the

period when land-use change acted as a carbon source, the uncertainty expanded in tandem with emission intensity. As mean

annual emissions rose from 75.51 Tg C during 1900-1949 to a peak of 102.95 Tg C during 1950-1982, the corresponding

mean upward uncertainty increased from 37.31 Tg C to 74.35 Tg C, reflecting the greater estimation challenges posed by the intense land-use activities of that era. A key transition occurred after 1982, when the land-use sector transitioned to a net carbon sink with a mean annual uptake of 60.90 Tg C. Notably, the uncertainty in this sink estimation exhibits a distinct asymmetry: the mean downward uncertainty (58.91 Tg C) is significantly larger than the mean upward uncertainty (34.56 Tg C), suggesting that the actual sink strength during the modern observational period is likely stronger than our estimate.

In summary, the comprehensive assessment framework proposed in this study, which accounts for the uncertainties arising from multiple allocation rules and stochastic parameter simulations, provides a robust quantification of uncertainty in the derived land-use carbon fluxes. This analysis reaffirms the trends and key turning points in China's millennial-scale carbon budget. To further narrow the associated uncertainty, future research should focus on improving the measurement accuracy and spatial representativeness of carbon density parameters and on enhancing the reliability of land-use reconstruction.

## 4. Discussion

### 4.1 Comparison with other land-use reconstructions

Existing global long-term land-use datasets often exhibit significant discrepancies with historical facts regarding China, both in quantitative estimates and spatial patterns (Fang et al., 2020; Zhao et al., 2022), making it difficult to accurately capture the complex historical reclamation processes. In contrast, domestic studies on historical land-use reconstruction in China have advanced significantly over the past two decades. This field has evolved from the quantitative reconstruction of single land-use types to multi-type gridded reconstructions, and the temporal scope has extended from the past 300 years to millennial or even longer scales, establishing a comprehensive methodological and data system covering multiple spatiotemporal scales.

This study is grounded in this robust foundation. We systematically compared the land-use datasets used in this study (He et al., 2023; He et al., 2025) with existing representative datasets (Liu and Tian, 2010; Cao et al., 2014; Yang et al., 2018; Yu et al., 2021). Our analysis reveals a high degree of consistency in the data sources of these studies: the provincial cropland data for the past 300 years reconstructed by Ge et al. (2004) serves as the common foundation for He et al. (2023), Liu and Tian (2010), and Yu et al. (2021); similarly, the forest data from He et al. (2008) is the shared source for He et al. (2025), Liu and Tian (2010), and Yang et al. (2018). Building on this foundation, each study has expanded in different dimensions: Liu and Tian (2010) pioneered the integration of high-resolution satellite data to construct a comprehensive dataset containing multiple land-use types; Yu et al. (2021) and Yang et al. (2018) enhanced spatial resolution by improving gridding methods; and the dataset used in this study incorporates the latest findings from historical documents to form a millennial series (He et al., 2023; He et al., 2025). Despite differences in methodologies, these datasets exhibit highly consistent trends over the overlapping period (the past 300 years) and at the provincial scale. Even for Cao et al. (2014), which adopted an independent

correction method, the reconstructed trend of cropland growth during the Qing Dynasty is largely consistent with our results. Compared to the aforementioned studies focusing on the past 300 or 100 years, the primary advantage of the dataset used in this study lies in providing a continuous perspective on a millennial scale. The introduction of such long-term data is crucial for comprehensively assessing the cumulative carbon emissions from land-use change in China.

### 4.2 Review of estimation methods

Compared with the carbon emission estimates from land-use changes in China over the past 300 years by Yang et al. (2023), this study updated and improved the land-use change data, carbon density data, and disturbance response curves. The specific improvements were as follows. First, building on multiple recent studies, the land-use change data for China from 1700 to 1980 were extended back to AD 1000 and forward to 2019, resulting in a land-use dataset with 131 time points spanning from 1000 to 2019. Second, in the calculation of land-use change rates, Yang et al. (2023) adjusted cropland data to align with 50- and 100-year time intervals, matching those of the forest and grassland data. However, cropland data from historical periods, which were reconstructed based on tax records for cropland areas in historical archives, are highly accurate and record rich information on cropland coverage changes. Adjusting to 50- and 100-year intervals often obscures many signals of cropland cover change, whereas this study preserved all such information.

Third,in the process of collecting carbon density data, we incorporated the results from China's Soil Survey Series, specifically the "Soil Series Atlas of China," which compiled 1,253 soil carbon density samples for various land cover types, including forests and grasslands. This significantly enriched the carbon density sample database, making the data more representative. Fourth, in the calculation of provincial carbon density, we assessed the normal distribution characteristics of the carbon density sample data for each province and chose either the arithmetic mean or median to obtain provincial average carbon densities, minimizing the influence of abnormally high or low values in the samples.

Fifth,Yang et al. (2023) noted that in historical periods, the area of forest converted to cropland was far greater than the area of cropland expansion. However, their study did not clearly explain how this "excess" forest loss was classified. To address this gap, our study classifies the remaining forest change (i.e., deforestation not resulting from cropland occupation) as shifting cultivation. This classification effectively captures the carbon emissions associated with the extensive vegetation loss and land degradation events described in the introduction, representing them as emissions from forest conversion to other land. This approach is based on the global historical land-use change scenarios of Houghton and Castanho (2023) and extensive historical records from China, and is more aligned with historical facts. Based on this, we developed land-use conversion rules suitable for provincial-level analysis in China and incorporated the methods and characteristics of cropland, forest, and grassland dataset reconstruction. Finally, the disturbance response curve was central to the bookkeeping model, driving the calculation of carbon budgets using annual land-use change rates and carbon density data. In this study, we used the latest published disturbance-response curve (Houghton and Castanho, 2023). In summary, this study updated and

improved both the data and models covering six specific areas, thereby increasing the reliability of carbon emission estimates.

## 4.3 Comparison with previous estimates

The results of this study were compared with a variety of published estimates (Table 3, Figure 11). Figure 11 illustrates the annual carbon fluxes estimated by various models, while Table 3 provides a summary of average annual fluxes across four historical stages. This table is further supplemented by literature reporting cumulative values of carbon fluxes only (Ge et al., 2008; Yang et al., 2019), for which the average fluxes were calculated from the reported totals. Significant discrepancies exist among the different estimates, primarily stemming from conceptual differences in how various models define the core concept of "land-use change emissions" (Obermeier et al., 2024; He et al., 2024; Yue et al., 2024; Gidden et al., 2023). These models can be broadly categorized into bookkeeping models, dynamic global vegetation models (DGVMs), and data from national greenhouse gas inventories (NGHGI).

**Table 3.** Comparison of average annual carbon flux estimates caused by land-use change in China

| Reference | Model Type[1] | Average annual carbon flux (Tg C yr$^{-1}$) | | | |
|---|---|---|---|---|---|
| | | Pre-1900 | 1900-1949 | 1950-1980 | Post-1980 |
| This Study | BKM | 40.50 | 75.51 | 106.39 | -55.23 |
| Yang et al. (2023) | BKM | 25.79 | 51.31 | 42.76 | / |
| Ge et al. (2008) | BKM | 18.48 | 49.70 | / | / |
| Yang et al. (2019) | BKM | 23.54 | 42.25 | 149.36 | / |
| Houghton and Castanho (2023) | BKM | 48.14 | 100.59 | 6.57 | -6.98 |
| Qin et al. (2024) | BKM | -3.81 | 77.06 | 236.13 | 73.68 |
| Gasser et al. (2020) | BKM | / | / | 82.50 | 20.23 |
| Hansis et al. (2015) | BKM | / | / | 126.70 | 212.73 |
| GCB2024[2] (Friedlingstein et al., 2025) | BKM | 21.20 | 83.64 | 164.33 | 36.27 |
| Zhu et al. (2025)[3] | Stock-Difference Method | / | / | / | -49.33 |
| TRENDYv8.present.day[4,5] | DGVM | / | / | 107.40 | 190.39 |
| FAOSTAT[4,6] | National Reports | / | / | / | -121.06 |
| NGHGI.DB.corrected[4,7] | National Reports | / | / | / | -180.36 |

Note: The values in the table represent the average annual carbon fluxes (Tg C yr$^{-1}$) for the specified time periods. A "/" indicates that data for that period is not available or not provided in the source study. Pre-1900 denotes the period 1700-1899.

[1] Model Type: BKM (Bookkeeping Models) aim to estimate carbon fluxes from direct anthropogenic land-use activities; DGVM (Dynamic Global Vegetation Models) simulate the integrated response of ecosystems to both land use and environmental changes, which includes the additional loss of sink capacity (LASC) that would otherwise occur, for example, on an actually cleared forest (refer to Gasser et al., 2020 for a detailed explanation); National Reports (NGHGI) are based on the IPCC's "managed land proxy" principle for accounting, which includes carbon sink driven by both direct anthropogenic land use and also environmental effects, but not the LASC.

[2] The value for GCB2024 is the mean of the carbon fluxes derived by the four bookkeeping models: Qin et al. (2024), Houghton and Castanho (2023), Gasser et al. (2020), and Hansis et al. (2015).

[3] Zhu et al. (2025) employs a stock-difference method, constructing high-resolution, dynamic carbon stock maps by integrating remote sensing, inventory data, and machine learning, and calculates the flux from changes in these maps over time.

[4] Data for TRENDYv8.present.day, NGHGI.DB.corrected, and FAOSTAT are from Obermeier et al., (2024).

[5] TRENDYv8.present.day represents DGVM simulation results run under fixed, modern environmental conditions (climate and $CO_2$ concentration), designed for conceptual alignment with bookkeeping models that use modern carbon densities.

[6] FAOSTAT provides bottom-up estimates by applying IPCC guidelines to country-reported activity data (e.g., from the 595 Forest Resources Assessment) and geospatial information. Conceptually, these estimates are closer to bookkeeping models as they often do not include the indirect environmental effects (such as $CO_2$ fertilization).

[7] NGHGI.DB.corrected is derived from the original national inventory data (NGHGI.DB) by subtracting the carbon fluxes on "managed land" that are caused by indirect environmental changes (e.g., $CO_2$ fertilization), as estimated by DGVMs. This makes this estimate conceptually aligned with the bookkeeping models.

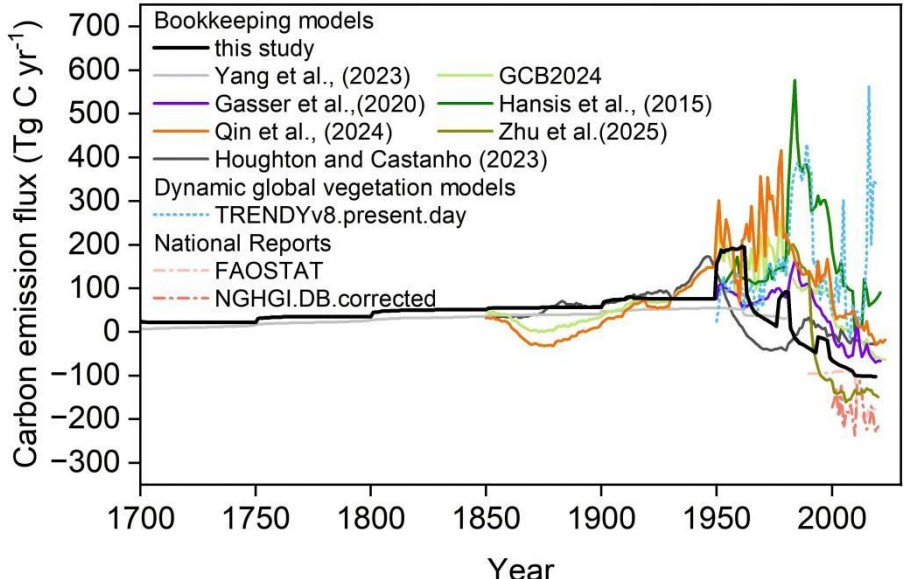

**Figure 11.** Chinese historical land-use change-induced carbon emission fluxes estimated by different studies. For detailed descriptions of all data sources, models, and methodologies shown, please refer to the notes of Table 3.

Bookkeeping models, the method employed in this study, aim to estimate carbon fluxes caused by direct human activities (e.g., deforestation, afforestation) by excluding the indirect environmental changes caused by anthropogenic greenhouse gas emissions to date. For this purpose, environmental conditions (such as $CO_2$ concentration and climate) are typically assumed to be constant to ensure that the indirect carbon sink/source effects induced by environmental changes are excluded. In contrast, DGVMs simulate the integrated response of ecosystems to both land use and environmental changes (e.g., climate, 610    $CO_2$) (Obermeier et al., 2024). Their estimates often include the "loss of additional sink capacity", which is the potential additional carbon sink that would have formed in the original ecosystem under favorable environmental conditions but was lost due to land-use change (e.g., converting a forest to cropland). This means their estimated emissions conceptually include an "opportunity cost" not included in bookkeeping models, leading to generally higher carbon emissions from

forest/grassland clearing. National Reports, based on the "managed land proxy" principle from IPCC inventory guidelines, treat all carbon fluxes on "managed land" (including both direct and indirect effects) as anthropogenic contributions but the "loss of additional sink capacity" was not included.

Taking into account the conceptual framework described above, a comparison of this study with other bookkeeping models (Table 3) reveals significant differences in the magnitude and dynamics of carbon fluxes across different periods (Fig. 11). For instance, in the pre-1900 period (the earliest year of all studies in Table 3 is 1700), estimates of the average annual flux range from a small sink of -3.81 Tg C yr⁻¹ (Qin et al., 2024) to a significant source of 48.14 Tg C yr⁻¹ (Houghton and Castanho, 2023), with our study estimating a source of 40.50 Tg C yr⁻¹. These discrepancies primarily reflect differences in historical land-use data reconstruction and model parameterization. In the modern era, particularly post-1980, the differences among bookkeeping models are more pronounced. Our study reports a net sink of -55.23 Tg C yr⁻¹. In contrast, the GCB2024 estimate (Friedlingstein et al., 2025), which is an average of four bookkeeping models, shows a net source of 36.27 Tg C yr⁻¹. This is mainly because the latter's underlying models primarily use global-scale FAO statistics or different versions of the LUH dataset, which have been shown to considerably underestimate recent forest expansion biases in China (Yu et al., 2022). It is noteworthy that the findings of Zhu et al. (2025) (average post-1980 sink of -49.33 Tg C yr⁻¹) are relatively close in magnitude to our estimates (-55.23 Tg C yr⁻¹) (Fig. 11). A common feature of both studies is the use of local land-use data, rather than relying on global-scale products. This shared data foundation allows both studies to independently corroborate a strong net carbon sink signal in China since the 1980s, a signal that models relying on global datasets have not been able to capture.

Comparison with other model types further highlights the influence of conceptual differences in terms of accounting boundaries of land-use carbon fluxes. The net carbon sink estimated in our study (-55.23 Tg C yr⁻¹) for the period 1980-2019 primarily reflects the outcomes of direct activities such as afforestation. In contrast, the estimate from TRENDYv8 (a DGVM) remains a strong carbon source of 190.39 Tg C yr⁻¹ (Table 3). One reason is that, similar to some of the aforementioned bookkeeping models underlying GCB2024, the TRENDYv8 DGVMs uniformly use the global-scale LUH2 dataset, which underestimated forest expansion in China. The second reason is the conceptual difference between TRENDYv8 and the bookkeeping approach of our study: the TRENDYv8 not only includes direct emissions from activities such as deforestation but also incorporates the "loss of additional sink capacity"—the potential carbon absorption that was lost because of the historical loss of forest areas.

The national report of NGHGI.DB.corrected dataset is the most conceptually comparable with the bookkeeping method used in this study, as it removed the indirect environmental effects. NGHGI.DB.corrected data shows a large net sink of -180.36 Tg C yr⁻¹ for the 2000-2020 period. This is consistent with our finding of a net carbon sink in the modern era, but the magnitude by the national report is substantially larger (Fig. 11). This might be because NGHGI includes carbon sink over lands that have constant land cover but subject to other forms of management, including forest protection which allows

forest recovery from post disturbances, grassland grazing exclusion, fire suppression and cropland intensification (Yue et al., 2024), but the carbon effects of these activities are not included in our study as it focuses on land use change only. This suggests that after accounting for key conceptual differences, reconciling the remaining discrepancy between national reports and bookkeeping models requires bookkeeping models to expand on the scope of land use activities.

In summary, the comparison of our study with other estimates establishes the robustness of our estimates in the context of the uncertainties, and the conceptual differences of different studies to date. Nonetheless, the unique contribution of this study lies in providing the long-term and most regionally detailed estimate of carbon fluxes driven by direct land-use activities in China.

**4.4 Implications and Applications**

For regional carbon budget assessment, the dataset provides a robust historical baseline for carbon fluxes from land-use change, enabling the separation of legacy emissions from contemporary fluxes. This is crucial for accurately attributing the drivers of the current terrestrial carbon sink and evaluating the effectiveness of ecological restoration efforts. In climate and Earth system modeling, the dataset serves as an independent benchmark for evaluating and refining DGVMs. Validation

against the provincially-resolved emission estimates from this study can help constrain model parameters related to ecosystem responses to land-use change. For policy evaluation, the dataset offers long-term quantitative evidence to assess the efficacy of land-use policies. The key transition from a carbon source to a sink around the 1980s strongly coincides with the implementation of China's large-scale ecological restoration policies, thus supporting the assessment of the potential effectiveness of such national-level interventions.

**4.5 Limitations**

This study utilizes data from the Second and Third National Land Surveys to link with reconstructed historical data. Compared to annual remote sensing-based land use products, the national survey data is more compatible with the reconstructed historical data in terms of its sources and inherent characteristics. However, we acknowledge that

discrepancies exist in the classification standards and statistical calibers across different periods, such as the changes between the Second and Third surveys. To mitigate the impact of these inconsistencies, we made a specific adjustment for the forest land category, a key factor in the carbon cycle. According to the definition in the source literature for the reconstructed historical data, its conceptualization of 'forest' most closely aligns with 'closed forest land' in the current classification standards. Therefore, we used the 'closed forest land' sub-category, under the primary 'forest land' category

from the Second and Third surveys, to ensure definitional consistency. Despite this effort, we recognize that the differences

in statistical calibers among data sources remain a source of uncertainty in our carbon budget estimation, which awaits future improvement through more refined data harmonization and calibration methods. The long-term land-use data used in this study, including reconstructed and derived data from statistical surveys, represent the net land-use values within the statistical units. However, actual land-use changes, as indicated by remote sensing data, show that within a given area of a particular land-use type, certain pixels represent either increased area or decreased area. The total change due to these increases and decreases is much greater than the net change, and such detailed variations cannot currently be captured by long-term historical land-use datasets. Therefore, uncertainty in basic land-use data leads to inherent uncertainty in the estimated carbon budget associated with land-use changes.

When calculating annual land-use change rates, the classification of land-use types is relatively coarse due to data limitations. Land-use types other than cropland, forest, and grassland are all grouped together as "other land," and land-use conversion rules are established based on this classification to calculate the annual land-use change rates. Compared with modern remote sensing-based land-use data, long-term land-use data are less detailed, which also affects the accuracy of carbon budget estimates related to land-use change.

Additionally, the spatiotemporal variability of basic carbon density values can influence the accuracy of the estimates. In this study, carbon density is addressed using a "present-day-for-past" substitution method. Although modern soil carbon densities were moderately adjusted by incorporating a large-scale soil sampling survey dataset from the post-1949 period in China, pre-industrial carbon stocks likely varied due to shifts in atmospheric $CO_2$ concentrations, climate fluctuations, ecological succession, and human land management. Vegetation and soil carbon densities were not static over the past millennium. Therefore, using static values to represent historical carbon densities may fail to capture temporal dynamics, thereby introducing uncertainties. Potential biases include overestimating human contributions if climate-driven increases in carbon density are ignored and overestimating modern carbon uptake if long-term baseline declines in carbon stocks are not included. Future studies should explore coupling DGVMs (e.g., LPJ or ORCHIDEE) to simulate combined impacts of historical climate, $CO_2$ levels, and human activities on carbon density. In this study, to quantitatively assess the uncertainty introduced by the assumption of static carbon density, we conducted a sensitivity analysis. This analysis was based on two core hypotheses: 1) historical vegetation carbon density was 20% lower than modern values, accounting for the reduced $CO_2$ fertilization effect in the pre-industrial era; and 2) historical soil carbon density was 20% higher, reflecting less intensive anthropogenic disturbance and more intact ecosystems. The results show that this scenario leads to a systematic reduction in the estimated annual net carbon emissions and reveals distinct temporal patterns. During the dominant carbon source periods, this reduction was relatively stable (mean annual difference of approx. -2 Tg C yr$^{-1}$). In contrast, during carbon sink periods, which were often accompanied by intense land-use change, the discrepancy between the two estimates exhibited greater volatility, leading to a significant amplification of uncertainty in these phases. This trend indicates that using static modern carbon density values may lead to an overestimation of the historical carbon source effect.

We reiterate that the carbon emission accounting method in the present study does not include wood harvesting. Considering that wood harvesting represents a significant historical source of anthropogenic emissions, the absence of these data may lead to a certain degree of underestimation in the corresponding carbon emission fluxes. Fortunately, Houghton and Castanho (2023) estimated China's long-term carbon emissions from wood harvesting and found values of 5 Tg C yr$^{-1}$ for 2011–2020, approximately 20–30 Tg C yr$^{-1}$ around the 1950s, approximately 5–10 Tg C yr$^{-1}$ in the 1900s, and less than 5 Tg C yr$^{-1}$ before 1900. These estimates can serve as a reference when regional long-term reconstructed data on wood harvesting and their corresponding carbon emission estimates are unavailable.

## 5. Data availability

The dataset of annual carbon emissions from land-use change in China (1000–2019) is available at https://doi.org/10.5281/zenodo.14557386 (Yang et al., 2025).

## 6. Conclusion

Reducing the uncertainty in carbon budget estimates from land-use change has become a frontier in global change science and is receiving widespread attention, as it plays a crucial role in achieving the global "carbon neutrality" target. This study provides an estimation of the annual carbon emissions from land-use changes in China from 1000 to 2019. High-confidence long-term land-use change datasets, extensive vegetation and soil carbon density sampling data, and the latest published disturbance-response curves effectively minimized the uncertainties in previous long-term carbon budget estimates for China.

From 1000 to 2019, carbon emissions resulting from land-use changes in China amounted to 19.61 Pg C. Four distinct phases were identified. The first phase, which occurred before the early 18th century (1000–1700), saw a slow increase in carbon sources, with a total emission of 6.60 Pg C, accounting for 30.17% of the total, at an average annual rate of 9.46 Tg C yr$^{-1}$. The second phase, which occurred from the early 18th century to the early 1980s (1700–1980), experienced rapid growth in carbon sources (15.27 Pg C, 69.86%, 54.09 Tg C yr$^{-1}$). The third phase, which occurred from the 1980s to the late 1990s (1980–1998), saw a reversal in the carbon balance, with land-use changes shifting from carbon sources to carbon sinks (carbon sink of 0.12 Pg C, 16.85 Tg C yr$^{-1}$). The fourth phase, which occurred from the late 1990s to 2019 (1998–2019), saw a further enhancement of the carbon sink (1.85 Pg C, 88.21 Tg C yr$^{-1}$).

**Author contributions.** FY and GD designed the work. FY, FH, and ML provided historical provincial cropland, forest, and grassland data for quantifying China's annual carbon budget from land-use change. FY, WL, ZL, and XZ collected vegetation and soil carbon density sample point data. RAH provided the disturbance-response curve parameter table. FY,

XM, RAH, and YG devised the land-use transition rules. FY, GD, XM, HZ, PW, FH, QM, YY, BL, and CY reviewed and edited the text. FY prepared the manuscript and drafted the manuscript with contributions from all coauthors.

**Competing interests.** One of the (co-)authors is a member of the editorial board of Earth System Science Data.

**Disclaimer.** Publisher's note: Copernicus Publications remains neutral with regard to jurisdictional claims made in the text, published maps, institutional affiliations, or any other geographical representation in this paper. While Copernicus Publications makes every effort to include appropriate place names, the final responsibility lies with the authors. Publisher's

remark: please note that Figs. 1, 4, 8, and B1 contain disputed territories.

**Financial support.** This research has been supported by the National Key Research and Development Program of China (No. 2023YFB3907403), the National Natural Science Foundation of China (No. 42201263), and the Key Scientific and Technological Project of Henan Province (No. 232102321103).

# Appendix A

**Table A1.** Detailed reference for the second and third national land survey bulletins.

| Items | Time point | Land-use types | Province | Data source/Download link |
|---|---|---|---|---|
| The Second National Land Survey of China | 2009 | cropland, forest, and grassland | all provinces | https://gtdc.mnr.gov.cn/shareportal#/ |
| The Third National Land Survey of China | 2019 | cropland, forest, and grassland | Henan | https://www.henan.gov.cn/2022/04-18/2433857.html |
| | | | Shanxi | http://www.shanxi.gov.cn/ywdt/zwlb/bmkx/202201/t20220127_6441197.shtml |
| | | | Shandong | http://dnr.shandong.gov.cn/zwgk_324/xxgkml/ywdt/tzgg_29303/202112/t20211216_3810111.html |
| | | | Hebei | https://zrzy.hebei.gov.cn/heb/gongk/gkml/gggs/qtgg/zrdcc/10671417206794772480.html |
| | | | Liaoning | https://www.ln.gov.cn/web/ywdt/jrln/wzxx2018/EFA7CA9476D44D8D85578D867D70EA56/index.shtml |
| | | | Jilin | http://www.jl.gov.cn/szfzt/xwfb/xwfbh/xwfbh2021/jlsdssjrmdbdhdychy_409635/ |
| | | | Heilongjiang | http://www.dview.com.cn/rjcp_zz_3741.html |
| | | | Jiangsu | http://news.yznews.com.cn/2021-12/31/content_7347606.htm |
| | | | Zhejiang | https://zrzyt.zj.gov.cn/art/2021/12/3/art_1289933_58988406.html |
| | | | Anhui | https://zrzyt.ah.gov.cn/public/7021/146407571.html |
| | | | Fujian | http://zrzyt.fujian.gov.cn/zwgk/zfxxgkzl/zfxxgkml/tdgl_19753/202112/t20211231_5805488.htm |
| | | | Jiangxi | http://bnr.jiangxi.gov.cn/art/2021/12/29/art_35804_3810534.html |
| | | | Hubei | https://zrzyt.hubei.gov.cn/fbjd/xxgkml/sjfb/tdzytjsj/202112/t20211217_3919353.shtml?eqid=e3b66db3004cc51a00000006647fe835 |
| | | | Hunan | http://www.hunan.gov.cn/hnszf/zfsj/sjfb/202112/t20211207_21275973.html?share_token=83aa6011-7231-4c49-8a14-a14f9ae0c29b |
| | | | Guangdong | http://www.jiangmen.gov.cn/jmzrj/gkmlpt/content/2/2507/post_2507058.html?eqid=87430417001fc53900000003648a53ca#187 |
| | | | Hainan | https://www.hainan.gov.cn/hainan/0101/202110/8c92db59ef6f4468b96b058465ba60b2.shtml |
| | | | Sichuan | http://dnr.sc.gov.cn/scdnr/scsdcsj/2022/1/18/3e1bc5eb55db44628498b5db740eac5b.shtml |
| | | | Guizhou | http://www.guizhou.gov.cn/zwgk/zdlygk/jjgzlfz/zrzy/zrzydcjcgl/202201/t20220121_72378280.html |
| | | | Yunnan | https://www.yn.gov.cn/sjfb/tjgb/202112/t20211221_231929.html |

| | |
|---|---|
| Shaanxi | https://zrzyt.shaanxi.gov.cn/info/1038/57862.htm?eqid=892a22b000028c4b00000006644b72c5 |
| Gansu | https://baijiahao.baidu.com/s?id=1712097491056856575&wfr=spider&for=pc |
| Qinghai | https://zrzyt.qinghai.gov.cn/gk/sj/zrzygb/content_4922 |
| Beijing | https://ghzrzyw.beijing.gov.cn/zhengwuxinxi/sjtj/tdbgdctj/202111/t20211105_2529986.html |
| Tianjin | https://ghhzrzy.tj.gov.cn/zwgk_143/tzgg/202111/t20211118_5712899.html |
| Shanghai | https://ghzyj.sh.gov.cn/zcfg-tdgl/20220107/b513d306e88b41bebc7b7b8a5b5cc56c.html |
| Chongqing | http://tjj.cq.gov.cn/zwgk_233/fdzdgknr/tjxx/sjzl_55471/tjgb_55472/202111/t20211125_10031239.html |
| Inner Mongolia | https://zrzy.nmg.gov.cn/zwgk/tztg/202205/t20220507_2051673.html |
| Guangxi | https://dnr.gxzf.gov.cn/zfxxgk/fdzdgknr/tjfx/zhtj/t16084757.shtml |
| Xizang | http://zrzyt.xizang.gov.cn/gk/gsgg/202112/t20211224_276279.html |
| Ningxia | https://www.nx.gov.cn/zwgk/tzgg/202112/t20211206_3205422_zzb.html |
| Xinjiang | http://zrzyt.xinjiang.gov.cn/xjgtzy/gzdt/202201/c7061f858692402da4f7b65e376cd2fb.shtml |

* Last access: May 2024.

## Appendix B

**Table B1.** Detailed information for soil series in China

| Title | Publisher | Year |
|---|---|---|
| Soil Series in China: Anhui | Science Press | 2017 |
| Soil Series in China: Beijing and Tianjin | Science Press | 2016 |
| Soil Series in China: Hebei | Science Press | 2017 |
| Soil Series in China: Shandong | Science Press | 2019 |
| Soil Series in China: Henan | Science Press | 2019 |
| Soil Series in China: Jiangsu | Science Press | 2017 |
| Soil Series in China: Shanghai | Science Press | 2017 |
| Soil Series in China: Hubei | Science Press | 2017 |
| Soil Series in China: Fujian | Science Press | 2017 |
| Soil Series in China: Zhejiang | Science Press | 2017 |
| Soil Series in China: Hainan | Science Press | 2018 |
| Soil Series in China: Heilongjiang | Science Press | 2020 |
| Soil Series in China: Jilin | Science Press | 2019 |
| Soil Series in China: Liaoning | Science Press | 2020 |
| Soil Series in China: Guangdong | Science Press | 2017 |
| Soil Series in China: Central and Western Volume: Shanxi | Science Press Longmen Press | 2020 |
| Soil Series in China: Central and Western Volume: Shaanxi | Science Press Longmen Press | 2020 |
| Soil Series in China: Central and Western Volume: Inner Mongolia | Science Press Longmen Press | 2020 |
| Soil Series in China: Central and Western Volume: Ningxia | Science Press Longmen Press | 2020 |
| Soil Series in China: Central and Western Volume: Qinghai | Science Press Longmen Press | 2020 |
| Soil Series in China: Central and Western Volume: Hunan | Science Press Longmen Press | 2020 |
| Soil Series in China: Central and Western Volume: Jiangxi | Science Press Longmen Press | 2020 |
| Soil Series in China: Central and Western Volume: Sichuan | Science Press Longmen Press | 2020 |
| Soil Series in China: Central and Western Volume: Chongqing | Science Press Longmen Press | 2020 |
| Soil Series in China: Central and Western Volume: Gansu | Science Press Longmen Press | 2020 |
| Soil Series in China: Central and Western Volume: Guangxi | Science Press Longmen Press | 2020 |
| Soil Series in China: Central and Western Volume: Guizhou | Science Press Longmen Press | 2020 |
| Soil Series in China: Central and Western Volume: Yunnan | Science Press Longmen Press | 2020 |
| Soil Series in China: Central and Western Volume: Xinjiang | Science Press Longmen Press | 2020 |

**Table B2.**   Disturbance response curve parameter.

| Ecological zone | Land-use change | Soil decay | Soil recovery |
|---|---|---|---|
| Temperate Desert | FC, FO, GC | 3% per year (first 4 years) <br> 1% per year (last 11 years ) | 0.3% per year (50 years) |
| Temperate Steppe | FC, FO, GC | 3% per year (first 4 years) <br> 1% per year (last 11 years ) | 0.41% per year (37 years) |
| Temperate Continental | FC, FO, GC | 3% per year (first 4 years) <br> 1% per year (last 11 years ) | 0.3125% per year (48 years) |
| Subtropical Humid | FC, FO, GC | 3% per year (first 4 years) <br> 1% per year (last 11 years ) | 0.3125% per year (48 years) |

| Ecological zone | Land-use change | Living vegetation decay | Living vegetation recovery |
|---|---|---|---|
| Temperate Desert | FC, FO | 95% per year (1 year) | 0.02% per year (50 years) <br> 0.06% per year (30 years) |
| Temperate Desert | FG | 95% per year (1 year) | 0.05% per year (28 years) <br> 0.09% per year (20 years) |
| Temperate Steppe | FC, FO | 95% per year (1 year) | 0.24% per year (37 years) <br> 0.09% per year (50 years) |
| Temperate Steppe | FG | 90% per year (1 year) | 0.11% per year (37 years) <br> 0.09% per year (50 years) |
| Temperate Continental | FC, FO | 95% per year (1 year) | 1.63% per year (48 years) <br> 0.56% per year (50 years) |
| Temperate Continental | FG | 90% per year (1 year) | 1.99% per year (37 years) <br> 0.56% per year (50 years) |
| Subtropical Humid | FC, FO | 95% per year (1 year) | 1.63% per year (48 years) <br> 0.56% per year (50 years) |
| Subtropical Humid | FG | 90% per year (1 year) | 1.99% per year (37years) <br> 0.56% per year (50 years) |

| Ecological zone | Land-use change | Fraction Goes to Slash | Decay rate |
|---|---|---|---|
| Temperate Desert, Temperate Steppe, Temperate Continental | FC, FO | 50% | 10% each year based on the value of the previous year |
| Temperate Desert, Temperate Steppe, Temperate Continental | FG | 33% | 10% each year based on the value of the previous year |
| Subtropical Humid | FC, FO | 50% | 50% each year based on the value of the previous year |
| Subtropical Humid | FG | 33% | 50% each year based on the value of the previous year |

Note: FC refers to the conversion between Forest and Cropland. FO refers to the conversion between Forest and Other land. FG refers to the conversion between Forest and Grassland. GC refers to the conversion between Grassland and Cropland.

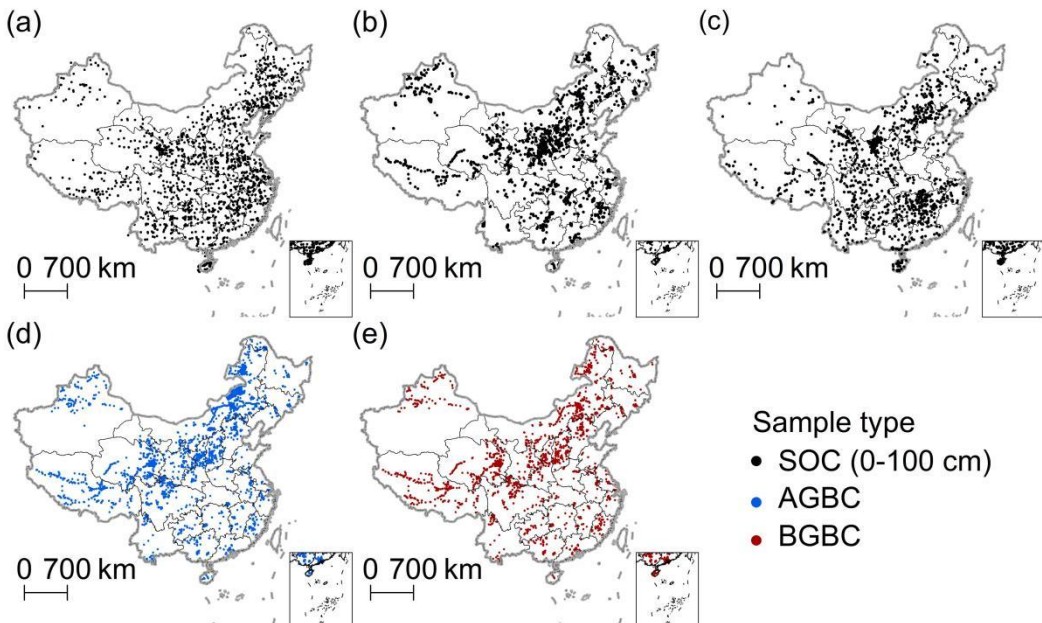

**Figure B1.** Distribution of sample points for vegetation carbon density and soil carbon density. (a) SOC is derived from the 2010s China's terrestrial ecosystem carbon density dataset (Xu et al., 2019). (b) SOC is derived from the "Soil Chronicles of China." (c) SOC is derived from the "Soil Series of China." SOC refers to soil organic carbon. AGBC refers to above-ground biomass carbon, and BGBC refers to below-ground biomass carbon.

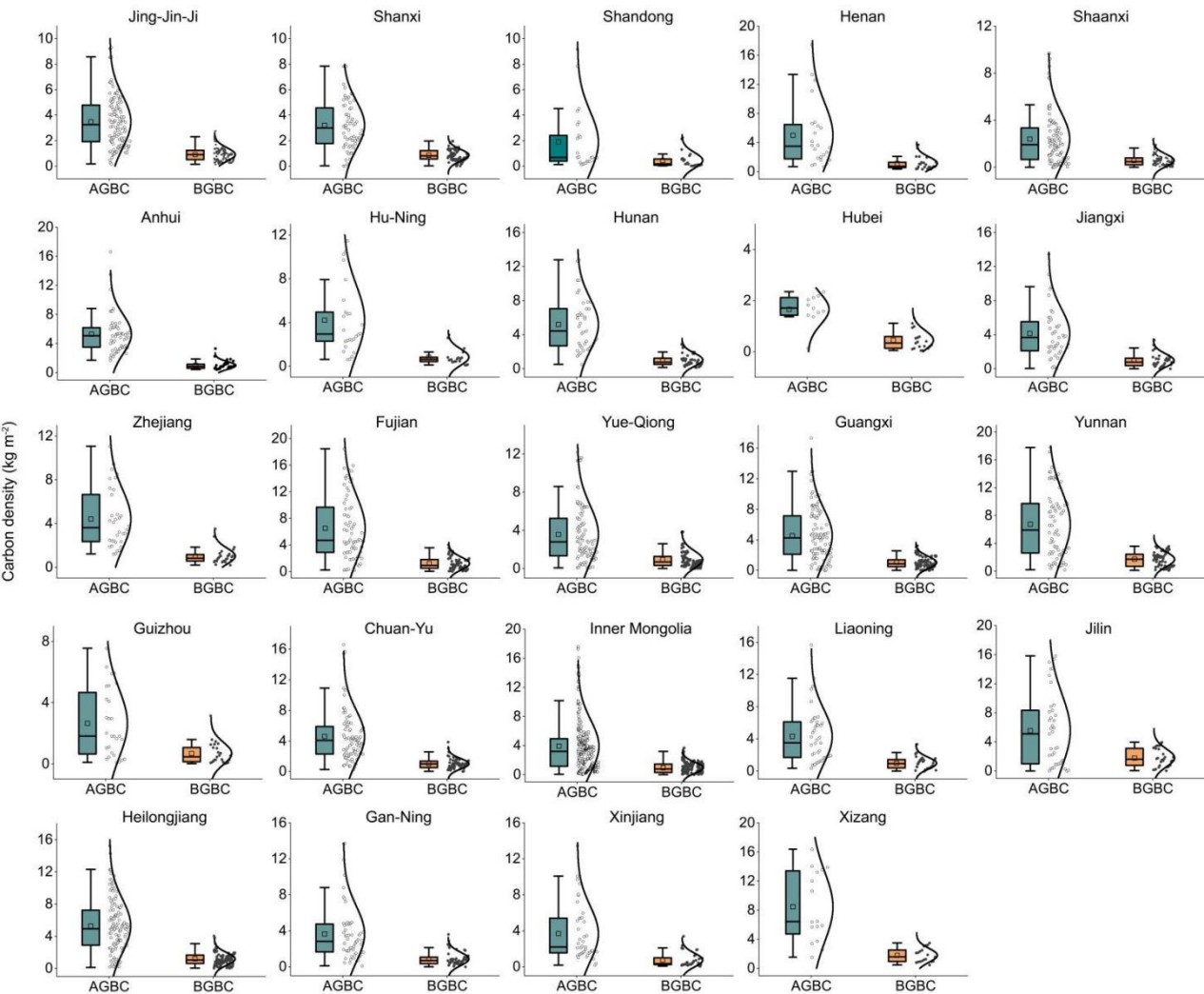

**Figure B2.** Distribution of forest vegetation carbon density at provincial scale. AGBC refers to above-ground biomass carbon, and BGBC refers to below-ground biomass carbon.

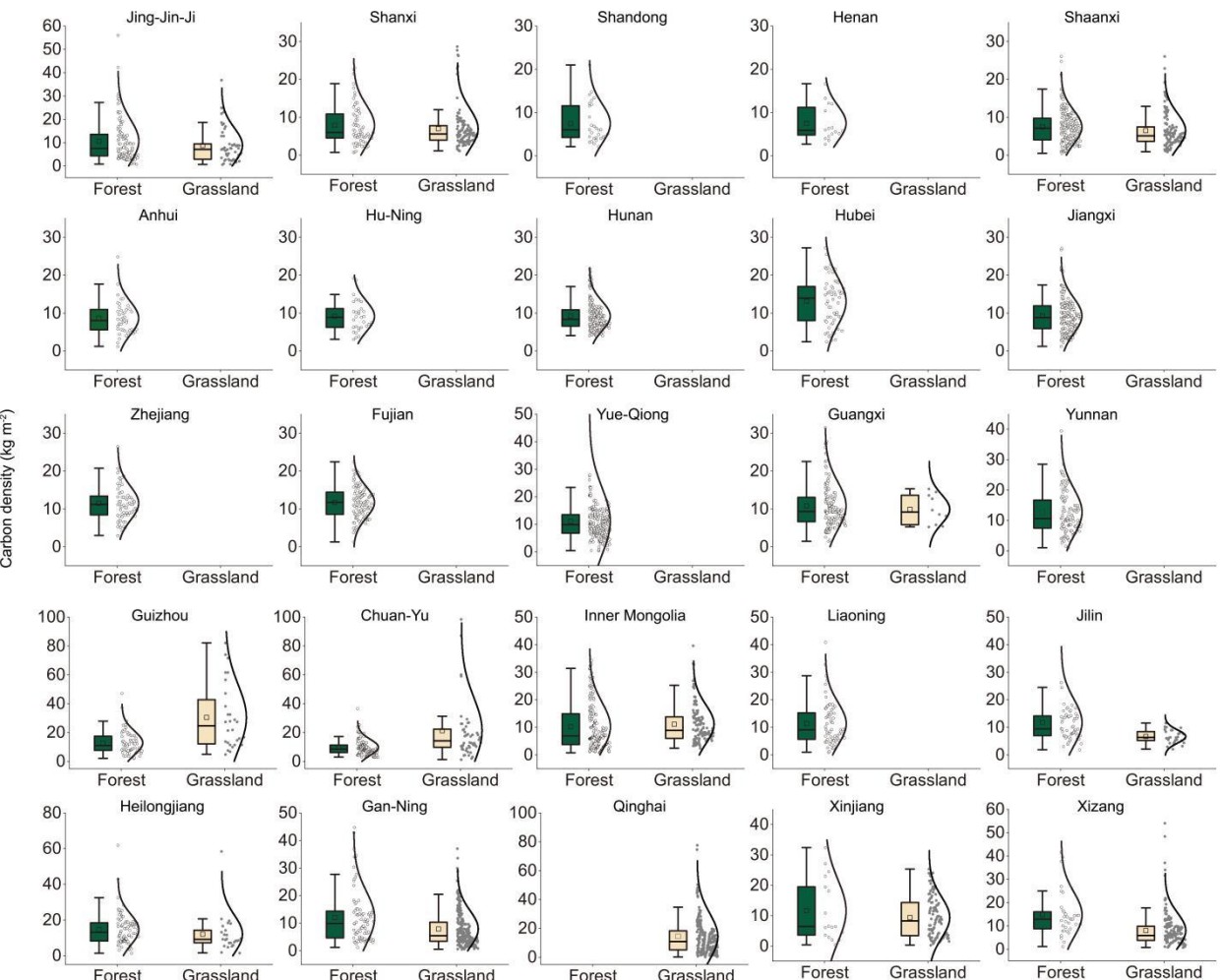

**Figure B3.** Distribution of soil carbon density in forests and grasslands at provincial scale.

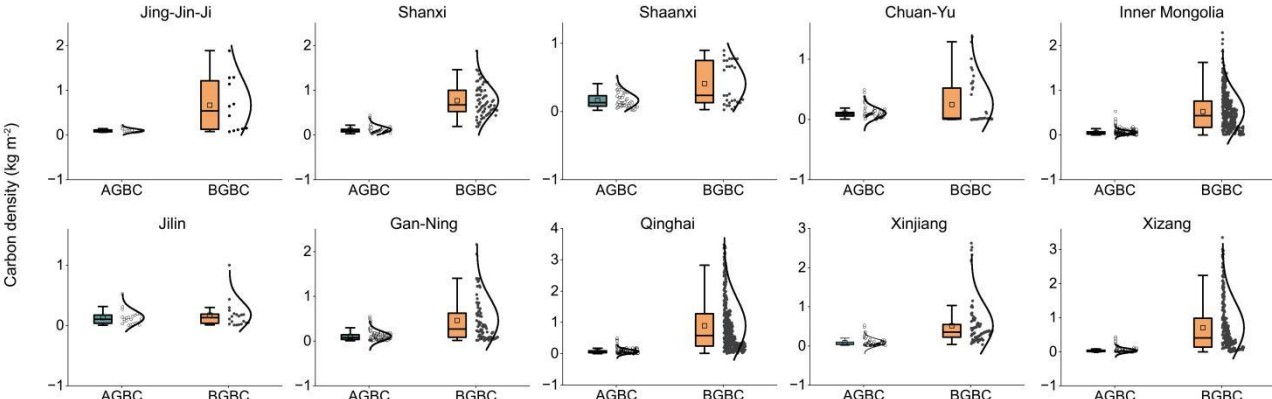

**Figure B4.** Distribution of grassland vegetation carbon density at provincial scale. AGBC refers to above-ground biomass carbon, and BGBC refers to below-ground biomass carbon.

## Appendix C

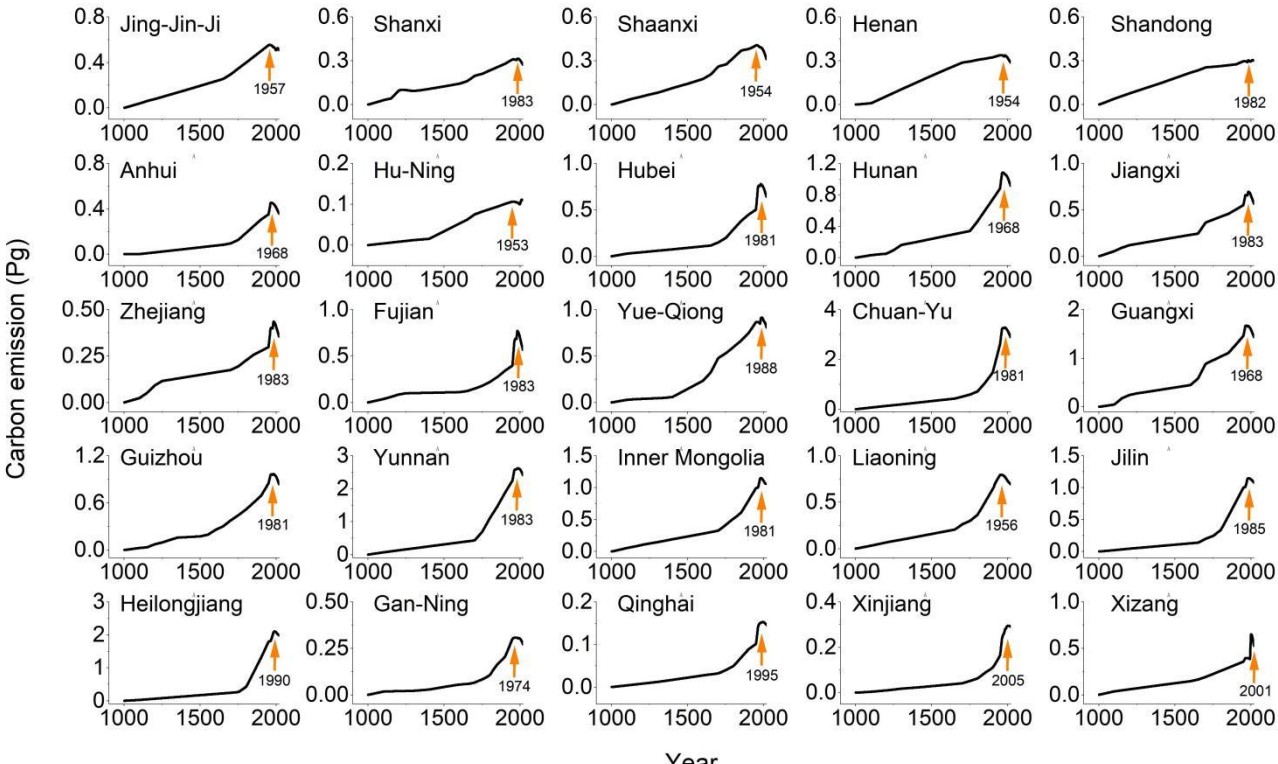

**Figure C1.** Cumulative carbon emissions from land-use changes at the provincial level. Arrows indicate the turning points from carbon sources to carbon sinks, with numbers representing the corresponding years of the turning points.

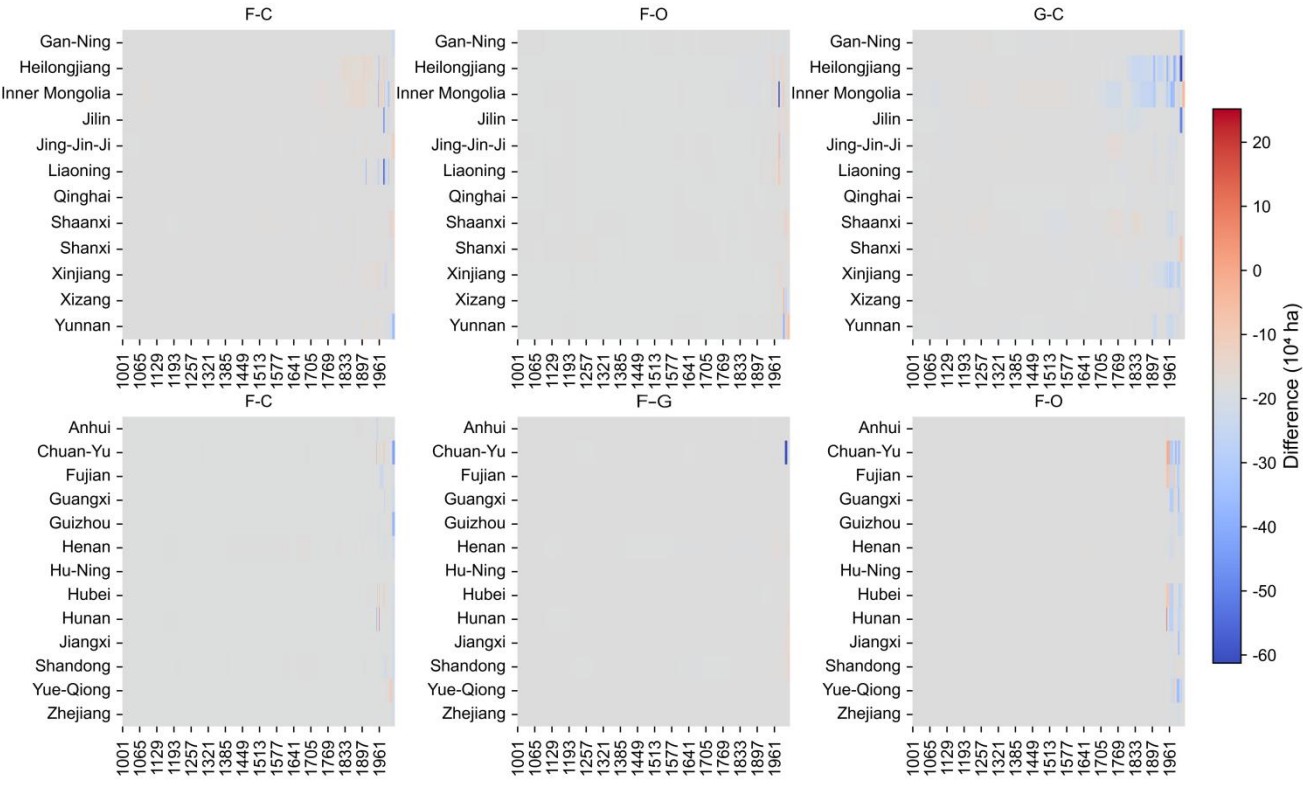

**Figure C2**. Differences in annual land-use transitions between the grassland-priority and area-weighted allocation methods. F-C denotes the conversion between Forest and Cropland, F-G represents Forest-Grassland conversion, F-O represents Forest-Other land conversion, and G-C represents Grassland-Cropland conversion. A positive difference indicates that the grassland-priority result is lower than the area-weighted result, and vice versa.

**Table C1.** Carbon emissions from land-use change and their uncertainties in different historical periods

| Period | Mean annual estimate (Tg C) | Upward uncertainty | Downward uncertainty |
|---|---|---|---|
| Pre-1900 | 16.35 | 7.49 | 5.20 |
| 1900-1949 | 75.51 | 37.31 | 24.65 |
| 1950-1982 | 102.95 | 74.35 | 44.48 |
| Post-1982 | -60.90 | 34.56 | 58.91 |

Notes: All values are in the unit of Tg C yr$^{-1}$. The mean upward uncertainty is the average of the annual differences between the maximum value and the estimate within each period. The mean downward uncertainty is the average of the annual differences between the estimate and the minimum value.

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
