# Peer review of "Annual carbon emissions from land-use change in China from 1000 to 2019"

_Earth System Science Data, 2025_

## Author Response (AR1)

**Response to Reviewer 1 Comments**

This study provides a unique millennium-scale perspective on land-use change (LUC) emissions in China, addressing critical gaps in reconstructing historical LUC data and updating contemporary emissions modeling. While the data and modeling are not perfect at this point, the study has made great improvements to LUC data since the 1000s, and updated carbon densities for current biomass and soil. The manuscript is well-structured and easy to follow, but its contributions and methodological choices require further clarification to strengthen its impact. I would recommend publication after revisions.

Response: Thank you for this comment and your recognition of our manuscript. Your comments enable us to pinpoint issues within the manuscript accurately and provide us with guidance for improvement. I am glad to have such an opportunity to communicate with you. We have carefully revised the manuscript according to your comments and suggestions. For detailed revisions to the manuscript text, please refer to the revised draft where changes are highlighted in red font.

Thanks again for your help in improving the manuscript.

**Major Concerns:**

**Point 1.** The study's novelty should be explicitly contextualized. Why is a millennium-scale analysis of LUC emissions critical, given the inherent uncertainties in pre-industrial data? How does this long-term perspective enhance our understanding of anthropogenic impacts on carbon cycling, even when CO2 levels were relatively stable before industrialization? China's uniquely long historical record enables this work, but how might its findings inform global LUC emission estimates, particularly for regions with limited historical documentation?

Response: Thank you for this comment.

Your comment is very important for improving our manuscript. According to your suggestions, we have made the necessary revisions in the introduction to address why

we chose the past millennium, what makes this period special, the significance of conducting research over such a long time scale, and whether the results can provide insights or references for other countries and regions. Please refer to lines 71–84 in the main text, where the revised content is marked in red. The revised excerpt is as follows:

"Although most global and regional studies on land-use change focus on the post-industrial era or the past three centuries, China's intensive and extensive land-use activities date back at least a millennium, thus representing a unique historical trajectory (He et al., 2025, 2023). From approximately AD 1000 (coinciding with the Northern Song Dynasty), ecological degradation in China showed a marked rise. This degradation was manifested through multiple pathways: accelerated erosion on the Loess Plateau, recurrent floods in the lower Yellow River Basin, large-scale lake siltation and disappearance in northern China, and progressive soil erosion coupled with natural vegetation loss in the southern hill regions (Wu et al., 2020; Chen et al., 2012). Such millennial-scale land-use transitions would have generated substantial carbon emissions, particularly from deforestation. However, the relatively stable pre-industrial global $CO_2$ concentrations likely obscured these regionally significant anthropogenic carbon fluxes because localized emissions in areas such as China could have been offset by concurrent carbon sinks elsewhere. Additionally, the full trajectory or specific stages of historical land-use change in China can serve as a "historical analogue" for other developing countries. For many countries and regions, systematically revealing the processes and mechanisms of land-use change and associated carbon emissions—driven by long-term population growth and policy shifts—can help overcome the limitations associated with a lack historical records and reliance on static assumptions."

**Point 2.** Regarding LUC data: It is challenging, if not impossible, to validate the LUC over the past millennium. The "reliability assessment" of historical LUC data needs elaboration. How does this assessment validate the reconstructed data, given the absence of direct validation methods for pre-industrial periods? Clarify whether this

approach evaluates internal consistency, cross-references with alternative proxies (e.g., tax records), or quantifies uncertainty ranges. Please explicitly state what distinguishes the LUC dataset in this study from prior publications by He et al. Is the novelty in data synthesis, spatial resolution, or integration of new historical sources (e.g., tax records)?

Response: Thank you for this comment.

Reliability assessment has always been an unavoidable yet unverifiable aspect of historical land-use reconstructions, as the actual historical conditions cannot be fully known and can only be reconstructed using proxy data. Therefore, the reliability of such reconstructions is typically evaluated by examining the data sources, the rationality of the reconstruction methods, and the degree to which the results align with historical records, historical events, or similar datasets. In response to this issue, a dedicated subsection—2.3.2 Reliability assessment of long-term land-use change data—has been included in this manuscript. This section briefly outlines the above-mentioned aspects to indirectly demonstrate the reliability of the reconstructed data.

Relevant revisions can be found in lines 221–229, 254–258, and 275–277 of the manuscript, and have been marked in red font.

**Point 3.** Regarding carbon density assumptions: The assumption of static carbon densities over millennia is problematic. While the authors update current biomass and soil densities, pre-industrial carbon stocks likely shifted due to CO2 changes, climatic variability, ecological succession, and human management. Discuss how these dynamics might bias emission estimates and propose strategies to address this in future work (e.g., coupling with DGVM outputs). The carbon density updates in the current work only scratched the surface of the issue, by improving the densities of "current" times. In GCB2024, there are four book-keeping models used, why do you choose H&N or H&C model (I assumed, you did not specify)? Is it because of spatial resolution or any particular features that match well with your current data, like using LUC "state" instead of LUC "transition"? The other three seem to incorporate

dynamic carbon densities to some extent (for instance including DGVM biomass data), but also with higher spatial resolution that may not match the provincial level in this study. I would suggest clarifying the rationale in the Methods, AND further discussing the uncertainties in the Discussions. This is not to deemphasize this work, but to urge future improvements.

Response: Thank you very much for your comments. Your feedback is professional, rigorous, and highly valuable for the further revision of our manuscript. It also provides insightful directions for potential future improvements, and we sincerely appreciate it.

First, following your suggestion, we have further clarified the origin of the bookkeeping method used in our study and the rationale for selecting this model in the Methods section (Section 2.3.1). Please refer to lines 192–197 in the manuscript, which have been highlighted in red for easy identification. The revised excerpt is as follows:

"The bookkeeping method (a statistical model) proposed by Houghton and Castanho (2023) was employed to estimate the annual carbon emissions caused by land-use changes in China from 1000 to 2019. Due to data limitations, long-term historical land-use reconstructions in China are primarily constrained to land-use "states" (e.g., total cropland or forest area at national/provincial levels for specific years) rather than spatially explicit land-use transitions. This characteristic, combined with the provincial-level spatial resolution of our data, makes such reconstructions inherently compatible with the bookkeeping model adopted here (Houghton and Castanho, 2023)."

In Section 4.3 Uncertainty Analysis, we provide a detailed discussion on static versus dynamic carbon density, the potential uncertainties associated with using static carbon density values, and directions for future improvements. Specific revisions were made in lines 565–573 of the main text and have been marked in red font. The excerpt is as follows:

"Although modern soil carbon densities were moderately adjusted by incorporating

a large-scale soil sampling survey dataset from the post-1949 period in China, pre-industrial carbon stocks likely varied due to shifts in atmospheric $CO_2$ concentrations, climate fluctuations, ecological succession, and human land management. Vegetation and soil carbon densities were not static over the past millennium. Therefore, using static values to represent historical carbon densities may fail to capture temporal dynamics, thereby introducing uncertainties. Potential biases include overestimating human contributions if climate-driven increases in carbon density are ignored and overestimating modern carbon uptake if long-term baseline declines in carbon stocks are not included. Future studies should explore coupling DGVMs (e.g., LPJ or ORCHIDEE) to simulate combined impacts of historical climate, $CO_2$ levels, and human activities on carbon density."

**Point 4.** About uncertainty quantification: The current "uncertainty" section (4.3) primarily discusses limitations rather than quantifying uncertainties. Incorporate a robust quantitative analysis (e.g., Monte Carlo simulations) to assess how data gaps (e.g., historical LUC, carbon density variability) propagate into emission uncertainties. This will enhance the study's rigor and reproducibility. The 4.3 section is not technically an "uncertainty analysis", it is simply discussions of limitations and possible future work.

Response: Thank you for your insightful comments on the uncertainty quantification. We fully agree that the original Section 4.3 focused more on qualitative discussions of limitations and future work, rather than providing a rigorous quantitative uncertainty analysis. To address this, we have implemented the following key revisions.

(1) In the Methods section of the manuscript, we have added subsection 2.3.4, 'Uncertainty assessment,' which elaborates on how we utilized Monte Carlo simulations to assess the uncertainty in carbon emission results. For the full description, please refer to lines 351-362 of the main manuscript. An excerpt is provided below:

"To evaluate the uncertainty in estimating carbon emission fluxes, this study employed Monte Carlo simulations with 1000 iterations. The uncertainty primarily

stems from two key parameters: carbon density and land-use change area. For the carbon densities in the forest (aboveground, belowground, and soil) and grassland (aboveground, belowground, and soil) components, the mean and standard deviation were calculated based on input sample data. During the simulations, values for these densities were randomly sampled from normal distributions parameterized based on these statistics measures. Regarding the land-use change area, the original input value for the annual conversion area of each land-use type served as the mean for its sampling distribution, with the standard deviation set to 10% of this mean. Values were then randomly sampled from a normal distribution defined by these parameters in each iteration. Subsequently, in every iteration, the annual carbon emission flux was re-estimated using the parameters sampled in that specific iteration. After aggregating the results from all iterations, the minimum and maximum simulated carbon emission flux values for each year were used to define the uncertainty interval for that year's estimates."

(2) In subsection 4.3, 'Uncertainty analysis,' we plotted the Monte Carlo simulation results as Figure 10 and subsequently analyzed them. For full details, please refer to lines 540-551 of the main text. An excerpt is provided below:

"This study employed Monte Carlo simulations (1000 iterations) to systematically assess the uncertainty in annual carbon emission flux estimates (Fig.11). The results revealed that the average annual uncertainty interval, which was derived from the maximum and minimum simulated carbon emissions, was 18.75 Tg C. This interval exhibited significant interannual variation, ranging from a minimum of 3.77 Tg C to a maximum of 143.67 Tg C. Such variation indicates that the uncertainty in the estimation results increased in years characterized by substantial fluctuations in land-use change data. Overall, the Monte Carlo simulations effectively highlighted the impact of parameter uncertainty on carbon emission estimates and provided a quantitative basis for evaluating the credibility of the carbon flux results. To further constrain parameter variability, future efforts should focus on improving the resolution of measured carbon density data and the reliability of land-use data."

[Figure]

**Figure 10.** Uncertainty in annual carbon emissions from land-use change

**Minor points:**

**Point 5.** "China": this needs to be better defined in this study! You used the current mainland China as the country boundary, and merged the 30+ provinces into 25 regions. I understand your reasoning for compromising here, but you must make this crystal clear in the Abstract and Methods. In Fig. 1, you may also cite specific studies for each map for different dynasties.

Response: Thank you for this comment. I fully agree with your proposal. Revised.

Clear research scope and fundamental units can significantly enhance the manuscript's readability. I have explicitly defined our study area and provincial-level units in both the Abstract (Lines 21-23) and Methods section (Section 2.1 Study Area - Lines 112-115), with additional clarification on the sources of historical territorial and administrative boundaries data provided in Line 124.

For detailed revisions to the manuscript text, please refer to the revised draft where changes are highlighted in red font.

Thank you once again!

**Point 6.** L171: regarding the bookkeeping model, did you use the Houghton model, or simply used their structure and data? This can be made more explicit.

Response: Thank you. Revised.

The computational structure of Professor Richard A. Houghton's bookkeeping model is relatively simple, as shown in Equation 3. The distinctive feature of this method lies in its parameterization of disturbance-response curves. These curves define the long-term carbon release and sequestration patterns by different vegetation types and their associated soils following land-use conversions, which constitutes the core mechanism of carbon accounting. We obtained region-specific parameters for China from Professor Richard A. Houghton for implementing this carbon budget calculation.

In Section 2.3.1 (Bookkeeping Method), we have explicitly clarified this point by citing that the disturbance-response curves were sourced from Houghton and Castanho (2023). The modifications in the manuscript are located in Lines 211-212 and are highlighted in red font. Since the values of the disturbance-response curves are derived from existing literature rather than our own analysis, they have been compiled in Appendix Table B2.

**Point 7.** L200: "local expert and knowledge", delete "and"?

Response: Thank you. Revised.

Thank you once again for your help—not only did you improve the overall logic and readability of our work, but you also took the time to pay close attention to the details of the manuscript. We are especially grateful!

**Point 8.** L206: using tax records is a great idea, but how does this help this particular study? Any quantitative evidence?

Response: Thank you for this comment.

In this study, the cropland data covering dozens of time slices over the past millennium are primarily reconstructed based on historical tax records from successive Chinese dynasties. This approach is fundamentally different from the

global datasets and serves as strong support for the higher reliability of our data. It has been well-documented that land-use reconstructions based on region-specific historical records tend to be more accurate than large-scale simulations, especially in regions with rich documentary evidence, such as China.

To address the issue of data reliability, we have explicitly described the sources and general reconstruction processes of historical cropland, forest, and grassland data in Section 2.3.2 ("Reliability assessment of long-term land-use change data"). This section aims to show that the dataset used in our study is currently the only one in China that covers major land-use types over a long time span with high reliability. Therefore, we have deliberately dedicated substantial space in the manuscript to explain the basis and credibility of our data in detail, in order to enhance the confidence of reviewers and readers. (Line 219-279)

Regarding the suggestion to provide quantitative evidence, this manuscript focuses on the application of our reconstructed land-use datasets. The quantitative procedures of the data have been comprehensively presented in a series of previous publications by our team. These key references have been listed in Table 1 for readers and reviewers to trace and examine if needed.

Do you agree with our response? If you have any questions, please raise them again. We will continue to make targeted modifications in the second round. Thank you once again!

**Point 9.** L224: this is out of context, what exactly is "inverted S-shaped" relationship?
Response: Thank you for raising this question. I completely agree with you—without having read the cited reference, it is indeed difficult to understand what the "inverted S-shaped" relationship specifically refers to.

To improve readability, we have added an explanation of the "inverted S-shaped" relationship. The corresponding revision has been made in lines 254–258 of the main text, and the changes are marked in red. The revised content is as follows:

"The "inverted S-shaped" curve reflects the dynamic relationship between historical population size and deforestation. In the early stages, when the population is

relatively small, forest resources are plentiful and the rate of deforestation remains slow. As the population grows, deforestation accelerates rapidly, resulting in a significant loss of forest cover. Eventually, despite the population continuing to increase, the scarcity of remaining forests causes the rate of deforestation to slow down. "

**Point 10.** L243: cite the data used.

Response: Thank you for this comment. Revised (Lines 275–277).

**Point 11.** L270: Fig 3. The whole study is at the provincial level, why do you use gridded data here in the map? What data are they? What criteria did you use to separate west vs. east of China, or to draw the "forest-grassland boundary"? Over 1000 years, did this boundary move at all?

Response: Thank you for this comment.

First, the historical land-use data used in this study is a composite of multiple datasets covering three main categories: cropland, forest, and grassland. Taking into account factors such as spatial-temporal resolution and data operability, we chose the provincial level as the basic unit of calculation.

Figure 3 illustrates the historical land-use change transition rules. Here, we use two gridded maps of forest and grassland data to better convey the information in spatial form. These maps help to visually distinguish forest-dominated and grassland-dominated regions. The two maps are derived from: *He, F., Yang, F., and Wang, Y., 2025. Reconstructing forest and grassland cover changes in China over the past millennium. Science China Earth Sciences, 68(1): 94–110.* They represent the spatial distribution of forests and grasslands in AD 1000 at a 10 km resolution.

The basis for dividing China into western and eastern regions follows: *Su, D.X. The regional distribution and productivity structure of the Chinese grassland resources. Acta Agrestia Sinica, 1994, 2: 71–77.* This study was primarily used to distinguish

grassland types, dividing the country into regions primarily comprising the northern temperate zone and the Tibetan Plateau in western China, and nonzonal secondary grasslands in eastern China. Later, the following two studies built upon this division and further classified the "northern temperate zone and the Tibetan Plateau in western China" as zonal grasslands, defining it as the grassland-dominated western region, while the eastern part was historically forest-dominated: *Yang, F., He, F., and Li, S., 2020. Spatially explicit reconstruction of anthropogenic grassland cover change in China from 1700 to 2000. Land, 9(8): 270. He, F., Yang, F., and Wang, Y., 2025. Reconstructing forest and grassland cover changes in China over the past millennium. Science China Earth Sciences, 68(1): 94–110.*

This is the origin of the regional division used in this study.

Finally, we acknowledge that this regional division does not represent a strict boundary. As you rightly pointed out in your comments, the boundary between forests and grasslands may have shifted over the past millennium. However, both our study and the aforementioned literature use this broad division to determine whether a given provincial unit was generally forest-dominated or grassland-dominated. Therefore, even though the boundary may have changed over time, the impact on determining provincial-level affiliation is minimal.

The above explains the details of the data we used, as well as all the background we could think of in response to your comments. We hope this clarifies your concerns. If you have any further questions, we would be happy to provide additional explanation and make further revisions in the next round of responses. Once again, thank you for your thoughtful and constructive comments, which have been instrumental in improving the academic quality of our manuscript.

**Point 12.** L290: the whole argument about shifting ag. in China is not strongly supported. This happens in Africa and S. America, but it is not as common in China. What does recent remote sensing suggest? It would be more convincing to show some

direct evidence than simply claim "…has been recorded extensively in Chinese historical documents."

Response: Thank you for this comment.

Shifting agriculture is an ancient form of agricultural production that was historically widespread. Today, it is mainly found in lowland and hilly areas of tropical rainforest regions, such as those mentioned by the reviewer—Africa and South America. In China, however, this form of cultivation has virtually disappeared since the founding of the People's Republic, as it is a highly extensive and inefficient mode of production. Currently, we have not found any studies that detect this type of agriculture in China using remote sensing data. Therefore, from a data perspective, it is difficult to obtain empirical support for its presence today.

However, from a different angle, because shifting agriculture is such an old production method, if we extend the timeline to several hundred or even a thousand years and broaden the source materials to include historical documents and related scholarly works, we can easily find references to shifting agriculture. In China, it is known as "slash-and-burn" farming. There are numerous historical records about it, although, to our knowledge, no studies besides our own provide detailed quantitative estimates of its extent.

Shifting agriculture is frequently mentioned in historical records and is closely tied to key historical events. Since the mid-Qing Dynasty, the implementation of the "head tax into land tax" (摊丁入亩) policy by the Qing government greatly encouraged population growth. Many scholars describe this as a population explosion. During this period, many displaced people—often referred to as "shelter people" (棚民)—were forced by economic hardship to migrate into previously undeveloped mountainous areas to clear land. In the process, large areas of forest were destroyed, but in fact, very little of this land was converted into permanent farmland. Most of it was temporary cultivation.

Based on this historical background and the records, combined with the reconstructed forest and cropland datasets used in our study, we quantified the area of forest converted to other land. The trend of this change corresponds closely to the

historical timeline of "shelter people" expanding into mountainous areas. Therefore, in Figure 5b, we present this data and infer that the primary land-use process responsible was shifting agriculture.

**Point 13.** Fig. 4-5, did you compare the LUC data with other sources, like LUH2, to examine the differences and causes?

Response: Thank you for this comment.

In Figures 4 and 5, we did not compare our reconstruction results with global datasets such as LUH2, primarily for the following reasons:

To our knowledge, LUH2's long-term historical land-use data largely derives from the HYDE dataset. HYDE is a globally recognized land-use dataset that spans the entire Holocene and includes the historical period covered in our study for China.

Given its widespread application, scholars have long conducted studies to assess the reliability of HYDE data in China. For example:

For cropland in Northeast China over the past 300 years:
Li, B., Fang, X., Ye, Y., & Zhang, X., 2010. Regional accuracy assessment of global land-use datasets: A case study of Northeast China. Science China Earth Sciences, 40(08): 1048–1059.
For traditional agricultural regions:
He, F.N., Li, S.C., Zhang, X.Z., Ge, Q.S., & Dai, J.H., 2013. Comparisons of cropland area from multiple datasets over the past 300 years in the traditional cultivated region of China. Journal of Geographical Sciences, 23(6): 978–990.
For cropland across China over the past millennium:
Zhao, C., He, F., Yang, F., & Li, S., 2022. Uncertainties of global historical land use scenarios in past-millennium cropland reconstruction in China. Quaternary International, 641(20): 87–96.

There are also regional evaluations:
Qinghai–Tibet Plateau:
*Li, S.C., He, F.N., Zhang, X.Z., & Zhou, T.Y., 2019. Evaluation of global historical land use scenarios based on regional datasets on the Qinghai–Tibet Area. Science of the Total Environment, 657: 1615–1628.*
Xinjiang:
*Li, M., He, F., Zhao, C., & Yang, F., 2022. Evaluation of global historical cropland datasets with regional historical evidence and remotely sensed satellite data*

*from the Xinjiang Area of China. Remote Sensing, 14(17): 4226.*

There are also evaluations of global dataset accuracy for forest and grassland in China:

For forest:

*Yang, F., He, F.N., Li, M.J., & Li, S.C., 2020. Evaluating the reliability of global historical land use scenarios for forest data in China. Journal of Geographical Sciences, 30(7): 1083–1094.*

For pasture:

*He, F., Li, S.C., Yang, F., & Li, M.J., 2018. Evaluating the accuracy of Chinese pasture data in global historical land use datasets. Science China Earth Sciences, 61(11): 1685–1696.*

Overall, extensive research has already been conducted to evaluate global datasets such as HYDE, as well as others like PJ, KK10, and SAGE, with a focus on the Chinese region. In Section 2.3.2, "Reliability assessment of long-term land-use change data," we briefly summarize and cite key literature related to the evaluation of global datasets for cropland, forest, and grassland in China, for the benefit of reviewers and readers.

In light of the substantial body of existing work, we decided not to include a direct comparison with global datasets in this study.

**Point 14.** Fig. 5: please clarify the meaning of secondary axis. In (a), does the y-axis suggest "changes" or absolute area? Same for (b), absolute or relative area? For (c) and (d), what does the pie suggest, 1000-yr cumulative or annual?? Please be more specific.

Response: Thank you for this comment. Revised.

In lines 374–378, we have added new statements to further clarify what the y-axes in panels (a) and (b) represent and their units. We have also clearly explained the meaning of the pie charts in panels (c) and (d). The revisions are marked in red font.

"(a) Cropland, forest, and grassland areas (absolute values), in units of 106 hectares. (b) Proportions of four land-use types in each period, with all remaining terrestrial cover—excluding the reconstructed cropland, forest, and grassland—classified as

other land. (c) Cumulative carbon emissions from land-use changes across different carbon pools. (d) Cumulative carbon emissions from different land-use transitions. In (c) and (d), the two pie charts represent the shares of different carbon pools and land-use transitions in the cumulative carbon emissions over the millennium, respectively."

**Point 15.** Fig. 6: Does the negative biomass value show carbon sink? Specify in the caption.

Response: Thank you for this comment. Revised. (Line 408)

**Point 16.** L435: Table3, this table is a summary not "comparison. These estimates cover different time period, so the emissions would be different. No surprise here. Could you compare them across the same or similar time, and include results from this study?

Response: Thank you for this comment.

The studies listed in Table 3 have differences in time periods, and some of the differences in their results are due to this. Therefore, as described in lines 427-428 of this manuscript, they are strictly speaking not comparable.

Given that the data from the studies listed in Table 3 are not open access, we cannot modify their data to obtain consistent time periods across all these studies. However, our data represents annual carbon budgets, and we extracted overlapping time periods from both this study and the existing studies. We compared our results (the last column of Table 3) with those from the existing studies for the same time periods and analyzed the reasons for the differences in the manuscript.

Since Table 3 is quite long, with 7 columns, and our results are in the last column, you may not have noticed this column. The new Table 3 is the result of revisions made in response to the suggestions of another reviewer.

**Table 3.** Comparison of existing long-term carbon emission estimation results caused by land-use change in China

| Region | Land use type | Method | Time period | Previous study (Pg C) | Reference | This study (Pg C) |
|---|---|---|---|---|---|---|
| China | Cropland, Forest, Grassland | Bookkeeping model (Early version) | 1700–1980 | 9.05 | Yang et al. (2023) | 15.17 |
| China | Cropland | Bookkeeping model (Early version) | 1661–1980 | 3.78 | Yang et al. (2019) | 16.13 |
| China | Cropland, Forest | Bookkeeping model (Early version) | 1700–1949 | 6.18 | Ge et al. (2008) | 11.87 |
| Northeast China (Heilongjiang, Jilin, and Liaoning) | Cropland | Bookkeeping model (Early version) | 1680–1980 | 1.45 | Li et al. (2014) | 3.33 |
| Global | Cropland, Forest, Grassland, Other land | Bookkeeping model (Latest version) | 1850–2019 | 7.36 | Houghton and Castanho (2023) | 7.72 |
| China | Cropland, Forest | Land ecosystem model | 1900–1980 | 6.90 | Yu et al. (2022) | 7.07 |
| China | Cropland, Forest | Land ecosystem model | 1980–2019 | 8.90 | Yu et al. (2022) | 2.25 |
| China | Cropland, Forest, Grassland, Other land | Bookkeeping model (Latest version) | 1000-2019 | 19.61 | This study | |

**Point 17.** L476: Is this required? It seems odd with a data availability statement in the middle.

Response: Thank you for this comment.

Since ESSD is primarily a data-focused journal and we are submitting a data description article, according to the journal's template, the Data Availability section is required, and the data must be shared on an open access platform.

**Point 18.** Appendix A and B: is the information in these tables used in this study? Or do they simply support previous work on LUC data.

Response: Thank you for this comment.

The information in Appendices A and B is used in this study and serves as important supporting data for the results presented in the manuscript. Due to limitations in length, structure, and logical flow, we placed this content in the appendices.

Specifically, Table A1 provides detailed sources for the second and third national

land survey bulletins; Table B1 lists soil series in China; Table B2 presents the disturbance response curve parameters; Figures B1–B4 show the sample points for soil carbon density.

Thank you again for your thorough and professional feedback on our manuscript. We truly appreciate your time and expertise. We are open to any additional questions or suggestions in the next round of review and are committed to further improving the paper.

**Response to Reviewer 2 Comments**

**Manuscript Title:** Annual carbon emissions from land-use change in China from 1000 to 2019

**Recommendation:** Major Revision

**General Comments**

This manuscript presents an ambitious reconstruction of carbon emissions from land-use change (LUC) in China over the past millennium. Using a provincial-scale bookkeeping model and extensive historical records, the authors estimate annual LUC emissions from 1000 to 2019, supported by updated carbon density datasets. The work contributes a long-term dataset of carbon fluxes that could support both paleoclimate-carbon research and national greenhouse gas (GHG) accounting.

However, the manuscript falls short in clearly articulating its scientific motivation, ensuring methodological transparency, and validating the results. Of particular concern is the assumption that vegetation and soil carbon densities remain constant over 1000 years, which critically weakens the interpretability of the results. In addition, the complete absence of quantitative uncertainty analysis and comparison with existing datasets limits the credibility and broader applicability of the findings.

I recommend major revision to address the following concerns.

Response: Thank you very much for taking the time to review our manuscript. We appreciate your recognition of our topic and work.

In response to the questions and concerns you raised, we have made substantial revisions to the manuscript. These include adding more detailed descriptions of our methodology to enhance its transparency and quantifying the estimation uncertainties using Monte Carlo simulations. Please find our point-by-point responses below. All changes in the manuscript have been marked in red font.

Thank you once again for your valuable feedback.

**Major Comments**

**Point 1. The scientific rationale, challenges, and innovation of a millennial-scale reconstruction are insufficiently articulated**

While reconstructing LUC-related carbon emissions since AD 1000 is conceptually valuable, the manuscript does not sufficiently explain:

- Why this timescale is necessary for understanding anthropogenic impacts on the carbon cycle;

- What methodological or conceptual challenges exist in performing such long-term reconstructions;

- How this study specifically overcomes those challenges or improves upon prior work.

- The novelty of the study must be made more explicit. For example:

- How does this reconstruction differ from studies that begin in 1700 or 1850?

- What new historical sources, spatial refinements, or analytical methods are introduced?

**Recommendation:** Include a comparative table summarizing key differences between this study and prior LUC carbon emission reconstructions (e.g., in time span, resolution, input data, model approach, and validation).

Response: Thank you for this comment. We have added substantial content to the Introduction section to elaborate on why we conducted a study over such a long period, why the past 1000 years are of particular importance to China, and the implications of this work for China and other nations. For details, please see lines 70-84 of the main text, which are excerpted as follows:

"China has a vast territory and a long history of land use, making it an important contributor to global terrestrial carbon dynamics caused by anthropogenic land-use change and land management. Although most global and regional studies on land-use change focus on the post-industrial era or the past three centuries, China's intensive and extensive land-use activities date back at least a millennium, thus representing a

unique historical trajectory (He et al., 2025, 2023). From approximately AD 1000 (coinciding with the Northern Song Dynasty), ecological degradation in China showed a marked rise. This degradation was manifested through multiple pathways: accelerated erosion on the Loess Plateau, recurrent floods in the lower Yellow River Basin, large-scale lake siltation and disappearance in northern China, and progressive soil erosion coupled with natural vegetation loss in the southern hill regions (Wu et al., 2020; Chen et al., 2012). Such millennial-scale land-use transitions would have generated substantial carbon emissions, particularly from deforestation. However, the relatively stable pre-industrial global $CO_2$ concentrations likely obscured these regionally significant anthropogenic carbon fluxes because localized emissions in areas such as China could have been offset by concurrent carbon sinks elsewhere. Additionally, the full trajectory or specific stages of historical land-use change in China can serve as a "historical analogue" for other developing countries. For many countries and regions, systematically revealing the processes and mechanisms of land-use change and associated carbon emissions—driven by long-term population growth and policy shifts—can help overcome the limitations associated with a lack historical records and reliance on static assumptions."

In our updated Table 3, building upon the original comparison with previous studies, we have further clarified key information such as the time span, input land-use types, and model employed, in order to highlight the critical differences between the various studies. For details, please see lines 499-501 of the main text, which the updated table is excerpted below:

**Table 3.** Comparison of existing long-term carbon emission estimation results caused by land-use change in China

| Region | Land use type | Method | Time period | Previous study (Pg C) | Reference | This study (Pg C) |
|--------|--------------|--------|-------------|----------------------|-----------|-------------------|
| China | Cropland, Forest, Grassland | Bookkeeping model (Early version) | 1700–1980 | 9.05 | Yang et al. (2023) | 15.17 |
| China | Cropland | Bookkeeping model (Early version) | 1661–1980 | 3.78 | Yang et al. (2019) | 16.13 |
| China | Cropland, Forest | Bookkeeping model | 1700–1949 | 6.18 | Ge et al. (2008) | 11.87 |

(Early version)

| Northeast China (Heilongjiang, Jilin, and Liaoning) | Cropland | Bookkeeping model (Early version) | 1680–1980 | 1.45 | Li et al. (2014) | 3.33 |
|---|---|---|---|---|---|---|
| Global | Cropland, Forest, Grassland, Other land | Bookkeeping model (Latest version) | 1850–2019 | 7.36 | Houghton and Castanho (2023) | 7.72 |
| China | Cropland, Forest | Land ecosystem model | 1900–1980 | 6.90 | Yu et al. (2022) | 7.07 |
| China | Cropland, Forest | Land ecosystem model | 1980–2019 | 8.90 | Yu et al. (2022) | 2.25 |
| China | Cropland, Forest, Grassland, Other land | Bookkeeping model (Latest version) | 1000-2019 | 19.61 | This study | |

Note: Bookkeeping model (Early version) refers to the initial model developed by Houghton and Hackler (2003).

Bookkeeping model (Latest version) refers to the most recently updated model by Houghton and Castanho (2023).

**Point 2. The assumption of static carbon densities undermines the long-term credibility of the reconstruction**

A central concern lies in the assumption that vegetation and soil carbon densities remain constant over the entire 1000-year period. While this may be a necessary simplification given limited historical data, it significantly weakens the scientific credibility of the estimated carbon fluxes—especially for earlier centuries.

Carbon densities are not time-invariant: they are influenced by changes in climate, atmospheric $CO_2$, ecosystem succession, species composition, and land-use intensity. Assuming present-day carbon densities for all historical periods' risks introducing systemic bias in the emission estimates, particularly during major climatic or socio-ecological transitions (e.g., the Little Ice Age, or the Qing Dynasty agricultural expansion).

This assumption is particularly problematic because the technical challenge—and scientific value—of millennial-scale carbon accounting lies precisely in addressing such temporal variability. If a key driver like carbon density is held static, the study risks becoming an arithmetic exercise rather than a meaningful reconstruction, and its findings may not substantially differ from earlier studies based on heuristic

extrapolation.

**Recommendation:**

- Clearly state which carbon density datasets are used, and how they are applied across the time domain;

- Acknowledge the limitations of assuming static carbon densities, and discuss the potential magnitude and direction of bias this may introduce;

- Propose a pathway for future work, such as incorporating carbon density outputs from process-based vegetation models (e.g., DGVMs) or paleoecologically reconstructions;

- Emphasize that confronting this assumption is essential for enhancing the interpretive value and novelty of the study.

Response: I strongly agree with the issue you've raised.

The bookkeeping model used in this study is primarily driven by land-use change data and utilizes observed vegetation and soil carbon density data and specific disturbance response curves for each land-use transition type. As this method excludes the influence of unchanged land-use types and environmental changes, such as carbon dioxide concentrations and climate change, it quantifies direct anthropogenic fluxes and ignores carbon fluxes driven by environmental changes (Dorgeist et al., 2024; Houghton and Castanho, 2023). However, at the same time, as you mentioned, it overlooks the long-term impacts of environmental changes (e.g., climate, $CO_2$ concentration) on carbon stocks.

In our discussion section, we have added a discussion on the limitations of static carbon densities and pathways for future improvement in lines 565-573. Here is the excerpt:

"Additionally, the spatiotemporal variability of basic carbon density values can influence the accuracy of the estimates. In this study, carbon density is addressed using a "present-day-for-past" substitution method. Although modern soil carbon densities were moderately adjusted by incorporating a large-scale soil sampling survey dataset from the post-1949 period in China, pre-industrial carbon stocks likely varied due to shifts in atmospheric $CO_2$ concentrations, climate fluctuations,

ecological succession, and human land management. Vegetation and soil carbon densities were not static over the past millennium. Therefore, using static values to represent historical carbon densities may fail to capture temporal dynamics, thereby introducing uncertainties. Potential biases include overestimating human contributions if climate-driven increases in carbon density are ignored and overestimating modern carbon uptake if long-term baseline declines in carbon stocks are not included. Future studies should explore coupling DGVMs (e.g., LPJ or ORCHIDEE) to simulate combined impacts of historical climate, $CO_2$ levels, and human activities on carbon density."

**Point 3. Modern-era results lack validation and comparison with existing datasets**

The study spans from 1000 to 2019, but observational and model-constrained datasets are available primarily for the post-1950 period. Yet the manuscript does not compare its estimates to:

- National or global LUC carbon emission inventories (e.g., FAO, Houghton, LUH2);
- Remote sensing-based datasets of forest loss or biomass change;
- Process-based models such as DGVMs or spatially explicit bookkeeping models (e.g., BLUE).

These comparisons are essential for establishing the reliability of the methodology and providing a reference point for earlier trends.

**Recommendation:** Include a table comparing national and/or provincial LUC emissions from this study with at least 3–4 widely used datasets over overlapping time periods, accompanied by discussion on differences and their likely causes.

Response: Thank you for this comment.

In Figure 10, we present a comparison between the reconstruction results of this study and those of other relevant studies, particularly for the period since 1950 (see line 523 in the main text for details). Furthermore, we have provided a detailed

discussion on the discrepancies among different research findings and their primary causes, as detailed in lines 510-548 and excerpted below.

"The estimates from the other three bookkeeping models aligned more closely with the trends in the DGVM estimates, which were markedly different from our estimations. This discrepancy primarily stems from two key aspects. First, DGVM estimates often account for the "loss of additional sink capacity". This concept refers to the diminished carbon absorption that occurs when the land-use type of a parcel of land that could have absorbed more carbon dioxide under current environmental conditions if left in its original natural state (e.g., as a forest) is altered by human activities (e.g., conversion to cropland), thereby reducing its actual carbon dioxide uptake. This "reduction in absorbed amount" constitutes the loss of additional sink capacity. Gasser et al. (2020) revealed that the inclusion or exclusion of loss of additional sink capacity leads to significant differences in estimated values. Second, disparities in land-use change forcing data represent another significant factor contributing to divergent estimates among different models. DGVM estimates are typically driven by long-term global land-use datasets, such as LUH2 (Obermeier et al., 2024; Friedlingstein et al., 2019; Hansis et al., 2015). Thus, these models that differ due to the inclusion of loss of additional sink capacity and the use of varying land-use change data tend to significantly overestimate the carbon emission flux from land-use changes relative to the results of this study."

[Figure]

**Figure 10.** Chinese historical land-use change-induced carbon emission flux estimated by different methods.

"Additionally, the estimates from this study differed considerably from national report-based data (e.g., NGHGIs and FAOSTAT) (Fig. 10) (Obermeier et al., 2024). The core difference between NGHGIs and bookkeeping models in land-use change carbon flux estimation lies in the carbon accounting boundary, especially regarding the attribution of indirect fluxes on managed land (Gidden et al., 2023; He et al., 2024). NGHGIs tend to consider all carbon fluxes on managed land (including both direct fluxes and indirect fluxes triggered by environmental changes) as anthropogenic contributions. In contrast, bookkeeping models primarily account for direct fluxes generated by direct human activities but exclude indirect fluxes, which are considered natural ecosystem responses, from anthropogenic inventories of land-use change. The fact that national reports specifically account for afforestation and ecological restoration projects with high carbon removal potential might also influence the results. The most direct example is the similarity between our estimated carbon emissions (1900–1980) and the results of Yu et al. (2022) (Table 3) because of the lack of significant or widespread land management or engineering projects in China during this period. However, the estimates for 1980–2019 differed greatly

because land management practices during this period had a substantial impact. As revealed by Yue et al. (2024), land management has played a crucial role in China's land-carbon balance since 1980."

**Point 4. Absence of quantitative uncertainty analysis limits credibility**

Section 4.3 is labeled "Uncertainty Analysis" but provides only qualitative reflections on limitations. This is insufficient given the range of assumptions, spatial heterogeneity, and sparse data for earlier centuries.

**Recommendation:**

- Include a quantitative uncertainty analysis (e.g., via Monte Carlo simulations or scenario analysis);

- Report confidence intervals or uncertainty bounds for cumulative and decadal emissions;

- Indicate how uncertainty varies across time, especially between well-documented (post-1950) and poorly constrained (pre-1700) periods.

Response: Thank you for this comment. In response to your comments, we have estimated the uncertainty associated with our carbon emission results using Monte Carlo simulations. The details are provided in lines 351-362 and 540-548. An excerpt is provided below:

2.3.4 Uncertainty assessment

To evaluate the uncertainty in estimating carbon emission fluxes, this study employed Monte Carlo simulations with 1000 iterations. The uncertainty primarily stems from two key parameters: carbon density and land-use change area. For the carbon densities in the forest (aboveground, belowground, and soil) and grassland (aboveground, belowground, and soil) components, the mean and standard deviation were calculated based on input sample data. During the simulations, values for these densities were randomly sampled from normal distributions parameterized based on these statistics measures. Regarding the land-use change area, the original input value

for the annual conversion area of each land-use type served as the mean for its sampling distribution, with the standard deviation set to 10% of this mean. Values were then randomly sampled from a normal distribution defined by these parameters in each iteration. Subsequently, in every iteration, the annual carbon emission flux was re-estimated using the parameters sampled in that specific iteration. After aggregating the results from all iterations, the minimum and maximum simulated carbon emission flux values for each year were used to define the uncertainty interval for that year's estimates.

4.3 Uncertainty analysis

This study employed Monte Carlo simulations (1000 iterations) to systematically assess the uncertainty in annual carbon emission flux estimates (Fig.11). The results revealed that the average annual uncertainty interval, which was derived from the maximum and minimum simulated carbon emissions, was 18.75 Tg C. This interval exhibited significant interannual variation, ranging from a minimum of 3.77 Tg C to a maximum of 143.67 Tg C. Such variation indicates that the uncertainty in the estimation results increased in years characterized by substantial fluctuations in land-use change data. Overall, the Monte Carlo simulations effectively highlighted the impact of parameter uncertainty on carbon emission estimates and provided a quantitative basis for evaluating the credibility of the carbon flux results. To further constrain parameter variability, future efforts should focus on improving the resolution of measured carbon density data and the reliability of land-use data."

[Figure]

**Figure 11.** Uncertainty in annual carbon emissions from land use change

**Minor Comments**

**Point 5. Clarify the bookkeeping framework**

Indicate whether this is a "statistical bookkeeping model" or incorporates spatially explicit components to distinguish it from models such as BLUE or OSCAR.

Response: Thank you for this comment. Revised.

Per your suggestion, we have now clarified in the Methods section that this is a statistical model (see lines 192-197 of the manuscript, highlighted in red). The excerpt is as follows:

"The bookkeeping method (a statistical model) proposed by Houghton and Castanho (2023) was employed to estimate the annual carbon emissions caused by land-use changes in China from 1000 to 2019. Due to data limitations, long-term historical land-use reconstructions in China are primarily constrained to land-use "states" (e.g., total cropland or forest area at national/provincial levels for specific years) rather than spatially explicit land-use transitions. This characteristic, combined with the provincial-level spatial resolution of our data, makes such reconstructions inherently compatible with the bookkeeping model adopted here (Houghton and Castanho, 2023)."

**Point 6. Add a conceptual model diagram**

A schematic showing the flow from land-use data → transition → response curve → carbon flux would clarify the modeling approach.

Response: Thank you for this comment. Revised.

We thank you for this valuable suggestion. Accordingly, we have constructed Figure 2 (Framework for calculating annual carbon emissions based on the bookkeeping model). This has been added to the revised manuscript, with the details provided on lines 182-183 and 188-189 (highlighted in red for clarity).

[Figure]

**Figure 2.** Framework for calculating annual carbon emissions based on the bookkeeping model.

**Point 7. Improve Table 2**

Include the number of observations or sample density for each province to help readers assess data quality.

Response: Thank you for this comment. Revised.

As suggested, we added sample sizes for forest and grassland carbon density (both vegetation and soil) for each province to Table 2. The changes are highlighted in red in lines 179-180.

**Table 2.** Provincial vegetation and soil carbon density data

| Province/region | Forest (Mg/ha) | | Grassland (Mg/ha) | |
| --- | --- | --- | --- | --- |
| | SOCD | VCD | SOCD | VCD |
| Chuan-Yu | 98.83 (n=132) | 55.96 (n=159) | 143.09 (n=50) | 1.25 (n=142) |
| Inner Mongolia | 69.38 (n=179) | 41.60 (n=263) | 88.79 (n=119) | 5.77 (n=416) |
| Liaoning | 91.13 (n=70) | 44.74 (n=43) | 77.71 (n=35) | 3.32 (n=25) |
| Jilin | 95.09 (n=57) | 73.85 (n=39) | 67.09 (n=30) | 3.06 (n=24) |
| Heilongjiang | 145.45 (n=91) | 64.63 (n=114) | 93.58 (n=28) | 2.98 (n=22) |
| Gan-Ning | 99.44 (n=88) | 36.80 (n=57) | 54.66 (n=236) | 3.80 (n=159) |
| Qinghai | 75.87 (n=20) | 30.54 (n=36) | 108.60 (n=249) | 6.45 (n=385) |
| Xinjiang | 64.32 (n=22) | 25.59 (n=42) | 93.97 (n=119) | 4.09 (n=91) |
| Xizang | 129.33 (n=35) | 82.43 (n=20) | 58.89 (n=167) | 4.20 (n=291) |
| Jing-Jin-Ji | 75.39 (n=104) | 43.83 (n=117) | 88.32 (n=53) | 7.61 (n=19) |
| Shanxi | 59.98 (n=65) | 40.63 (n=66) | 56.13 (n=115) | 8.77 (n=71) |
| Shaanxi | 74.29 (n=174) | 29.78 (n=101) | 64.75 (n=110) | 4.03 (n=45) |
| Shandong | 60.42 (n=30) | 42.29 (n=26) | / | / |
| Henan | 59.03 (n=17) | 42.41 (n=24) | / | / |
| Anhui | 86.90 (n=44) | 63.06 (n=57) | / | / |
| Hu-Ning | 91.79 (n=31) | 37.63 (n=27) | / | / |
| Hunan | 92.60 (n=174) | 51.94 (n=42) | / | / |
| Hubei | 139.57 (n=63) | 48.00 (n=20) | / | / |
| Jiangxi | 93.29 (n=162) | 50.81 (n=44) | / | / |
| Zhejiang | 115.13 (n=69) | 54.14 (n=35) | / | / |
| Fujian | 117.71 (n=114) | 58.80 (n=72) | / | / |
| Yue-Qiong | 111.36 (n=233) | 37.33 (n=92) | / | / |
| Guangxi | 108.26 (n=156) | 55.87 (n=105) | 99.32 (n=17) | / |
| Yunnan | 105.84 (n=110) | 76.26 (n=67) | 100.52 (n=14) | / |
| Guizhou | 129.37 (n=64) | 50.31 (n=29) | 284.18 (n=35) | / |

SOCD refers to soil organic carbon density, VCD refers to vegetation carbon density.

**Point 8. Clarify carbon density preprocessing**

Indicate whether carbon density values were standardized (e.g., by reference depth) and whether outliers were removed.

Response: Thank you.

The selection of carbon density data points and the calculation of provincial-level carbon densities were subject to specific filtering and processing. Notably, for soil carbon density, data points with a profile depth of at least 100 cm were chosen, and their soil carbon density was calculated for a 100 cm depth (detailed in manuscript lines 162-164).

In general, the data for above-ground, below-ground, and soil carbon densities for each province were normally distributed (see Appendix Figs. B2–B4). The arithmetic mean was used to calculate the provincial-level average carbon density. For provinces with exceptionally high or low values, the median was used to represent the central tendency and to minimize the influence of outliers (see lines 174-176).

**Point 9. Enhance regional time-series presentation (e.g., Fig. 7)**

Add temporal trends for individual regions, not just cumulative bar plots.

Response: Thank you for this comment.

Based on the comments and suggestions of the three reviewers, we have added a conceptual model diagram and an uncertainty analysis chart based on Monte Carlo simulations to the original nine figures in the main text. There are now a total of 11 figures in the main text. Considering the large number of figures, we have decided to place the chart showing the trend of carbon emissions over time for each province in Appendix C as Figure C1. This will make it easier for reviewers and readers to understand the details of carbon emission changes in each province.

**Appendix C**

[Figure]

**Figure C1.** Cumulative carbon emissions from land-use changes at the provincial level. Arrows indicate the turning points from carbon sources to carbon sinks, with numbers representing the corresponding years of the turning points.

**Point 10. Clarify the role of Appendices A and B**

State whether the sources listed in the appendices were used directly in this study or referenced for historical context only.

Response: Thank you for this comment.

The information in Appendices A and B is used in this study and serves as important supporting data for the results presented in the manuscript. Due to limitations in length, structure, and logical flow, we placed this content in the appendices.

Specifically, Table A1 provides detailed sources for the second and third national land survey bulletins; Table B1 lists soil series in China; Table B2 presents the disturbance response curve parameters; Figures B1–B4 show the sample points for soil carbon density.

**Point 11. Ensure consistent terminology**

Maintain consistent use of terms like "carbon sink" vs. "carbon sequestration."

Response: Thank you for this comment. Revised.

Thank you for your suggestion. To avoid unnecessary ambiguity, we have carefully reviewed the entire manuscript and replaced all instances of the word "sequestration." For details, please see lines 440-442, 512, and 571, which are marked in red font.

**Point 12. Improve figure quality**

Enhance the resolution of Figures 5–8 and define all abbreviations in figure captions (e.g., "Yue-Qiong").

Response: Thank you. Revised.

We apologize that the figures were not clear enough during your review, which may have been due to low initial resolution or issues during the Word-to-PDF conversion process. In this revision, we have re-inserted higher-resolution figures into the Word manuscript. However, since the conversion to PDF is handled by the editorial office, we do not have control over this step and are unsure to what extent the conversion process might affect the final image resolution.

If the figures are still not clear enough during your review, we ask for your understanding. Should our manuscript be accepted, we will upload high-resolution image files separately to ensure the clarity of the figures in the final publication. Thank you again for your understanding.

Additionally, we have added definitions for all abbreviations in the captions of the relevant figures. For details, please see Figure 9 and lines 448-450.

**Point 13.** Refine the title for clarity

Consider including terms such as "provincial reconstruction" or "bookkeeping-based estimate" to better reflect the methodological approach.

Response: Thank you for this comment.

We, the authors, have had extensive discussions regarding your suggestion about revising the title. We agree that the original title's strengths are its conciseness, clarity,

and broad appeal, while your proposed change would highlight the methodological and data-driven contributions, which would help attract a more specific audience. This is indeed a trade-off between "conciseness and clarity" and "richness of information."

We seriously considered adding these terms to the title. However, after our discussions, we still lean towards keeping the original title for two main reasons. First, we believe the current concise title—"Annual carbon emissions from land-use change in China from 1000 to 2019"—most effectively communicates the core findings to a broader audience, including policymakers and scientists from other disciplines. Second, we consider the primary contribution of this study to be the millennium-scale emissions dataset itself, and a title focused on the results best reflects this contribution.

To ensure that our methods and data characteristics are immediately apparent to readers, we have revised the abstract and methods section to explicitly highlight the key terms "provincial reconstruction" and "bookkeeping-based estimate."

Thank you again for your valuable guidance.

In summary, thank you once again for providing so many valuable comments and suggestions on our manuscript. We have fully absorbed them and have made revisions to the best of our ability. Your feedback has led us to re-examine the shortcomings in the structure and expression of the original manuscript, significantly improving its readability and scientific rigor.

We are very grateful for this valuable opportunity to engage with you. There may still be areas in the manuscript that are not entirely satisfactory, and we welcome you to point them out in the next round of review. We will strive to better understand your comments and suggestions and work to further improve the manuscript.

Thank you again!

**Response to Reviewer 3 Comments**

This manuscript presents a millennial-scale reconstruction of carbon emissions from land-use change in China using a bookkeeping model approach. While the study addresses an important research gap and provides valuable historical context for understanding China's carbon budget, there are several major concerns that must be addressed before this work is suitable for publication.

Response: Thank you very much for taking the time to review our manuscript and for your positive and encouraging comments. We have carefully considered the four main areas for revision or questions you raised. We have made detailed modifications and responses to each opinion or suggestion, and these changes have been marked in red font in the main text. Thank you again for your hard work; your comments have significantly improved the scientific quality of our manuscript.

**Point 1.** The conversion rules in Figure 3 appear somewhat arbitrary. I recommend testing the uncertainty in your transition matrix calculations. While your rule-based priority system is clear, how would results differ if you used an area-weighted approach instead? For example, allocating transitions proportionally based on the relative magnitude of area changes between different biomes rather than using predetermined priorities. This uncertainty analysis would be valuable given the millennium-long timeframe of your study, where even small methodological differences could compound into significant variations in results.

Response: Thank you for this comment. We apologize if our previous explanation of the land-use transition rules in Figure 4 was not detailed enough and caused confusion. We have further clarified this section in the revised manuscript, and these revisions are marked in red font. For details, please see lines 305-326.

Please allow me to briefly explain.

Firstly, the conversion rules are determined based on the attributes of the published

data used, which is a prerequisite for establishing the land use transition rules in this study. Specifically, when reconstructing historical grassland data in western China, it reflects the occupation of grassland due to the reclamation of cropland in history. In eastern China, historical grasslands mainly consist of secondary grasslands resulting from the secondary succession of deforested lands. The reconstruction rules for historical grassland data are the basis for formulating grassland-related land use conversion rules in this study.

After the land use transition rules related to grassland were established, whether it was the conversion of forest to cropland or forest to other land, historically, the essence was deforestation for reclamation. After deforestation, if the land could be cultivated for a long period, it was converted to cropland. If it became temporary cropland due to reasons such as loss of fertility, it is defined as other land in this study. According to Table B2 in the appendix, in the bookkeeping model used in this study, the disturbance response curves for the conversion of forest to cropland and forest to other land are identical. Therefore, once the land use conversion rules related to grassland are established, regardless of whether we use our set priorities or other methods (such as area weighting) to handle forest-related land use conversions, the final carbon emission calculation results will not be affected by the specific classification of forest conversion into cropland or other land.

The excerpt is as follows:

"First, the conversion rules were determined based on the attributes of the published data used, which was a prerequisite for establishing the land-use transition rules in this study. The land-use change data revealed the changes in grassland area and their conversion relationships were the most clearly defined. The reconstruction rules for historical grassland data formed the basis of the grassland-related land-use conversion rules in this study. Specifically, when reconstructing historical grassland data in western China, the data reflect the occupation of grassland due to the reclamation of cropland in history (He et al., 2024). Therefore, for western China, where grassland ecosystems dominate, changes in grassland areas primarily reflect the encroachment of croplands, and the conversion between grassland and cropland was

determined first based on changes in grassland area (Fig. 4). Second, the reduction in forest area was prioritized for conversion to cropland, followed by conversion to other land. In eastern China, where forest ecosystems are predominant, historical grasslands mainly consisted of secondary grasslands because of the secondary succession of deforested lands (He et al., 2024). Hence, in eastern provinces dominated by forest ecosystems, the conversion between grassland and forest can be similarly determined based on changes in the grassland area. The remaining forest area was then prioritized for conversion to cropland, followed by conversion to other land. Based on these rules, we calculated the annual land-use change rates in China from 1000 to 2019.

Historical conversion of forest to cropland or forest to other land was primarily performed for land reclamation, and if the deforested land supported cultivation over a long period, it was converted to cropland. For cropland that failed to support cultivation due to reasons such as a loss of fertility, it was defined as other land in this study. According to Table B2 in the appendix, in the bookkeeping model used in this study, the disturbance response curves for the conversion of forest to cropland and forest to other land were identical. Therefore, once the land-use conversion rules related to grassland were established, regardless of whether the set priorities or other methods (such as area weighting) were used to handle forest-related land-use conversions, the final carbon emission calculation results were not be affected by the specific classification of forest conversion into cropland or other land."

**Point 2.** The authors state that "this study updated and improved the land-use change data, carbon density data, and disturbance response curves," but upon careful reading, it appears they did not actually update or improve the disturbance response curves themselves. Rather, they simply adopted the data from Houghton and Castanho (2023) without modification. To avoid misleading readers, I suggest the authors clarify that they utilized the most recently published parameters from the literature rather than implying they developed improvements to the response curve themselves.

Response: Thank you!

I completely agree with your opinion. Yes, we directly used the latest published disturbance-response curve from Houghton and Castanho (2023). Several statements in the original manuscript regarding this curve might have been misleading or ambiguous for readers and reviewers. Therefore, in the revised manuscript, we have amended the relevant descriptions to clarify that we directly used the latest published disturbance-response curve from Houghton and Castanho (2023) without any further modifications. For details, please see lines 25-27, 105-106, 475-476, and 587-590 in the main text, highlighted in red font.

**Point 3.** I also noticed that the bookkeeping model used in this study does not account for wood harvest pools, which is understandable given that it would require reconstructing additional historical wood harvest data. However, this limitation should be explicitly stated in the methodology section. The authors should clarify this omission and briefly discuss its potential implications for carbon flux estimates, especially since wood harvest can be a significant component of land-use change emissions in forested regions of China.

Response: Yes, I completely agree with your point.

We have clarified this in the methods section of the revised manuscript: Due to data limitations, this accounting does not consider carbon emissions from wood harvest. For details, please see lines 186-187 of the manuscript.

Simultaneously, in the discussion section, lines 574-580 (marked in red font), we re-emphasized that the current accounting does not include wood harvest, as well as the potential impacts arising from the omission of wood harvest. By integrating existing relevant literature, reference values for carbon emissions from wood harvest were provided. The excerpt is as follows:

"We reiterate that the carbon emission accounting method in the present study does not include wood harvesting. Considering that wood harvesting represents a significant historical source of anthropogenic emissions, the absence of these data may lead to a certain degree of underestimation in the corresponding carbon emission

fluxes. Fortunately, Houghton and Castanho (2023) estimated China's long-term carbon emissions from wood harvesting and found values of 5 Tg C yr$^{-1}$ for 2011–2020, approximately 20–30 Tg C yr$^{-1}$ around the 1950s, approximately 5–10 Tg C yr$^{-1}$ in the 1900s, and less than 5 Tg C yr$^{-1}$ values before 1900. These estimates can serve as a reference when regional long-term reconstructed data on wood harvesting and their corresponding carbon emission estimates are unavailable."

**Point 4.** The explanation of differences between NGHGI and bookkeeping estimates should focus on carbon accounting boundaries rather than restoration projects (Gidden et al., 2023, Nature; He et al., 2024, Nature Communications). For DGVMs vs. bookkeeping models, note that DGVMs include the loss of additional sink capacity, leading to higher emission estimates, alongside differences in LUC forcing data (Gasser et al., 2020, Biogeosciences). I suggest the authors provide a more systematic discussion to avoid misleading readers about these differences.

Response: Thank you very much for your comments. Your opinions are accurate and highly valuable, helping us to revise the relevant content in section 4.2 Comparison with previous estimates, strengthen the comparison between different results, and make the relevant explanations more scientific and persuasive. Thank you again.

We have collected and consulted relevant literature and content. Incorporating your suggestions, we have revised this part, detailed in lines 510-538, and marked the changes in red font. We are not entirely certain if our revisions have fully addressed and alleviated your concerns regarding our manuscript. If there are any further questions or if our explanations are not adequate, please raise them in the next round of evaluation. We will further strive to understand your opinions to improve the manuscript. The relevant revisions are excerpted as follows:

"The estimates from the other three bookkeeping models aligned more closely with the trends in the DGVM estimates, which were markedly different from our estimations. This discrepancy primarily stems from two key aspects. First, DGVM estimates often account for the "loss of additional sink capacity". This concept refers to the diminished carbon absorption that occurs when the land-use type of a parcel of

land that could have absorbed more carbon dioxide under current environmental conditions if left in its original natural state (e.g., as a forest) is altered by human activities (e.g., conversion to cropland), thereby reducing its actual carbon dioxide uptake. This "reduction in absorbed amount" constitutes the loss of additional sink capacity. Gasser et al. (2020) revealed that the inclusion or exclusion of loss of additional sink capacity leads to significant differences in estimated values. Second, disparities in land-use change forcing data represent another significant factor contributing to divergent estimates among different models. DGVM estimates are typically driven by long-term global land-use datasets, such as LUH2 (Obermeier et al., 2024; Friedlingstein et al., 2019; Hansis et al., 2015). Thus, these models that differ due to the inclusion of loss of additional sink capacity and the use of varying land-use change data tend to significantly overestimate the carbon emission flux from land-use changes relative to the results of this study.

Additionally, the estimates from this study differed considerably from national report-based data (e.g., NGHGIs and FAOSTAT) (Fig. 10) (Obermeier et al., 2024). The core difference between NGHGIs and bookkeeping models in land-use change carbon flux estimation lies in the carbon accounting boundary, especially regarding the attribution of indirect fluxes on managed land (Gidden et al., 2023; He et al., 2024). NGHGIs tend to consider all carbon fluxes on managed land (including both direct fluxes and indirect fluxes triggered by environmental changes) as anthropogenic contributions. In contrast, bookkeeping models primarily account for direct fluxes generated by direct human activities but exclude indirect fluxes, which are considered natural ecosystem responses, from anthropogenic inventories of land-use change. The fact that national reports specifically account for afforestation and ecological restoration projects with high carbon removal potential might also influence the results. The most direct example is the similarity between our estimated carbon emissions (1900–1980) and the results of Yu et al. (2022) (Table 3) because of the lack of significant or widespread land management or engineering projects in China during this period. However, the estimates for 1980–2019 differed greatly because land management practices during this period had a substantial impact. As

revealed by Yue et al. (2024), land management has played a crucial role in China's land-carbon balance since 1980."

We are very grateful for your numerous valuable and constructive suggestions. In accordance with your comments, we have diligently revised the manuscript. We remain open to any new questions you may have in the subsequent round of review and are committed to further improving the paper. Thank you for your consideration.

---

## Author Response (AR2)

**Response to Reviewer 1 Comments**

The authors have addressed most of previous comments, and the manuscript has been substantially improved. Below are a few additional suggestions for consideration:

Response: Thank you very much for your valuable suggestions for revision. Following your advice, we have carefully and meticulously revised the manuscript. The main revisions include: 1) Based on your insightful comment, we have highlighted the methodological contribution of our carbon density analysis. 2) We have clarified the reason why the data table only includes land types that are directly relevant to our study's methodology. 3) We have updated the comparison data in the manuscript to reflect the latest research, including the Global Carbon Budget (GCB 2024), re-verified and explained the consistency between our data and the timing of the carbon sink transition, and, per your suggestion, added a discussion on the conceptual differences between the models.

We believe these revisions have further enhanced the rigor and clarity of the manuscript. Below are our point-by-point responses to your comments.

**Point 1.** While carbon density is a key component of the study, its novelty may be overstated in the context of a millennium-scale analysis. Given the inherent uncertainties over such extended timescales, the accuracy of current density mapping could be largely obscured. It may be more appropriately framed as a methodological contribution rather than a novel finding.

Response: Thank you for this comment.

We fully agree with your view that carbon density should be framed as a methodological contribution rather than a novel finding. Accordingly, we have carefully reviewed the entire manuscript and revised the wording in the Abstract (lines 25–27) and Methods section (lines 212–213) to reflect this perspective. Specifically, we now clarify that, based on previously published datasets, our contribution lies in the integration and compilation of existing data.

**Point 2.** Table 2, how about data for other land types, e.g. cropland, other land? They are not all zeros, correct?

Response: Thank you for this comment.

Please allow us to provide a brief clarification. In the literature and books on carbon density that we collected since 1980, the records primarily cover three major land categories: forest, grassland, and cropland, with only sporadic data available for other land types. For the purposes of this study, we employed the bookkeeping method (a statistical model) proposed by Houghton and Castanho (2023), which incorporates carbon density values for forests and grasslands. Therefore, in Table 2 we present only the data relevant to our analysis, and the carbon density values for other land-use types from the original sources are not displayed here.

**Point 3.** It is becoming clear that over the past few decades, LUC in China is shifting from a net CO2 source to a net sink (or at least neutral). This is evidenced by the latest GCB 2024 (Fig. 7b in Friedlingstein 2024; https://doi.org/10.5194/essd-17-965-2025) and a recent Nature Climate Change study (Fig. 1 and 3a in Zhu 2024; http://dx.doi.org/10.1038/s41558-025-02296-z). In particular, the Houghton model (H&C) in GCB 2024 shows negative fluxes for China since the 1960s (data is here: https://globalcarbonbudgetdata.org/latest-data.html). While your Fig. 10 seems to show negative fluxes since 1970-80s for "this study", Table 3 presents a positive 2.25 Pg C for 1980-2019. Any clarification on this discrepancy? Additionally, why do you choose older GCBs instead of the latest one? Please clarify or justify. I would encourage an updated discussion on "4.2 Comparison with previous estimates". In the discussions, please also note that the concept of LUC emissions differs between bookkeeping vs. DGVM models.

Response: Thank you for this comment. Please note that the previous Figure 10 is now referred to as Figure 11 in this revised document because of structural changes and the insertion of new figures.

1. Update to Figure 11

We have updated Figure 11 by replacing the comparison dataset "GCB 2019" with the latest "GCB 2024." In addition, we have incorporated the carbon emission data for China (1981–2020) from Zhu et al. (2025) and added corresponding descriptions in the text (see lines 557–559, 562-565).

2. Verification of carbon flux estimates

We have re-checked the carbon flux estimates in this study, which are also available on Zenodo (https://doi.org/10.5281/zenodo.14557386, 2025). Our results confirm that the carbon flux becomes negative (indicating a carbon sink) after 1980, rather than after 1970. Therefore, the curve shown in Figure 11 and the total value of +2.25 Pg C for 1980–2019 in Table 3 are not in conflict. The possible confusion may be due to the long time span of the curve (1700–2020) in Figure 11, which makes it difficult to visually distinguish between 1970 and 1980. You may refer to our openly available dataset for the exact values.

3. Clarification of LUC emission concepts

In response to your suggestion to note that the concept of LUC emissions differs between bookkeeping models and DGVMs, we have emphasized this distinction in Section 4.2 (lines 522–525).

We sincerely thank you for your two rounds of careful, professional, and timely reviews, which have led to a substantial improvement in the quality of our manuscript and a significant enhancement of its scientific rigor. Should you have any questions regarding our responses above, or additional comments and suggestions on the manuscript, please feel free to raise them during the third round of review, and we will make further careful revisions and improvements accordingly. Once again, we truly appreciate your dedicated efforts.

**Response to Reviewer 2 Comments**

Thank you for addressing most of my previous concerns. However, my main concerns in this second round focus on land-use transition matrix construction, Figure 10, and Table 3, which require substantial improvements for methodological validation and systematic comparison. The current approach lacks rigor in several key areas that undermine the study's credibility.

Response: We sincerely thank you for your profound and constructive feedback during the second round of review. We concur that methodological validation, figure clarity, and systematic comparison are vital to our study. In response to your primary concerns regarding the construction of the land-use transition matrix, Figure 10 (Labeled as Figure 11 in the current version), and Table 3, we have undertaken substantial revisions to strengthen the study's scientific rigor.

In detail, our revisions include: (1) performing a comprehensive robustness check by implementing an alternative "area-weighted" allocation method, which validates the reliability of our land-use reconstruction; (2) updating the data sources and caption for Figure 10 (Labeled as Figure 11 in the current version) and supplementing the appendix with detailed definitions and origins of the datasets used; (3) clarifying the distinct inclusion criteria for Figure 10 (Labeled as Figure 11 in the current version) and Table 3, which serve different comparative aims, and adding an in-text explanation to prevent any potential confusion; and (4) refining the positioning of our results relative to existing literature based on newly included data. All changes are marked in the revised manuscript using the "track changes" feature, and specific line numbers are referenced in our detailed responses.

We are confident that these systematic improvements have thoroughly addressed the issues you raised, thereby enhancing the methodological soundness and credibility of our findings. Your expert guidance has been invaluable in improving the quality of our work, and we thank you once again.

**Point 1.** Land-use transition matrix construction lacks methodological validation

The priority-based rules for allocating land-use transitions from aggregate area data introduce substantial uncertainty that remains unaddressed. The authors' justification that forest-to-cropland and forest-to-other-land have identical response curves misses the point─the issue is not the final carbon calculation but the arbitrary nature of the allocation rules themselves. Without testing alternative methods (area-weighted, probabilistic) or demonstrating robustness across different rule sets, the reliability of the entire reconstruction is questionable. This methodological uncertainty compounds over the millennium timescale and undermines confidence in the results.

Response: Thank you for this comment.

We thank the reviewer for their valuable and insightful comments, which have been instrumental in improving our manuscript. Your specific concern regarding the land use transfer allocation rule and its effect on the reliability of our findings is a critical point, which we have sought to address thoroughly.

Following your recommendation, we performed a comprehensive robustness check. We adopted an area-weighted allocation method—an objective, unbiased alternative—to recalculate the land use transfer matrices for the entire 1000–2019 study period. The rationale for this test has been added to the "Methods" section (revised manuscript, 2.3.3 Uncertainty assessment, lines 361–367).

Our systematic comparison of the original "priority-based" results with the new "area-weighted" results reveals high consistency. First, the absolute differences between the two sets of annual land use transfer matrices are minimal, as illustrated in a new heatmap (Appendix Fig C2), confirming the stability of our land use change reconstruction. Second, when the new land use data were used to re-estimate carbon emissions, the resulting trends and turning points were nearly identical to our original findings, with all new values falling completely within our previously reported uncertainty bounds (Figure 11).

**Appendix C**

[Figure]

**Figure C2** Differences in annual land-use transitions between the priority-based and area-weighted allocation methods. F-C denotes the conversion between Forest and Cropland, F-G represents Forest-Grassland conversion, F-O represents Forest-Other land conversion, and G-C represents Grassland-Cropland conversion. A positive difference indicates that the priority-based result is lower than the area-weighted result, and vice versa.

[Figure]

**Figure 11.** Comparison of carbon fluxes from land-use change using different calculation methods, with uncertainty assessment.

To integrate these results, we have added a new paragraph to the "Result" section

(3.3 Uncertainty and Sensitivity Analysis, lines 467–483), supported by the new figures. This analysis demonstrates that the core conclusions of our study are robust against the choice of allocation methodology. We are confident that this additional analysis directly addresses your primary concern and provides a more solid foundation for our conclusions.

Thank you again for your time and expertise. Your feedback has significantly improved the rigor of our study. We await your further review.

**Point 2.** Figure 10 legend and data sources lack essential details. -Data sources for all model results need clear citation.-Update to latest GCB2024 data instead of GCB2019. Figure 10 presents Gasser (2020), Hansis et al.(2015), and Houghton (2023) alongside GCB2019 without explaining that these three studies are the component models underlying GCB estimates. This may mislead readers about the independence of these approaches. -NGHGI.DB and NGHGI.DB.corrected are undefined. What specific corrections were applied and how? -Justify TRENDYv8 selection (if due to additional scenarios isolating LASC effects, state this explicitly)

Response: Thank you for this comment.

1) We have updated the data from the Global Carbon Budget (GCB) 2019 to the latest version, GCB 2024.

2) We have amended the caption for Figure 10 (Labeled as Figure 11 in the current version) to clarify that the GCB estimate is not an independent data point. The caption now states that "The GCB estimate synthesizes the findings of Gasser (2020), Hansis et al. (2015), and Houghton (2023)." (lines 580–581)

3) The NGHGI.DB, NGHGI.DB.corrected, and TRENDYv8 datasets used in our study are all adopted from Obermeier et al. (2024) (*Obermeier, W. A., Schwingshackl, C., Bastos, A., Conchedda, G., Gasser, T., Grassi, G., Houghton, R. A., Tubiello, F. N., Sitch, S., and Pongratz, J.: Country-level estimates of gross and net carbon fluxes from land use, land-use change and forestry, Earth System Science Data, 16, 605-645, 10.5194/essd-16-605-2024, 2024.*). Detailed descriptions of these datasets, including their specific definitions, distinctions, and the correction procedures applied, are provided in Appendix Table D1. Specifically, the

TRENDYv8 dataset allows for the isolation of direct LULUCF impacts through the comparison of different scenarios (e.g., with and without land-use change). (Lines 551-555)

**Appendix D**

**Table D1.** Definitions and methodologies for the NGHGI.DB, NGHGI.DB.corrected, and TRENDYv8 datasets.

| Dataset | Source and Description | Core Processing and Application |
|---|---|---|
| NGHGI.DB | National Greenhouse Gas Inventory (NGHGI) data reported by countries to the UNFCCC, with gap-filling applied. | Serves as the baseline data representing officially reported carbon fluxes from managed land. |
| NGHGI.DB.corrected | A corrected version of NGHGI.DB, adjusted to align with model-estimated anthropogenic fluxes. | Carbon fluxes from natural and indirect effects (e.g., $CO_2$ fertilization, climate change) are subtracted. This component is estimated by TRENDYv11 models under a scenario without land-use change. |
| TRENDYv8 | Ensemble mean of nine Dynamic Global Vegetation Models (DGVMs). | Used to isolate the direct impacts of LULUCF by comparing results from different scenarios (e.g., with and without land-use change). |

**Point 3.** Table 3 is incomplete and inconsistent with Figure 10. The current Table 3 lacks systematic collection of comparable studies and shows inconsistency with Figure 10 content. While Figure 10 includes multiple DGVM studies for China, Table 3 only presents Yu et al. as the sole DGVM representative without justification for this selective inclusion, and other bookkeeping model results as well as NGHGI/FAOSTAT shown in Figure 10 are not included in Table 3.

Response: Thank you for your valuable feedback. Your observation regarding the discrepancy between Table 3 and Figure 10 is very insightful, and you have correctly identified an important point that requires clarification. The two formats were intentionally designed for different comparative purposes, which we are happy to explain here. We will also add a note to the manuscript to clarify this for readers.

Table 3 and Figure 10 are not merely different presentations of the same data; they are complementary comparisons targeting two categories of studies with distinct data attributes. Our criteria for inclusion were as follows:

- Figure 10 Inclusion Criterion: This figure compiles results from all studies for which we could obtain annual-resolution time-series data. This format allows for a direct, year-by-year visual comparison of dynamic trends. Consequently, it includes multiple Dynamic Global Vegetation Models (DGVMs), bookkeeping models driven by remote sensing or global datasets (e.g., HYDE/LUH), and annual data from NGHGI/FAOSTAT.

- Table 3 Inclusion Criterion: This table focuses on key studies that did not provide publicly available annual time-series data but reported total or average estimates over specific periods. These studies, particularly the foundational works by Yang et al. (2023), Yang et al. (2019), Li et al. (2014), and Ge et al. (2008), are crucial for understanding the historical carbon budget of China based on century-scale land-use reconstructions. We present their core findings in the table to facilitate a direct comparison of their period-aggregated results.

Based on these principles, we also included two specific studies in Table 3 for the following reasons:

- Houghton and Castanho (2023): This study was included because we used a bookkeeping model identical to theirs, whereas several other studies in the table used earlier versions of this model. Including it provides a direct methodological benchmark for comparison.

- Yu et al. (2022): Although this is a DGVM study with time-series data (as shown in Fig. 10), we also included it in Table 3 because it shares a critical attribute with the other studies in the table: it uses a century-scale, historically reconstructed land-use dataset specifically for the China region as input. This distinguishes it from most studies in Fig. 10 that rely on global datasets, making its input data and resulting estimates highly comparable to the other works listed in Table 3.

We hope this explanation clarifies the design rationale for Figure 10 and Table 3. To prevent any confusion for future readers, we will add a concise explanation to the manuscript (Lines 525-529). Thank you again for your insightful comments and

suggestions.

**Point 4.** The statement "estimates in this study fall within the range of existing model estimates at an intermediate level" is incorrect. Figure 10 shows your results are among the most negative values post-2000 (excluding NGHGI data due to different definitional boundaries).

Response: Thank you for your insightful feedback. You are correct, and we agree that in the previous version of Figure 10 (Labeled as Figure 11 in the current version), our post-2000 estimates were among the most negative values (the lowest, excluding NGHGI and FAOSTAT data), not at an "intermediate level." Your observation was spot-on.

In this revision, per the suggestion of Reviewer 1, we have incorporated the recent study by Zhu et al. (2025) (Zhu, Y., Xia, X., Canadell J., Piao, S., Lu, X., Mishra, U., Wang, X., Yuan, W., and Qin, Z.: China's Carbon Sinks from Land-Use Change Underestimated. Nature Climate Change, 15, 4: 428-35, 2025.) into Figure 10 (Labeled as Figure 11 in the current version). The findings from this study indicate an even larger carbon sink between 1992 and 2020. Consequently, when compared against this new dataset, our estimates are no longer the lowest in the range.

We have revised the relevant text in the manuscript to accurately reflect this updated comparison. Please see Lines 557-559 for the specific changes.

**Response to Reviewer 3 Comments**

General Comments. This paper estimates carbon emissions from land-use change in China over the past millennium, which is highly relevant and aligns well with ESSD's scope. The use of historically reconstructed land-use datasets—based on China's unique archival records rather than proxy indicators like population—is particularly valuable. While the authors have addressed some concerns in prior revisions, several issues remain regarding paper structure, integration of historical and modern land-use data, spatial resolution of historical datasets, and practical applications of the results. Additionally, some of my comments overlap with previous reviewers' feedback; I urge the authors to prioritize these shared concerns.

Response: Thank you for your insightful and constructive comments, which have been crucial for improving our manuscript. We have thoroughly revised the paper based on your suggestions, focusing on three key areas: strengthening our scientific rationale, improving the logical structure, and clarifying the practical applications of our dataset.

   Key revisions include restructuring the manuscript for a clearer separation of the Methods, Results, and Discussion sections. We have also expanded our methodology to better explain the integration of historical and modern data and have introduced a new sensitivity analysis to quantify the impact of our core assumption of static carbon densities. Finally, to highlight the dataset's value as encouraged by ESSD, we added a dedicated section on its applications in climate research and policy assessment.

   We believe these changes substantially strengthen the paper. Below are our detailed responses to each of your points. Thank you again for your time and expertise.

Specific Comments.

**Point 1.** -Data Integration Issues. Clarify how reconstructed data (e.g., cropland from tax records) align with survey-based statistics (e.g., the Second and Third National Land Surveys). Land-use definitions evolved between surveys (e.g., the Second

National Land Survey [2009] and Third National Land Survey [2019]). Discuss potential errors introduced by these definitional shifts.

Response: Thank you for your valuable feedback. We have carefully considered the issue of inconsistent statistical calibers in our data integration and would like to provide a detailed explanation here.

Your point is crucial. In processing the land-use data from 1980 to the present, we faced two primary options: one being the annual-resolution land use/cover datasets derived from remote sensing interpretation, and the other being the national-level, survey-based statistical data, namely the Second and Third National Land Surveys. We chose the latter primarily because the reconstructed historical data we used is more closely aligned and compatible with the national survey data in terms of its sources, methodologies, and nature (e.g., statistics and mapping based on administrative units). We believe that linking datasets of a similar nature helps maintain consistency in the long-term trends and mechanisms.

We fully agree with your observation that even between the Second and Third National Land Surveys, the land classification standards have evolved and differ, which poses challenges for direct data linkage. To minimize the errors introduced by these definitional discrepancies, we performed specific harmonization and adjustments for the most sensitive land class in our carbon flux model: forest. Specifically, the original research literature for the historical forest data explicitly states that its definition of 'forest' is conceptually closest to the 'closed forest land' sub-category in current classification standards. Therefore, when linking with modern data, we did not use the total area of the primary 'forest land' category. Instead, we precisely extracted the data for the 'closed forest land' sub-category from both the Second and Third surveys to ensure maximum definitional consistency with the historical reconstructed data.

Despite these efforts, we acknowledge that the definitional evolution of other land classes (e.g., cropland, grassland) across different survey periods, along with the inherent discrepancies in statistical calibers between the reconstructed and survey-based data, remains a source of uncertainty in this study. These differences

will inevitably affect the accuracy of the final carbon budget estimation. We will explicitly address this point in the discussion section of our paper and identify it as an important area for future research, which could be advanced through data fusion or the development of more optimal classification conversion algorithms (see lines 608-619 of the manuscript).

**Point 2.** -Line 141. are considered highly credible. Cite references for this statement.

Response: Thank you for this comment. Revised.

Thank you for your valuable feedback. Regarding your comment on the supporting evidence for our statement that the National Land Survey data "are considered highly credible" (Line 141), we have carefully considered the point and revised the manuscript. We fully agree that providing a clear justification for the reliability of this key dataset is essential. In our revision, we reflected on the best way to establish this credibility. For official census data of this nature—organized by the highest state administrative body and mobilizing national resources—its authority and reliability are typically accepted as a consensus or a benchmark in the academic community. It serves as a foundational starting point for research, rather than a debatable claim requiring repeated justification. Therefore, we concluded that the most rigorous and direct method to demonstrate its credibility is not by citing an indirect evaluation from another study, but by elucidating the rigorous nature of the data production process itself. Based on this reasoning, we have revised the original, more general statement and replaced it with a specific description of the survey process. Please see the revised text in lines 142-143.

**Point 3.** -Line 145. The text describes vegetation carbon density first but later details soil carbon density before vegetation. Revise for logical flow.

Response: Thank you for this comment. Revised. (Line 212)

**Point 4.** -Temporal Stability of Carbon Densities. Soil/vegetation carbon densities are treated as static over the millennium. The Discussion notes this limitation, but quantify its impact: Would assuming stable densities overestimate or underestimate emissions?

Response: Thank you for your insightful suggestion to quantify the impact of assuming static carbon density over time. We fully agree that this is a critical scientific issue. Accordingly, we have designed and completed a sensitivity analysis to assess the potential effects of this assumption on our estimation results. Our analysis is based on the posited systematic differences between historical and modern carbon pools in vegetation and soil. We hypothesized that historical vegetation carbon density was likely systematically lower than modern levels, a premise primarily based on the limited 'CO$_2$ fertilization effect' under significantly lower pre-industrial atmospheric CO$_2$ concentrations (approx. 280 ppm vs. >420 ppm today). Conversely, we posited that historical soil carbon density was likely higher than the modern average, mainly due to less intensive anthropogenic disturbance, which allowed soil organic carbon pools in extensive ecosystems to remain closer to a state of natural saturation.

  Based on this rationale, we designed a scenario assuming that historical vegetation carbon density was 20% lower and soil carbon density was 20% higher than modern values. After recalculating based on this scenario, we conducted an in-depth analysis of the annual differences between the new and original estimates, revealing distinct temporal patterns. During the carbon source periods, which constitute the vast majority of the study period (approx. 982 years), the new estimates were consistently lower than the original values, with a mean annual difference of approximately -2 Tg/yr, indicating a smaller and more stable range of deviation. In contrast, during the few years identified as carbon sink periods (approx. 37 years), influenced by the intense land-use change during those times, the discrepancy between the two estimates showed greater uncertainty and volatility, with differences ranging from -5 to +11 Tg/yr. This period-segmented analysis indicates that our original methodology may lead to a systematic overestimation of carbon fluxes, and that the uncertainty of

this estimation is particularly pronounced during carbon sink periods. We believe this new, more in-depth analysis substantively addresses your concerns and significantly enhances the rigor of our paper's discussion on uncertainties (the detailed revisions in the main text can be found in lines 640-649).

Thank you again for your valuable feedback, which has greatly improved the quality of our research.

**Point 5.** -Line 164–165. Briefly summarize the framework of the transfer function for bulk density estimation. Technical details can remain in cited sources.

Response: Thank you very much for your valuable feedback. In our study, for sample points that lacked measured bulk density data, we employed an empirical transfer function established and validated in Yang et al. (2007) for estimation. This function is based on the significant negative correlation between soil organic matter (SOM) content and bulk density, a relationship that has been widely confirmed in soil science studies. According to that paper, the specific formula for estimation is:

$$BD = 0.29 + 1.2033 \times e^{-0.0775 \times SOM}$$

where BD is the bulk density to be estimated (unit: g/cm³ ) and SOM is the percentage of organic matter content in the corresponding soil layer (%). The model's goodness-of-fit ($r^2$) is 0.81 (p<0.01), which indicates a high degree of reliability. To make our research methods clearer and more transparent, we have followed your suggestion and added this specific formula and explanation to the methods section of the revised manuscript. Please see lines 232-235 for the detailed revisions in the main text.

Thank you again for your guidance.

**Point 6.** -Table 2. Add province/region codes (e.g., "No.1" for Jing-Jin-Ji) to align with Figure 1.

Response: Thank you for this comment.

Following your recommendation, we have revised Table 2 to ensure its consistency with Figure 1. We have added a new column, "Code," and reordered the rows to align with the regional numbering (No. 1, No. 2, ...) presented in Figure 1. This modification greatly improves the coherence between the table and the figure. The revision can be found on lines 247-250.

**Point 7.** -Figure 2. Explain color schemes in the flowchart. In other words, What is the meaning of each color in the flowchart?

Response: We thank the reviewer for their constructive feedback and agree that the color scheme in Figure 2 required clarification.

To address this, we have added an explanatory sentence to the figure caption, defining the module represented by each color (please see lines 258-260 of the revised manuscript). We are confident that this revision improves the clarity of our research framework and the overall readability of the figure.

**Point 8.** -Line 195. Spatially explicit cropland/forest/grassland data exist (e.g., SCES literature). Justify why provincial-scale aggregation was used instead.

Response: Thank you for your valuable feedback. The question you raised regarding our choice to use provincial-scale summary data is a critical methodological consideration of our study. Our decision to use the provincial scale as the primary analytical unit was made deliberately, based on a comprehensive assessment of multiple factors, including data reliability, time-series consistency, and scale matching with key parameters (i.e., carbon density). The specific reasons are detailed below:

1. Data Reliability and Uncertainty:

While the spatially explicit (i.e., gridded) long-term Land Use/Cover Change (LUCC) datasets you mentioned do offer a high-resolution perspective, they are typically generated through techniques such as spatial downscaling or data fusion. This process inevitably introduces uncertainties stemming from model assumptions.

The reliability of such datasets is particularly challenging for historical periods. In contrast, provincial-level statistics are aggregated from long-term, relatively standardized administrative reporting systems. Although they have a lower spatial resolution, they represent the fundamental unit for historical land-use records in China and possess a high degree of reliability.

2. Time-Series Consistency and Continuity:

This study aims to construct a long-term inventory of carbon emissions from land use, for which data continuity and consistency are paramount. The data for the later period of our study were linked and calibrated with the "Second National Land Survey (2009)" and the "Third National Land Survey (2019)". Currently, the authoritative and fully open-access versions of these two surveys, which share a consistent statistical scope, are primarily available at the provincial summary level. Adopting the provincial scale thus maximizes the consistency of data sources and standards throughout the entire study period.

3. Scale Matching with Key Parameters (Carbon Density):

The core of our research is to estimate carbon emissions driven by land-use change, which requires coupling land-use area data with corresponding carbon density data. The carbon density datasets we employed—including data from the Second National Soil Survey of China (1979–1985), the China land ecosystem carbon density dataset by Xu et al. (2019), and the more recent Chinese Soil Series (since 2008)—are all fundamentally derived from field surveys and measurements at sample points. These sample points are spatially discrete and do not provide complete grid coverage. Therefore, aggregating both the land-use data and the carbon density sample data to the provincial scale is a more methodologically robust approach that ensures better compatibility between them.

In summary, while spatially explicit data offers advantages in displaying spatial patterns, we chose the provincial scale as it is the most appropriate and robust strategy for our research objectives. These objectives prioritize the construction of a long-term time series, the assurance of data reliability and consistency, and the scientifically sound coupling of land-use data with carbon density data derived from sample points.

This decision was a trade-off made after carefully evaluating the strengths and weaknesses of different data sources to ensure the accuracy and reliability of our final estimates.

**Point 9.** -Line 263. Provide references for "SAGE" and "PJ" datasets at first mention.

Response: Thank you for this comment. Revised.

For details, please see lines 192-193 of the revised manuscript.

**Point 10.** -Section 2.3.2 (Reliability Assessment). This section describes data sources rather than evaluating reliability. Move it to Section 2.2.1 (Land-use Data) for cohesion. If you put it in Methods, then the reliability results belong in Results/Discussion.

Response: We thank the reviewer for this insightful comment. We agree that the content previously in Section 2.3.2 was a description of data sources and was misplaced. Accordingly, we have relocated this text to Section 2.2 Data sources to improve the manuscript's structure. This change is reflected in lines 147-210 of the revised version.

**Point 11.** -Line 500. Clarify key improvements in the latest bookkeeping model, including the updated disturbance-response curves, refined land-use transition rules? Or anything else?

Response: Thank you for this comment.

First, the new model refines the simulation of wood harvest to better reflect actual harvesting practices. The adjustments include correcting the post-harvest carbon allocation between 'wood products' and 'slash' to align with FAOSTAT data , and reducing the harvest intensity in secondary forests. This latter change necessitates simulating a larger harvested area to meet the same wood volume, thereby increasing

the gross carbon sink from forest recovery.

Second, a key update is the proposal and simulation of alternative interpretations for 'Forest Conversion to Other Land' (FCO), a phenomenon observed in many tropical countries where the net loss of forest area exceeds the net gain in agricultural land. In contrast to the Houghton and Nassikas (2017) study which assumed a single pathway ('recovering forest') , this new research explores additional land-use conversion rules, including statistical error, 'shifting cultivation,' and 'degraded land,' to assess their distinct impacts on carbon emissions. This constitutes an in-depth refinement of the model's land conversion module and a robust uncertainty analysis.

Finally, this study did not update the response curves themselves (e.g., the rates and shapes of forest growth and soil carbon decomposition). The model continues to use prescribed, time-invariant response curves to simulate changes in per-hectare carbon stocks across different ecosystems and land-use change types.

In our research on long-term land-use carbon budget estimation for the China region, we adapted this model's framework to specifically address the issue of 'Forest Conversion to Other Land' (FCO), a topic detailed in our methodology (e.g., Sect. 2.3.2 Calculating annual land-use change). To resolve this, we analyzed the specific circumstances of FCO in China and selected combinations of response curves better suited to local characteristics. Consequently, while the fundamental forms of the response curves remain unchanged, their application rules for the China region were more thoroughly explored and refined.

**Point 12.** -Section 4.3 (Uncertainty Analysis) & Figure 11. Since you mention "2.3.4 Uncertainty assessment" in Methods, the Monte Carlo simulation results should appear in the Results. Reserve methodological limitations for the Discussion.

Response: Thank you for this comment.

We fully agree with your assessment. This is an excellent suggestion that significantly improves the logical structure of our manuscript and aligns it more closely with standard scientific writing conventions. Following your advice, we have

restructured the paper by creating a new subsection, Section 3.3 'Uncertainty and Sensitivity Analysis,' within the Results. We have moved the paragraphs detailing the results of the sensitivity analysis and Monte Carlo simulation, along with the corresponding Figure 11, from the Discussion to this new section. The remaining content discussing methodological limitations has been retained in the Discussion under the revised, more precise heading Section 4.3 'Limitations.' We are confident that these changes create a clearer distinction between our findings and their limitations (please see the revised manuscript, lines 467-492).

**Point 13.** -Data Implications for ESSD. As ESSD emphasizes data utility, expand on Applications: How can this dataset advance regional carbon budget assessments, climate modeling, or policy evaluations?

Response: Thank you for this comment.

Following your valuable suggestion, we fully acknowledge the importance of providing a more detailed discussion on the practical applications of our dataset. To this end, we have introduced a new section into the manuscript, Section 4.3 ("Implications and Applications"), to specifically elucidate how our dataset can facilitate future scientific research in the three critical areas of regional carbon budget assessment, climate modeling, and policy evaluation. Within this new section (lines 597-607), we have specifically detailed the following aspects:

1) For regional carbon budget assessment, the dataset provides a robust historical baseline for carbon fluxes from land-use change, enabling the separation of legacy emissions from contemporary fluxes. This is crucial for accurately attributing the drivers of the current terrestrial carbon sink and evaluating the effectiveness of ecological restoration efforts.

2) In climate and Earth system modeling, the dataset serves as an independent benchmark for evaluating and refining Dynamic Global Vegetation Models (DGVMs). Validation against the provincially-resolved emission estimates from this study can help constrain model parameters related to ecosystem responses to land-use change.

3) For policy evaluation, the dataset offers long-term quantitative evidence to assess the efficacy of land-use policies. The key transition from a carbon source to a sink around the 1980s strongly coincides with the implementation of China's large-scale ecological restoration policies, thus supporting the assessment of the potential effectiveness of such national-level interventions.

We are confident that this comprehensive elaboration has thoroughly addressed your concerns, effectively showcasing the scientific importance and practical value of our dataset for promoting frontier research in related fields.

**Point 14.** -Line 507. Table 2 should be Table 3?

Response: Thank you for this comment. Revised.

**Point 15.** -Regarding Figure 4: Is the term 'forest-grassland boundary' (林草界线) conventionally accepted? Suggest revising it to simply 'Eastern/Western China' for clarity.

Response: Thank you for this comment. Revised.

---

## Author Response (AR3)

**Response to Reviewer 1 Comments**

The authors have addressed most of previous comments, and the manuscript has been substantially improved. Below are a few additional suggestions for consideration:

Response: Thank you very much for your valuable suggestions for revision. Following your advice, we have carefully and thoroughly revised the manuscript. The main revisions are as follows:

● Regarding the framing of the carbon density analysis: We have accepted your suggestion and explicitly framed this work as a methodological contribution in the manuscript—namely, the integration and compilation of existing data—and have revised the wording in the Abstract and Methods sections accordingly.

● Regarding the presentation of the data table: We have clarified the reason for including only the land types (forests and grasslands) that are directly relevant to the "bookkeeping model" methodology employed in this study.

● Regarding the comparative analysis and data consistency: We have made several key updates. First, we have updated the comparison data to the latest Global Carbon Budget 2024 (GCB 2024). Second, we have re-verified our own carbon flux estimates, confirming and explaining the internal consistency between the figure's curve and the table's data regarding the source-to-sink transition point (after 1980). Most importantly, guided by your insightful suggestion, we have completely rewritten the comparative analysis section (Section 4.2) to systematically elucidate the core conceptual differences between various models (bookkeeping models vs. DGVMs).

**Point 1.** While carbon density is a key component of the study, its novelty may be overstated in the context of a millennium-scale analysis. Given the inherent uncertainties over such extended timescales, the accuracy of current density mapping could be largely obscured. It may be more appropriately framed as a methodological contribution rather than a novel finding.

Response: Thank you for this comment. Revised.

We fully agree with your view and have revised the manuscript accordingly. We have carefully reviewed the entire manuscript and revised the wording in the Abstract (lines 25–27) and Methods section (lines 212–213) to reflect this perspective. Specifically, we now clarify that, based on previously published datasets, our contribution lies in the integration and compilation of existing data.

**Point 2.** Table 2, how about data for other land types, e.g. cropland, other land? They are not all zeros, correct?

Response: Thank you for this comment.

In the literature and books on carbon density that we collected since 1980, the records primarily cover three major land categories: forest, grassland, and cropland, with only sporadic data available for other land types. For the purposes of this study, we employed the bookkeeping method (a statistical model) proposed by Houghton and Castanho (2023), which incorporates carbon density values for forests and grasslands. Therefore, in Table 2 we present only the data relevant to our analysis, and the carbon density values for other land-use types from the original sources are not displayed here.

**Point 3.** It is becoming clear that over the past few decades, LUC in China is shifting from a net CO2 source to a net sink (or at least neutral). This is evidenced by the latest GCB 2024 (Fig. 7b in Friedlingstein 2024; https://doi.org/10.5194/essd-17-965-2025) and a recent Nature Climate Change study (Fig. 1 and 3a in Zhu 2024; http://dx.doi.org/10.1038/s41558-025-02296-z). In particular, the Houghton model (H&C) in GCB 2024 shows negative fluxes for China since the 1960s (data is here: https://globalcarbonbudgetdata.org/latest-data.html). While your Fig. 10 seems to show negative fluxes since 1970-80s for "this study", Table 3 presents a positive 2.25 Pg C for 1980-2019. Any clarification on this discrepancy? Additionally, why do you choose older GCBs instead of the latest one? Please clarify or justify. I would encourage an updated discussion on "4.2 Comparison

with previous estimates". In the discussions, please also note that the concept of LUC emissions differs between bookkeeping vs. DGVM models.

Response: Thank you for this comment. Revised.

Section 4.2 (see lines 545-631) presents a completely rewritten comparative analysis against previous research. Its central focus is the systematic clarification of conceptual distinctions between different estimation approaches. The analysis categorizes existing estimates into three main types—Bookkeeping Models (BKM), Dynamic Global Vegetation Models (DGVM), and National Reports—and explains the fundamental differences in their respective accounting scopes. Please note that the previous Figure 10 is now referred to as Figure 11 in this revised document because of structural changes and the insertion of new figures.

(1) Update to Figure 11. We have updated Figure 11 by replacing the comparison dataset "GCB 2019" with the latest "GCB 2024." In addition, we have incorporated the carbon emission data for China (1981–2020) from Zhu et al. (2025) and added corresponding descriptions in the text (see lines 545-631).

[Figure]

**Figure 11.** Chinese historical land-use change-induced carbon emission flux estimated by different methods. For detailed descriptions of all data sources, models, and methodologies shown, please refer to the notes of Table 3.

(2) Verification of carbon flux estimates. We have re-checked the carbon flux estimates in this study, which are also available on Zenodo

(https://doi.org/10.5281/zenodo.14557386, 2025). Our results confirm that the carbon flux becomes negative (indicating a carbon sink) after 1980, rather than after 1970. Therefore, the curve shown in Figure 11 is not in conflict. The possible confusion may be due to the long time span of the curve (1700–2020) in Figure 11, which makes it difficult to visually distinguish between 1970 and 1980. You may refer to our openly available dataset for the exact values.

(3) Clarification of LUC emission concepts. In response to your suggestion to note that the concept of LUC emissions differs between bookkeeping models and DGVMs, we have emphasized this distinction in Section 4.2 (see lines 545-631).

We sincerely thank you for your two rounds of careful and professional reviews, which have substantially improved the quality and scientific rigor of our manuscript.

**Response to Reviewer 2 Comments**

Thank you for addressing most of my previous concerns. However, my main concerns in this second round focus on land-use transition matrix construction, Figure 10, and Table 3, which require substantial improvements for methodological validation and systematic comparison. The current approach lacks rigor in several key areas that undermine the study's credibility.

Response: We sincerely thank you for your profound and constructive feedback during this round of review.

In response to your key concerns regarding the construction of the land-use transition matrix and the associated uncertainties, and the incomplete comparison of different studies (Table 3, Figure 10, now Figure 11 in the revised manuscript), we have undertaken comprehensive, in-depth revisions to enhance the scientific rigor of our study.

The revisions are centered around three aspects:

(1) Regarding the selection of allocation rule in land conversion, following the review's suggestion, we additionally included the area-weighted method. Instead of using a 'probabilistic' approach, we further included the 'forest-priority' method, which, together with the 'grassland-priority' method used in the original manuscript, formed two opposing land conversions and helped building a full range of possible carbon fluxes.

In particular, we apologize for the fact that this comment was raised by the reviewer in the previous round of comment but not fully addressed. We now made our best effort to address this point.

(2) Inspired by the review's comment, we now improved the uncertainty quantification in the revised manuscript by incorporating the uncertainty caused by three sources: the reconstructed land-use data used as model input, the land-use transition rule, and the carbon density parameters applied in carbon flux accounting. In particular, the uncertainty caused by the allocation is a new aspect that is inspired by the reviewer.

(3) We improved the consistency, clarity and completeness of Figure 11 and Table 3. More importantly, the Section 4.2 was substantially revised to accommodate a complete, logic-based comparison among different studies. This section now begins by addressing the fundamental conceptual differences in the core concept of "land-use change emissions" among various estimation methods before proceeding to the comparison of different datasets. This restructuring provides a clearer logical flow and a more rigorous comparison among different data sources.

We hope that these revisions have adequately addressed your concerns and strengthened the methodological foundation and credibility of our conclusions. Please refer to our detailed responses to each of your comments below.

Thank you once again for your valuable guidance.

**Point 1.** Land-use transition matrix construction lacks methodological validation

The priority-based rules for allocating land-use transitions from aggregate area data introduce substantial uncertainty that remains unaddressed. The authors' justification that forest-to-cropland and forest-to-other-land have identical response curves misses the point—the issue is not the final carbon calculation but the arbitrary nature of the allocation rules themselves. Without testing alternative methods (area-weighted, probabilistic) or demonstrating robustness across different rule sets, the reliability of the entire reconstruction is questionable. This methodological uncertainty compounds over the millennium timescale and undermines confidence in the results.

Response: Thank you for your valuable feedback regarding the allocation rules. We fully agree on your suggestion that the choice of the allocation rule in land conversion can cause uncertainty in the estimated carbon fluxes, and therefore, alternative allocations rules should be explored.

Following your kind suggestion, to systematically assess the uncertainty introduced by the allocation method, we have constructed and compared multiple allocation rules and, based on this, further developed a more comprehensive uncertainty assessment

framework. The revision details are as follows:

(1) Setup and comparison of allocation rules (For detailed revisions, please see lines 320-346 of the main text, where all changes have been highlighted in red). To comprehensively assess the impact of different allocation rules on the results, we designed and compared the following three logically distinct rules.

The first rule remains the grassland-priority rule that was used in our original manuscript, which, to the best of our knowledge, has the highest accordance with the reality in China's land use history. More importantly, this rule maintains the internal consistency with the reconstruction methodology of the historical land-use dataset used in this study.

Following the review's suggestion, we used the area-weighted allocation method as the second rule. This rule assumes that outgoing land conversions are allocated to different incoming land types in proportion to their area's share of the total outgoing area.

For the third rule, instead of using a probabilistic-based approach as suggested by the reviewer, we used a forest-priority method which is in contrast to the 'grassland-priority' method. We argue that this method, together with the 'grassland-priority' method, helps to provide a minimum-maximum boundary in land use change areas, which further provides a complete estimate of the uncertainty in the derived land use carbon fluxes.

The details of these three allocation rules, along with a concrete example, are described in the revised manuscript of lines 320-346.

(2) Uncertainty assessment framework (For detailed revisions, please see lines 370-390 of the main text, where all changes have been highlighted in red).

The original uncertainty assessment incorporates only the uncertainties caused by land use data and the carbon density data but not the uncertainty from the allocation rule in the land conversion. The review's comment reminds us that the allocation rule is a critical source of uncertainty that we failed to account for in our original manuscript.

Hence, following the review's suggestion, we improved the uncertainty quantification in the revised manuscript by incorporating the uncertainty caused by three sources: the reconstructed land-use data used as model input, the land-use transition rule, and the carbon density parameters applied in carbon flux accounting. More specifically, for each of the three allocation rules, we performed a Monte Carlo simulation consisting of 1000 iterations to account for the uncertainties in the land use data and the carbon density parameters. To establish the final uncertainty in the estimated carbon flux, we aggregated the annual carbon emissions from all 3000 simulations (3 allocations rules × 1000 iterations for each rule) for each year and selected their maximum and minimum values.

We thank the review's comment which encourages us for a more complete estimation of the uncertainty as described above. The details are described in the lines 370-390 of the revised manuscript.

(3) Uncertainty and sensitivity analysis results (For detailed revisions, please see lines 480-516 of the main text, where all changes have been highlighted in red).

Following the reviewer's suggestion, we have constructed a more comprehensive uncertainty assessment framework. In this section, we present and interpret the results from this comprehensive uncertainty analysis.

To examine the influence of different allocation rules, we compared the annual land-use transition matrices generated by the grassland-priority method and the area-weighted method (see Appendix Fig. C2). The results indicate that although certain transition types exhibit numerical differences in area, the primary transition processes (e.g., the conversion of forest to cropland) show a high degree of consistency. This demonstrates that the core conclusions of our study are robust.

Nevertheless, the choice of allocation rule remains a significant source of overall uncertainty. The results of the comprehensive uncertainty assessment are presented in the revised Figure 10. This figure displays the millennial-scale carbon emission flux estimated using the grassland-priority allocation rule (black line), along with a comprehensive uncertainty interval (gray shaded area). This interval was constructed

by systematically integrating the results from all 3,000 Monte Carlo simulations (1000 for each of the three allocation rules) and selecting their maximum and minimum values. It thereby comprehensively reflects the combined uncertainties from the three main sources: input data, the allocation rule, and carbon density parameters.

Furthermore, a period-based analysis of the uncertainty (see Appendix Table C1) reveals its temporal evolution. The analysis shows that after 1982, when land use shifted to a carbon sink, the uncertainty exhibits a distinct asymmetry: the downward uncertainty (58.91 Tg C) is significantly larger than the upward uncertainty (34.56 Tg C). This result indicates that the actual sink strength of China's land use during the modern observational period is likely stronger than our current best estimate.

In summary, the new comprehensive assessment framework proposed in this study, by systematically considering multiple sources of uncertainty, provides a robust quantification for the estimated carbon fluxes. This analysis reaffirms the trends and key turning points in China's millennial-scale land-use carbon budget.

[Figure]

**Figure C2**. Differences in annual land-use transitions between the grassland-priority and area-weighted allocation methods. F-C denotes the conversion between Forest and Cropland, F-G represents Forest-Grassland conversion, F-O represents Forest-Other land conversion, and G-C represents Grassland-Cropland conversion. A positive difference indicates that the grassland-priority result is lower than the area-weighted result, and vice versa.

[Figure]

**Figure 10.** Carbon emission fluxes from land-use change in China with their uncertainties. The black line represents the mean result of 1000 Monte Carlo simulations using the grassland-priority allocation rule in land use conversion. The gray shaded area represents the uncertainty interval defined by the maximum and minimum values across all 3000 simulation runs (1000 iterations for each of the three allocation rules: grassland-priority, area-weighted, and forest-priority). This interval incorporates the combined uncertainties from input data, carbon density parameters, and the allocation rule. For details, refer to the sections 2.3.2 and 2.3.3.

**Table C1.** Carbon emissions from land-use change and their uncertainties in different historical periods

| Period | Mean annual estimate (Tg C) | Upward uncertainty | Downward uncertainty |
|---|---|---|---|
| Pre-1900 | 16.35 | 7.49 | 5.20 |
| 1900-1949 | 75.51 | 37.31 | 24.65 |
| 1950-1982 | 102.95 | 74.35 | 44.48 |
| Post-1982 | -60.90 | 34.56 | 58.91 |

Notes: All values are in the unit of Tg C/yr. The mean upward uncertainty is the average of the annual differences between the maximum value and the estimate within each period. The mean downward uncertainty is the average of the annual differences between the estimate and the minimum value.

**Point 2.** Figure 10 legend and data sources lack essential details. -Data sources for all model results need clear citation.-Update to latest GCB2024 data instead of GCB2019. Figure 10 presents Gasser (2020), Hansis et al.(2015), and Houghton (2023) alongside GCB2019 without explaining that these three studies are the component models underlying GCB estimates. This may mislead readers about the independence of these approaches. -NGHGI.DB and NGHGI.DB.corrected are undefined. What specific corrections were applied and how? -Justify TRENDYv8 selection (if due to additional

scenarios isolating LASC effects, state this explicitly)

Response: Thank you for these valuable comments. In response, we have comprehensively revised and restructured Section 4.2 Comparison with previous estimates to improve clarity regarding data sources, definitions, and their interrelationships. Specifically, we have implemented the following key changes:In response to your specific points in Comment 2 regarding data sources, definitions, and their logical relationships, we have systematically rewritten and comprehensively reorganized Section 4.2 Comparison with previous estimates.

(1) Updating to GCB2024: As suggested, we have updated our comparison dataset from GCB2019 to the latest GCB2024 version, ensuring our analysis is based on the most current data.

(2) Clarifying data relationships: We have revised the text, and the footnotes of Table 3 to explicitly state that the GCB2024 estimate is the mean of its component bookkeeping models. This directly addresses the potential for misunderstanding the independence of these datasets.

(3) Defining datasets and justifying selections: We have added clear definitions and justifications for our choice of comparison datasets:

NGHGI.DB.corrected is now defined as the original national inventory data adjusted by subtracting carbon fluxes from indirect environmental changes (e.g., $CO_2$ fertilization). This correction conceptually aligns its accounting boundary with our bookkeeping model.

The selection of the TRENDYv8 present.day scenario is now justified in the text. This scenario is specifically designed to be conceptually consistent with bookkeeping models by using fixed, modern environmental conditions.

(4) Improving the section's logical structure: Beyond addressing these specific points, we have fundamentally restructured Section 4.2. As part of this effort, we have substantially revised Table 3 to serve as a comprehensive summary for the numerical comparison, now supported by detailed footnotes that provide all necessary definitions. The main text now precedes this comparison by first explaining the core conceptual differences between the three main estimation methods (Bookkeeping

Models, DGVMs, and National Reports). This provides a clearer framework for the reader to understand the discrepancies between different estimates.

**Table 3.** Comparison of average annual carbon flux estimates caused by land-use change in China

| Reference | Model Type[1] | Average annual carbon flux (Tg C yr⁻¹) | | | |
|---|---|---|---|---|---|
| | | Pre-1900 | 1900-1949 | 1950-1980 | Post-1980 |
| This Study | BKM | 40.50 | 75.51 | 106.39 | -55.23 |
| Yang et al. (2023) | BKM | 25.79 | 51.31 | 42.76 | / |
| Ge et al. (2008) | BKM | 18.48 | 49.70 | / | / |
| Yang et al. (2019) | BKM | 23.54 | 42.25 | 149.36 | / |
| Houghton and Castanho (2023) | BKM | 48.14 | 100.59 | 6.57 | -6.98 |
| Qin et al. (2024) | BKM | -3.81 | 77.06 | 236.13 | 73.68 |
| Gasser et al. (2020) | BKM | / | / | 82.50 | 20.23 |
| Hansis et al. (2015) | BKM | / | / | 126.70 | 212.73 |
| GCB2024[2] (Friedlingstein et al., 2025) | BKM | 21.20 | 83.64 | 164.33 | 36.27 |
| Zhu et al. (2025)[3] | Stock-Difference Method | / | / | / | -49.33 |
| TRENDYv8.present.day[4,5] | DGVM | / | / | 107.40 | 190.39 |
| FAOSTAT[4,6] | National Reports | / | / | / | -121.06 |
| NGHGI.DB.corrected[4,7] | National Reports | / | / | / | -180.36 |

Note: The values in the table represent the average annual carbon fluxes (Tg C yr⁻¹) for the specified time periods. A "/" indicates that data for that period is not available or not provided in the source study.

[1] Model Type: BKM (Bookkeeping Models) aim to estimate carbon fluxes from direct anthropogenic land-use activities; DGVM (Dynamic Global Vegetation Models) simulate the integrated response of ecosystems to both land use and environmental changes, which includes the additional loss of sink capacity (LASC) that would otherwise occur, for example, on an actually cleared forest (refer to Gasser et al., 2020 for a detailed explanation); National Reports (NGHGI) are based on the IPCC's "managed land proxy" principle for accounting, which includes carbon sink driven by both direct anthropogenic land use and also environmental effects, but not the LASC.

[2] The value for GCB2024 is the mean of the carbon fluxes derived by the four bookkeeping models: Qin et al. (2024), Houghton and Castanho (2023), Gasser et al. (2020), and Hansis et al. (2015).

[3] Zhu et al. (2025) employs a stock-difference method, constructing high-resolution, dynamic carbon stock maps by integrating remote sensing, inventory data, and machine learning, and calculates the flux from changes in these maps over time.

[4] Data for TRENDYv8.present.day, NGHGI.DB.corrected, and FAOSTAT are from Obermeier et al., (2024).

[5] TRENDYv8.present.day represents DGVM simulation results run under fixed, modern environmental conditions (climate and $CO_2$ concentration), designed for conceptual alignment with bookkeeping models that use modern carbon densities.

6 FAOSTAT provides bottom-up estimates by applying IPCC guidelines to country-reported activity data (e.g., from the Forest Resources Assessment) and geospatial information. Conceptually, these estimates are closer to bookkeeping models as they often do not include the indirect environmental effects (such as $CO_2$ fertilization).

7 NGHGI.DB.corrected is derived from the original national inventory data (NGHGI.DB) by subtracting the carbon fluxes on "managed land" that are caused by indirect environmental changes (e.g., $CO_2$ fertilization), as estimated by DGVMs. This makes this estimated conceptually aligned with the bookkeeping models.

**Point 3.** Table 3 is incomplete and inconsistent with Figure 10. The current Table 3 lacks systematic collection of comparable studies and shows inconsistency with Figure 10 content. While Figure 10 includes multiple DGVM studies for China, Table 3 only presents Yu et al. as the sole DGVM representative without justification for this selective inclusion, and other bookkeeping model results as well as NGHGI/FAOSTAT shown in Figure 10 are not included in Table 3.

Response: Thank you for highlighting this critical inconsistency.

We agree that the original Table 3 was incomplete and not fully aligned with the figure. In response, we have comprehensively revised Section 4.2, with a particular focus on reconstructing Table 3 to ensure it is systematic, comprehensive, and consistent with Figure 11 (previously Figure 10). For detailed revisions, please see lines 553-577 of the main text, where all changes have been highlighted in red. The key revisions are as follows:

(1) Reconstructing Table 3 for Consistency and Clarity: The original Table 3 has been completely replaced. The new version now systematically includes all comparable studies presented in Figure 11, resolving the inconsistency you pointed out. To facilitate a more standardized comparison, the table is now structured to show period-based annual average fluxes for each study, providing a much clearer quantitative summary.

(2) Clarifying the Scope of the Table and Figure: We have clarified why minor differences in content between the table and figure persist. The revised table notes now explicitly explain that Table 3 includes some studies not plotted in Figure 11.

This is because those sources only report cumulative fluxes over a period and lack the annual time-series data required for plotting.

(3) Optimizing the Underlying Comparison Framework: We recognized that the inconsistency you identified was symptomatic of a weaker analytical framework in the original manuscript. Therefore, we have rewritten the entire Section 4.2. The new structure first establishes the conceptual differences between estimation methods (Bookkeeping Models, DGVMs, National Reports) before presenting the numerical comparison in the revised Table 3 and Figure 11. This provides a much more rigorous and logical foundation for comparing the different estimates.

**Point 4.** The statement "estimates in this study fall within the range of existing model estimates at an intermediate level" is incorrect. Figure 10 shows your results are among the most negative values post-2000 (excluding NGHGI data due to different definitional boundaries).

Response: Thank you for your keen observation; you are absolutely correct.

Our previous generalization that our estimate was at an "intermediate level" was inaccurate, and we have thoroughly reflected on this and made a complete correction. The key revisions are as follows:

(1) Removal of the inaccurate statement: We have completely removed this inaccurate generalization.

(2) Providing a More Rigorous Positioning and Discussion: The issue you identified stemmed from a less rigorous comparison framework in our original manuscript. As mentioned in our responses to your previous points, we have rewritten the entire Section 4.2, establishing a rigorous "concepts first, data second" comparison framework.

Within this new framework, instead of making a broad generalization, we use specific data comparisons to reach a more precise conclusion that aligns with your observation: Our results clearly indicate that China's land use has been a significant carbon sink (-55.23 Tg C yr$^{-1}$) in recent decades. This finding not only stands in stark

contrast to mainstream models that rely on global-scale datasets and show a carbon source (e.g., GCB2024), but is also corroborated by the results of another study using localized data (Zhu et al., 2025), together revealing a strong carbon sink signal that has been missed by global models.

This more accurate positioning and discussion are now presented in detail in the revised Section 4.2 (lines 545-631). We thank you again for your correction, which has led to a more precise and insightful interpretation of our own findings.

**Response to Reviewer 3 Comments**

General Comments. This paper estimates carbon emissions from land-use change in China over the past millennium, which is highly relevant and aligns well with ESSD's scope. The use of historically reconstructed land-use datasets—based on China's unique archival records rather than proxy indicators like population—is particularly valuable. While the authors have addressed some concerns in prior revisions, several issues remain regarding paper structure, integration of historical and modern land-use data, spatial resolution of historical datasets, and practical applications of the results. Additionally, some of my comments overlap with previous reviewers' feedback; I urge the authors to prioritize these shared concerns.

Response: Thank you for your insightful and constructive comments, which have been crucial for improving our manuscript. We have thoroughly revised the paper based on your suggestions, focusing on three key areas: strengthening our scientific rationale, improving the logical structure, and clarifying the practical applications of our dataset.

Key revisions include restructuring the manuscript for a clearer separation of the Methods, Results, and Discussion sections. We have also expanded our methodology to better explain the integration of historical and modern data and have introduced a new sensitivity analysis to quantify the impact of our core assumption of static carbon densities. Finally, to highlight the dataset's value as encouraged by ESSD, we added a dedicated section on its applications in climate research and policy assessment.

We believe these changes substantially strengthen the paper. Below are our detailed responses to each of your points. Thank you again for your time and expertise.

Specific Comments.

**Point 1.** -Data Integration Issues. Clarify how reconstructed data (e.g., cropland from tax records) align with survey-based statistics (e.g., the Second and Third National Land Surveys). Land-use definitions evolved between surveys (e.g., the Second

National Land Survey [2009] and Third National Land Survey [2019]). Discuss potential errors introduced by these definitional shifts.

Response: Thank you for your valuable feedback. Revised.

We have carefully considered the issue of inconsistent statistical calibers in our data integration. Your point is crucial. In processing the land-use data from 1980 to the present, we faced two primary options: one being the annual-resolution land use/cover datasets derived from remote sensing interpretation, and the other being the national-level, survey-based statistical data, namely the Second and Third National Land Surveys. We chose the latter primarily because the reconstructed historical data we used is more closely aligned and compatible with the national survey data in terms of its sources, methodologies, and nature (e.g., statistics and mapping based on administrative units). We believe that linking datasets of a similar nature helps maintain consistency in the long-term trends and mechanisms.

We fully agree with your observation that even between the Second and Third National Land Surveys, the land classification standards have evolved and differ, which poses challenges for direct data linkage. To minimize the errors introduced by these definitional discrepancies, we performed specific harmonization and adjustments for the most sensitive land class in our carbon flux model: forest. Specifically, the original research literature for the historical forest data explicitly states that its definition of 'forest' is conceptually closest to the 'closed forest land' sub-category in current classification standards. Therefore, when linking with modern data, we did not use the total area of the primary 'forest land' category. Instead, we precisely extracted the data for the 'closed forest land' sub-category from both the Second and Third surveys to ensure maximum definitional consistency with the historical reconstructed data.

Despite these efforts, we acknowledge that the definitional evolution of other land classes (e.g., cropland, grassland) across different survey periods, along with the inherent discrepancies in statistical calibers between the reconstructed and survey-based data, remains a source of uncertainty in this study. These differences will inevitably affect the accuracy of the final carbon budget estimation. We will

explicitly address this point in the discussion section of our paper and identify it as an important area for future research, which could be advanced through data fusion or the development of more optimal classification conversion algorithms (see lines 645-655 of the manuscript).

**Point 2.** -Line 141. are considered highly credible. Cite references for this statement.

Response: Thank you for this comment. Revised.

   Thank you for your valuable feedback. Regarding your comment on the supporting evidence for our statement that the National Land Survey data "are considered highly credible" (Line 141), we have carefully considered the point and revised the manuscript. We fully agree that providing a clear justification for the reliability of this key dataset is essential. In our revision, we reflected on the best way to establish this credibility. For official census data of this nature—organized by the highest state administrative body and mobilizing national resources—its authority and reliability are typically accepted as a consensus or a benchmark in the academic community. It serves as a foundational starting point for research, rather than a debatable claim requiring repeated justification. Therefore, we concluded that the most rigorous and direct method to demonstrate its credibility is not by citing an indirect evaluation from another study, but by elucidating the rigorous nature of the data production process itself. Based on this reasoning, we have revised the original, more general statement and replaced it with a specific description of the survey process. Please see the revised text in lines 142-143.

**Point 3.** -Line 145. The text describes vegetation carbon density first but later details soil carbon density before vegetation. Revise for logical flow.

Response: Thank you for this comment. Revised. (Line 212-213)

**Point 4.** -Temporal Stability of Carbon Densities. Soil/vegetation carbon densities are treated as static over the millennium. The Discussion notes this limitation, but quantify its impact: Would assuming stable densities overestimate or underestimate emissions?

Response: Thank you for your insightful suggestion to quantify the impact of assuming static carbon density over time. Revised.

We fully agree that this is a critical scientific issue. Accordingly, we have designed and completed a sensitivity analysis to assess the potential effects of this assumption on our estimation results. Our analysis is based on the posited systematic differences between historical and modern carbon pools in vegetation and soil. We hypothesized that historical vegetation carbon density was likely systematically lower than modern levels, a premise primarily based on the limited 'CO$_2$ fertilization effect' under significantly lower pre-industrial atmospheric CO$_2$ concentrations (approx. 280 ppm vs. >420 ppm today). Conversely, we posited that historical soil carbon density was likely higher than the modern average, mainly due to less intensive anthropogenic disturbance, which allowed soil organic carbon pools in extensive ecosystems to remain closer to a state of natural saturation.

Based on this rationale, we designed a scenario assuming that historical vegetation carbon density was 20% lower and soil carbon density was 20% higher than modern values. After recalculating based on this scenario, we conducted an in-depth analysis of the annual differences between the new and original estimates, revealing distinct temporal patterns. During the carbon source periods, which constitute the vast majority of the study period (approx. 982 years), the new estimates were consistently lower than the original values, with a mean annual difference of approximately -2 Tg/yr, indicating a smaller and more stable range of deviation. In contrast, during the few years identified as carbon sink periods (approx. 37 years), influenced by the intense land-use change during those times, the discrepancy between the two estimates showed greater uncertainty and volatility, with differences ranging from -5 to +11 Tg/yr. This period-segmented analysis indicates that our original methodology may lead to a systematic overestimation of carbon fluxes, and that the uncertainty of

this estimation is particularly pronounced during carbon sink periods. We believe this new, more in-depth analysis substantively addresses your concerns and significantly enhances the rigor of our paper's discussion on uncertainties (the detailed revisions in the main text can be found in lines 676-685).

Thank you again for your valuable feedback, which has greatly improved the quality of our research.

**Point 5.** -Line 164–165. Briefly summarize the framework of the transfer function for bulk density estimation. Technical details can remain in cited sources.

Response: Thank you very much for your valuable feedback. Revised.

In our study, for sample points that lacked measured bulk density data, we employed an empirical transfer function established and validated in Yang et al. (2007) for estimation. This function is based on the significant negative correlation between soil organic matter (SOM) content and bulk density, a relationship that has been widely confirmed in soil science studies. According to that paper, the specific formula for estimation is:

$$BD = 0.29 + 1.2033 \times e^{-0.0775 \times SOM}$$

where BD is the bulk density to be estimated (unit: g/cm³ ) and SOM is the percentage of organic matter content in the corresponding soil layer (%). The model's goodness-of-fit ($r^2$) is 0.81 (p<0.01), which indicates a high degree of reliability. To make our research methods clearer and more transparent, we have followed your suggestion and added this specific formula and explanation to the methods section of the revised manuscript. Please see lines 232-235 for the detailed revisions in the main text.

**Point 6.** -Table 2. Add province/region codes (e.g., "No.1" for Jing-Jin-Ji) to align with Figure 1.

Response: Thank you for this comment. Revised.

Following your recommendation, we have revised Table 2 to ensure its consistency with Figure 1. We have added a new column, "Code," and reordered the rows to align with the regional numbering (No. 1, No. 2, ...) presented in Figure 1. This modification greatly improves the coherence between the table and the figure. The revision can be found on lines 247-250.

**Point 7.** -Figure 2. Explain color schemes in the flowchart. In other words, What is the meaning of each color in the flowchart?

Response: We thank the reviewer for their constructive feedback and agree that the color scheme in Figure 2 required clarification. Revised.

To address this, we have added an explanatory sentence to the figure caption, defining the module represented by each color (please see lines 258-260 of the revised manuscript). We are confident that this revision improves the clarity of our research framework and the overall readability of the figure.

**Point 8.** -Line 195. Spatially explicit cropland/forest/grassland data exist (e.g., SCES literature). Justify why provincial-scale aggregation was used instead.

Response: Thank you for your valuable feedback. The question you raised regarding our choice to use provincial-scale summary data is a critical methodological consideration of our study. Our decision to use the provincial scale as the primary analytical unit was made deliberately, based on a comprehensive assessment of multiple factors, including data reliability, time-series consistency, and scale matching with key parameters (i.e., carbon density). The specific reasons are detailed below:

1. Data Reliability and Uncertainty:

While the spatially explicit (i.e., gridded) long-term Land Use/Cover Change (LUCC) datasets you mentioned do offer a high-resolution perspective, they are typically generated through techniques such as spatial downscaling or data fusion. This process inevitably introduces uncertainties stemming from model assumptions.

The reliability of such datasets is particularly challenging for historical periods. In contrast, provincial-level statistics are aggregated from long-term, relatively standardized administrative reporting systems. Although they have a lower spatial resolution, they represent the fundamental unit for historical land-use records in China and possess a high degree of reliability.

2. Time-Series Consistency and Continuity:

This study aims to construct a long-term inventory of carbon emissions from land use, for which data continuity and consistency are paramount. The data for the later period of our study were linked and calibrated with the "Second National Land Survey (2009)" and the "Third National Land Survey (2019)". Currently, the authoritative and fully open-access versions of these two surveys, which share a consistent statistical scope, are primarily available at the provincial summary level. Adopting the provincial scale thus maximizes the consistency of data sources and standards throughout the entire study period.

3. Scale Matching with Key Parameters (Carbon Density):

The core of our research is to estimate carbon emissions driven by land-use change, which requires coupling land-use area data with corresponding carbon density data. The carbon density datasets we employed—including data from the Second National Soil Survey of China (1979–1985), the China land ecosystem carbon density dataset by Xu et al. (2019), and the more recent Chinese Soil Series (since 2008)—are all fundamentally derived from field surveys and measurements at sample points. These sample points are spatially discrete and do not provide complete grid coverage. Therefore, aggregating both the land-use data and the carbon density sample data to the provincial scale is a more methodologically robust approach that ensures better compatibility between them.

In summary, while spatially explicit data offers advantages in displaying spatial patterns, we chose the provincial scale as it is the most appropriate and robust strategy for our research objectives. These objectives prioritize the construction of a long-term time series, the assurance of data reliability and consistency, and the scientifically sound coupling of land-use data with carbon density data derived from sample points.

This decision was a trade-off made after carefully evaluating the strengths and weaknesses of different data sources to ensure the accuracy and reliability of our final estimates.

**Point 9.** -Line 263. Provide references for "SAGE" and "PJ" datasets at first mention.

Response: Thank you for this comment. Revised.

For details, please see lines 192-193 of the revised manuscript.

**Point 10.** -Section 2.3.2 (Reliability Assessment). This section describes data sources rather than evaluating reliability. Move it to Section 2.2.1 (Land-use Data) for cohesion. If you put it in Methods, then the reliability results belong in Results/Discussion.

Response: We thank the reviewer for this insightful comment. Revised.

We agree that the content previously in Section 2.3.2 was a description of data sources and was misplaced. Accordingly, we have relocated this text to Section 2.2 Data sources to improve the manuscript's structure. This change is reflected in lines 147-210 of the revised version.

**Point 11.** -Line 500. Clarify key improvements in the latest bookkeeping model, including the updated disturbance-response curves, refined land-use transition rules? Or anything else?

Response: Thank you for this comment.

First, the new model refines the simulation of wood harvest to better reflect actual harvesting practices. The adjustments include correcting the post-harvest carbon allocation between 'wood products' and 'slash' to align with FAOSTAT data, and reducing the harvest intensity in secondary forests. This latter change necessitates simulating a larger harvested area to meet the same wood volume, thereby increasing

the gross carbon sink from forest recovery.

Second, a key update is the proposal and simulation of alternative interpretations for 'Forest Conversion to Other Land' (FCO), a phenomenon observed in many tropical countries where the net loss of forest area exceeds the net gain in agricultural land. In contrast to the Houghton and Nassikas (2017) study which assumed a single pathway ('recovering forest'), this new research explores additional land-use conversion rules, including statistical error, 'shifting cultivation,' and 'degraded land,' to assess their distinct impacts on carbon emissions. This constitutes an in-depth refinement of the model's land conversion module and a robust uncertainty analysis.

Finally, this study did not update the response curves themselves (e.g., the rates and shapes of forest growth and soil carbon decomposition). The model continues to use prescribed, time-invariant response curves to simulate changes in per-hectare carbon stocks across different ecosystems and land-use change types.

In our research on long-term land-use carbon budget estimation for the China region, we adapted this model's framework to specifically address the issue of 'Forest Conversion to Other Land' (FCO), a topic detailed in our methodology (e.g., Sect. 2.3.2 Calculating annual land-use change). To resolve this, we analyzed the specific circumstances of FCO in China and selected combinations of response curves better suited to local characteristics. Consequently, while the fundamental forms of the response curves remain unchanged, their application rules for the China region were more thoroughly explored and refined.

**Point 12.** -Section 4.3 (Uncertainty Analysis) & Figure 11. Since you mention "2.3.4 Uncertainty assessment" in Methods, the Monte Carlo simulation results should appear in the Results. Reserve methodological limitations for the Discussion.

Response: Thank you for this comment. Revised.

We fully agree with your assessment. This is an excellent suggestion that significantly improves the logical structure of our manuscript and aligns it more closely with standard scientific writing conventions. Following your advice, we have

restructured the paper by creating a new subsection, Section 3.3 'Uncertainty and Sensitivity Analysis,' within the results. We have moved the paragraphs detailing the results of the sensitivity analysis and Monte Carlo simulation, along with the corresponding Figure 11, from the Discussion to this new section. The remaining content discussing methodological limitations has been retained in the Discussion under the revised, more precise heading Section 4.3 'Limitations.' We are confident that these changes create a clearer distinction between our findings and their limitations (please see the revised manuscript, lines 480-516).

**Point 13.** -Data Implications for ESSD. As ESSD emphasizes data utility, expand on Applications: How can this dataset advance regional carbon budget assessments, climate modeling, or policy evaluations?

Response: Thank you for this comment. Revised.

Following your valuable suggestion, we fully acknowledge the importance of providing a more detailed discussion on the practical applications of our dataset. To this end, we have introduced a new section into the manuscript, Section 4.3 ("Implications and Applications"), to specifically elucidate how our dataset can facilitate future scientific research in the three critical areas of regional carbon budget assessment, climate modeling, and policy evaluation.Within this new section (lines 633-642), we have specifically detailed the following aspects:

1) For regional carbon budget assessment, the dataset provides a robust historical baseline for carbon fluxes from land-use change, enabling the separation of legacy emissions from contemporary fluxes. This is crucial for accurately attributing the drivers of the current terrestrial carbon sink and evaluating the effectiveness of ecological restoration efforts.

2) In climate and Earth system modeling, the dataset serves as an independent benchmark for evaluating and refining Dynamic Global Vegetation Models (DGVMs). Validation against the provincially-resolved emission estimates from this study can help constrain model parameters related to ecosystem responses to land-use change.

3) For policy evaluation, the dataset offers long-term quantitative evidence to assess the efficacy of land-use policies. The key transition from a carbon source to a sink around the 1980s strongly coincides with the implementation of China's large-scale ecological restoration policies, thus supporting the assessment of the potential effectiveness of such national-level interventions.

We are confident that this comprehensive elaboration has thoroughly addressed your concerns, effectively showcasing the scientific importance and practical value of our dataset for promoting frontier research in related fields.

**Point 14.** -Line 507. Table 2 should be Table 3?

Response: Thank you for this comment. Revised.

**Point 15.** -Regarding Figure 4: Is the term 'forest-grassland boundary' (林草界线) conventionally accepted? Suggest revising it to simply 'Eastern/Western China' for clarity.

Response: Thank you for this comment. Revised.

---

## Author Response (AR4)

**Topic editor**

Please address the comments from Reviewer #3 and also include a discussion comparing your land use/cover data with those from other studies. For exampe: Liu, M., and H. Tian (2010), China's land cover and land use change from 1700 to 2005: Estimations from high-resolution satellite data and historical archives, Global Biogeochem. Cycles, 24, GB3003, doi:10.1029/2009GB003687.

Response: Thank you for this comment.

Thank you for your and the reviewer's valuable comments. We have comprehensively revised the manuscript in response to the comments from Reviewer #3. Specifically, we have optimized the tense and logical structure of the Abstract and Introduction, and streamlined the map of the study area (Figure 1) to avoid potential confusion.

Furthermore, in response to your specific suggestion regarding the comparison of land-use data: We have added a new section, "4.1 Comparison with other land-use reconstructions," in the Discussion, which comprehensively reviews the significant biases of global long-term land-use datasets regarding China and systematically compares the land-use data used in this study with existing representative datasets (including Liu and Tian, 2010; Cao et al., 2014; Yang et al., 2018; Yu et al., 2021). Our analysis reveals a high degree of consistency in the data sources of these studies: the historical cropland and forest data reconstructed by Ge et al. (2004) and He et al. (2008) serve as the common foundation for multiple studies, including Liu and Tian (2010). This study is grounded in the same data system but utilizes the updated millennial-scale versions based on the latest historical documents. Although these studies differ in multi-type integration or gridding reconstruction methods, the data used in this study exhibit highly consistent trends with the aforementioned representative datasets during the overlapping period (the past 300 years) and at the provincial scale. Compared to most existing studies that focus on the past 300 or 100 years, the dataset used in this study provides a continuous perspective on a millennial scale (1000–2019), thereby enabling the capture of carbon emissions over a longer

historical period. For detailed discussion, please refer to Lines 513–535 in the main text.

**Response to Reviewer 3 Comments**

My previous comments and suggestions were well addressed. The paper's structure and readability have significantly improved. I have a few minor additional suggestions for the authors' reference.

**Point 1.** First, regarding the Abstract. Lines 19–21 (this part summarizes content from Lines 92–96 in the Introduction)—is it necessary to retain this in the Abstract? The statement that "uncertainty in land use carbon emission estimates for recent decades is greater than that for the past 300 years" may confuse readers, as land use data for the past 300 years are likely to have greater inherent uncertainty. This discrepancy might arise because more studies focus on recent decades, leading to larger variances in results, whereas the actual uncertainty in estimates for the past 300 years could be even higher. I wonder if this paper includes results for China similar to those reported in: Kaplan, J. O., Ruddiman, W. F., Crucifix, M. C., Oldfield, F. A., Krumhardt, K. M., Ellis, E. C., Ruddiman, W. F., Lemmen, C., and Klein Goldewijk, K.: Holocene carbon emissions as a result of anthropogenic land cover change, The Holocene, 21, 775–791, https://doi.org/10.1177/0959683610386983, 2011.

Response: Thank you for this comment. Revised.

We fully agree with the reviewer's assessment. The statement regarding uncertainty in the Abstract was indeed potentially misleading and has been removed. To clarify the data scope, the 150% discrepancy covers results from both global models (e.g., BLUE, H&N) and studies by Chinese scholars for the period 1950–2021, where substantial divergences in parameter settings, driver data, and accounting boundaries lead to a high degree of dispersion. In contrast, the 102% discrepancy pertains exclusively to research by Chinese scholars regarding the past 300 years (1700–2000) based on domestic historical documents, and this statistic does not include the study

by Kaplan et al. (2011).

Please refer to line 19 of the Abstract and lines 92–95 of the Introduction for the revisions.

**Point 2.** Second, Lines 76–78: Are the land use changes described in the Introduction reflected in the carbon emission results presented in this paper? Please ask the authors to verify this.

Response: Thank you for this comment. Revised.

We confirm that the land use changes and associated degradation processes described in the Introduction (Lines 76–78) are reflected in our carbon emission results. First, these environmental degradation phenomena (e.g., erosion on the Loess Plateau, vegetation destruction in southern hill regions) are essentially the ecological consequences of high-intensity anthropogenic land-use disturbances, processes that are fully captured by the historical land use reconstruction data used in this study. Second, we classified forest loss that was not converted into permanent cropland as "Forest conversion to Other Land," which primarily corresponds to historical shifting cultivation and degradation caused by over-exploitation. Our results show that this conversion type contributed 68.45% of the total historical emissions (see Fig. 6d in the main text). This indicates that our estimates robustly quantify the high-intensity land-use activities described in the Introduction and their resulting carbon emissions.

We have made the necessary clarifications in Section 4.1 of the Discussion (please refer to lines 555–557).

**Point 3.** Third, Lines 100–108: It is suggested to either rephrase this section using the simple future tense or simplify it in the Introduction. The focus should instead be on content to be described after completing this study, which could be better placed in the Discussion section.

Response: Thank you for this comment. Revised.

Following the reviewer's suggestion, we have revised the phrasing of this paragraph in the main text; please refer to lines 97–103 of the Introduction.

**Point 4.** Fourth, Figure 1 (map of the study area): It is recommended to retain only subfigure (f). Subfigures (a)–(e) do not appear to be referenced later in the text; including them here may confuse readers about the actual study area of this paper. I have a few minor additional suggestions for the authors' reference.

Response: Thank you for this comment. Revised.

We accept the reviewer's suggestion. To avoid confusion regarding the actual study area and to improve clarity, we have modified Figure 1 by retaining only subfigure (f) (the current boundary of China) and removing subfigures (a)–(e), as they were not referenced in the subsequent text. The figure caption has been updated accordingly. Please refer to lines 113–120.

We sincerely thank you for your two rounds of careful and professional reviews, which have substantially improved the quality and scientific rigor of our manuscript.